# A versatile oblique plane microscope for large-scale and high-resolution imaging of subcellular dynamics

Etai Sapoznik[1,2], Bo-Jui Chang[1], Jaewon Huh[1,2], Robert J Ju[3], Evgenia V Azarova[1], Theresa Pohlkamp[4], Erik S Welf[1,2], David Broadbent[5], Alexandre F Carisey[6], Samantha J Stehbens[3], Kyung-Min Lee[7], Arnaldo Marín[7,8], Ariella B Hanker[7], Jens C Schmidt[5,9], Carlos L Arteaga[7], Bin Yang[10], Yoshihiko Kobayashi[11], Purushothama Rao Tata[11], Rory Kruithoff[12], Konstantin Doubrovinski[1,13], Douglas P Shepherd[12], Alfred Millett-Sikking[14], Andrew G York[14], Kevin M Dean[1]*, Reto P Fiolka[1,2]*

[1]Department of Cell Biology, University of Texas Southwestern Medical Center, Dallas, United States; [2]Lyda Hill Department of Bioinformatics, University of Texas Southwestern Medical Center, Dallas, United States; [3]Institute for Molecular Bioscience, University of Queensland, Queensland, Australia; [4]Department of Molecular Genetics, University of Texas Southwestern Medical Center, Dallas, United States; [5]Institute for Quantitative Health Sciences and Engineering, Michigan State University, East Lansing, United States; [6]William T. Shearer Center for Human Immunobiology, Baylor College of Medicine and Texas Children's Hospital, Houston, United States; [7]Harold C. Simmons Comprehensive Cancer Center and the Department of Internal Medicine, University of Texas Southwestern Medical Center, Dallas, United States; [8]Department of Basic and Clinical Oncology, Faculty of Medicine, University of Chile, Santiago, Chile; [9]Department of Obstetrics, Gynecology, and Reproductive Biology, Michigan State University, East Lansing, United States; [10]Chan Zuckerberg Biohub, San Francisco, United States; [11]Department of Cell Biology, Duke University School of Medicine, Durham, United States; [12]Center for Biological Physics and Department of Physics, Arizona State University, Tempe, United States; [13]Cecil H. and Ida Green Comprehensive Center for Molecular, Computational and Systems Biology, University of Texas Southwestern Medical Center, Dallas, United States; [14]Calico Life Sciences LLC, South San Francisco, United States

*For correspondence:
Kevin.Dean@UTsouthwestern.edu
(KMD);
Reto.Fiolka@UTsouthwestern.edu
(RPF)

**Abstract** We present an oblique plane microscope (OPM) that uses a bespoke glass-tipped tertiary objective to improve the resolution, field of view, and usability over previous variants. Owing to its high numerical aperture optics, this microscope achieves lateral and axial resolutions that are comparable to the square illumination mode of lattice light-sheet microscopy, but in a user friendly and versatile format. Given this performance, we demonstrate high-resolution imaging of clathrin-mediated endocytosis, vimentin, the endoplasmic reticulum, membrane dynamics, and Natural Killer-mediated cytotoxicity. Furthermore, we image biological phenomena that would be otherwise challenging or impossible to perform in a traditional light-sheet microscope geometry, including cell migration through confined spaces within a microfluidic device, subcellular photoactivation of Rac1, diffusion of cytoplasmic rheological tracers at a volumetric rate of 14 Hz, and large field of view imaging of neurons, developing embryos, and centimeter-scale tissue sections.

## Introduction

Light-sheet fluorescence microscopy (LSFM) first generated significant interest in the biological community as a result of its ability to image developing embryos with single-cell resolution, inherent optical sectioning, low phototoxicity and high temporal resolution (*Huisken et al., 2004*). Since then, LSFM has undergone a revolution, and depending on the optical configuration, can now routinely image biological systems that range from reconstituted macromolecular complexes (*Keller et al., 2007*) to intact organs and organisms (*Chakraborty et al., 2019*; *Voigt et al., 2019*). Unlike other microscope modalities that are used to image three-dimensional specimens (e.g. confocal), LSFM delivers light to only the in-focus portion of the specimen, and therefore substantially decreases the illumination burden on the sample. Further, light-sheet excitation combines powerfully with the million-fold parallelization afforded by modern scientific cameras, permitting massive reductions in illumination intensities without compromising the signal-to-noise ratio, which significantly reduces the rate of photobleaching and phototoxicity. Consequently, LSFM enables imaging of biological specimens for 1000's of volumes (*Dean et al., 2017*).

Despite its advantages over other imaging modalities, its widespread adoption remains limited. In part, this is due to the slow adoption of cutting-edge LSFM systems by commercial entities, and consequently, the requirement that each lab assemble, align, maintain, operate their own LSFM instruments. Sample preparation is an additional problem, as the orthogonal geometry of LSFM systems often sterically occludes standard imaging dishes such as multi-well plates. Furthermore, the highest resolution LSFM systems, the reliance on high-NA water-dipping objectives places the sample in direct contact with non-sterile optical surfaces, which compromises long-term imaging. And these matters are made worse because modern LSFM systems often lack modalities that render microscopy routinely useful for non-experts, including sample environmental control (e.g. $CO_2$, temperature, and humidity), oculars, and hardware-based autofocusing schemes.

Given these concerns, several labs have worked to identify single-objective imaging systems that combine light-sheet illumination, more-traditional sample mounting, and high numerical aperture (NA) fluorescence detection. For example, using the same objective for illumination and detection, Gebhardt et al created a bespoke atomic force microscope cantilever that reflected a light-sheet into the ordinary viewing geometry of an inverted microscope (*Gebhardt et al., 2013*). Similarly, several labs developed specialized microfluidics with micromirrors positioned at 45 degrees (*Galland et al., 2015*; *Meddens et al., 2016*). Nonetheless, these approaches are only compatible with low-NA illumination (which reduces resolution and optical sectioning), have a limited volumetric imaging capacity, and require that the reflective surface be placed in immediate proximity to the specimen, which drastically limits the field of view. More recently, a high-NA oil immersion lens (1.49) was combined with lateral interference tilted excitation (*Fadero et al., 2018*). However, this system required specialized imaging chambers, used light-sheets that were several microns thick and tilted relative to the imaged focal plane, and owing to the large refractive index mismatch between the objective immersion media and the specimen, suffered from spherical aberrations, which ultimately impact the resolution and sensitivity of the microscope in a depth-dependent manner (*Fadero et al., 2018*). Thus, each of these systems are incompatible with many experimental designs, including biological workhorses like the 96-well plate and large-scale tissue sections.

There is however one form of LSFM, referred to as oblique plane microscopy (OPM), that avoids these complications (*Dunsby, 2008*). Unlike most LSFMs that require the light-sheet to be launched from the side or reflected off of a cantilever or microfluidic device, an OPM illuminates the sample obliquely and captures the fluorescence with the same objective (*Figure 1A*). As such, an OPM can be assembled using a standard inverted or upright microscope geometry and is entirely compatible with traditional forms of sample mounting (including multi-well plates), environment control, and laser-based auto-focusing. In OPM, the fluorescence captured by the primary objective is relayed in an aberration-free remote focusing (*Botcherby et al., 2012*) format to a secondary objective, which creates a 3D replica of the fluorescent signal at its focus that is then imaged by a tilted tertiary objective onto a camera (*Figure 1B*). To date, most oblique plane microscopes have had low optical resolution because the tertiary objective fails to capture all of the high-angle rays that are launched from the secondary objective (*Figure 1B*). Nevertheless, it was recently demonstrated that an OPM

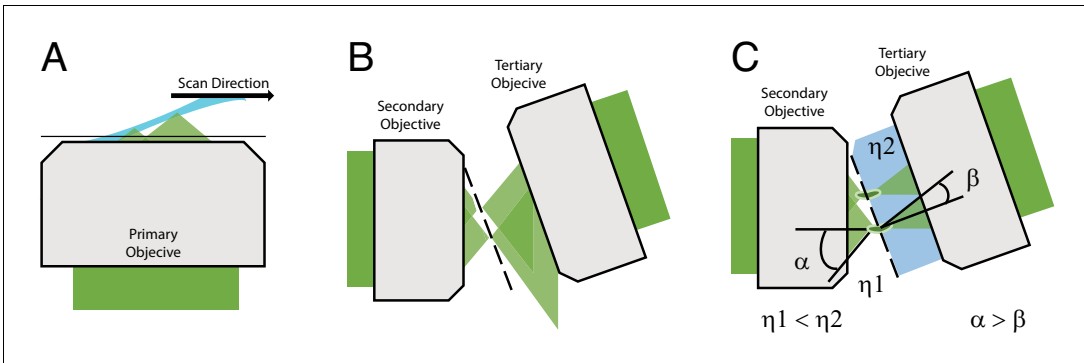

**Figure 1.** Optical principle of oblique plane microscopy. (**A**) The light-sheet, shown in light blue, is launched from the primary objective at an oblique illumination angle, and the resulting fluorescence cones of light, shown in green, are collected by the same objective. (**B**) In a traditional oblique plane microscope, a replica of the fluorescence collected from the primary objective is formed by the secondary objective at the focus of the tertiary objective. However, owing to the off-axis imaging geometry, high-angle rays cannot be captured by the tertiary objective. (**C**) If instead the light travels from a region of low refractive index ($\eta_1$, air) to high refractive index ($\eta_2$, water or glass), then the optical cone of light is compressed from an angle of $\alpha$ to $\beta$ and refracted toward the tertiary imaging system, thus permitting its capture, and maximizing the resolving power of the microscope. The online version of this article includes the following figure supplement(s) for figure 1:

**Figure supplement 1.** Geometrical considerations for the theoretical NA.
**Figure supplement 2.** Wavefront error associated with our remote focusing system.
**Figure supplement 3.** Improved illumination homogeneity with resonantly scanned multidirectional illumination.
**Figure supplement 4.** Schematic diagram of the illumination unit configured for light-sheet illumination.
**Figure supplement 5.** Schematic diagram of the illumination unit configured for widefield illumination.
**Figure supplement 6.** Schematic diagram of the illumination unit configured for light-sheet illumination with the far-red and red lasers and photo-activation with the blue laser.
**Figure supplement 7.** Schematic diagram of the oblique plane microscope.

can achieve ~300 nm scale resolution by combining air and water objectives with a coverslip-walled immersion chamber (*Yang et al., 2019*). Here, as light rays travel from a low to a high refractive index medium (e.g. air and water, respectively), they are refracted at the coverslip interface toward the tertiary objective in a manner that compresses the optical cone of light, and thereby permits capture of the higher angle rays (*Yang et al., 2019*; *Figure 1C* and *Figure 1—figure supplement 1*). Nevertheless, aligning this chamber, which requires that the coverslip be placed at the proper distance and angle relative to both the secondary and tertiary objectives, while maintaining a proper water immersion for the tertiary objective, although feasible, is technically challenging. Further, this arrangement was only compatible for secondary objectives with a maximum NA of 0.9, as higher NA objectives would collide with the water chamber.

Here, to mitigate these challenges, we built a high-NA OPM equipped with a recently developed glass-tipped tertiary objective (*Millett-Sikking and York, 2019*) that eliminates the need for an immersion chamber and further improves instrument performance (Appendix 1). Compared to other single-objective LSFMs and OPMs, we demonstrate that this OPM provides a unique and impressive combination of field of view, resolution, volumetric imaging capacity, and speed. As proof of principle, we image a number of biological processes that would otherwise be challenging to observe without this unique combination of microscope geometry, speed, resolution, and field of view, including nuclear rupture in melanoma cells as they migrate through tightly confined spaces in a microfluidic device, immunological synapse formation, cleavage furrow ingression in developmental systems, rheological cytosolic flows, optogenetic activation of Rac1, and intact imaging of an complete coronal murine brain slice.

## Results

### Microscope design

We designed an OPM capable of leveraging the maximum resolving power and field of view of a bespoke glass-tipped tertiary objective (*Millett-Sikking and York, 2019*). In this design, a high-NA primary objective (100X, NA 1.35) with an angular aperture of ~74 degrees is matched to a secondary air objective (40X, NA 0.95) with a similarly large angular aperture, which relays the fluorescence to a tertiary imaging system that is oriented 30 degrees off-axis (See Materials and methods). Other angles are possible (between 0 and 45 degrees), so long as the angle is matched to that of the light-sheet in sample space (Appendix 2). We chose a silicone immersion primary objective because the refractive index of living cells (~1.40) is closer to silicone (1.40) than water (1.333) or oil (1.52), thus reducing spherical aberrations and improving the overall imaging performance (*Phillips et al., 2012*). In cases where imaging through an aqueous boundary is unavoidable, the refractive index of the solution can be adjusted in a non-toxic manner using readily available reagents (*Boothe et al., 2017*). To achieve aberration- and distortion-free imaging, the optical train was carefully designed to properly map the pupils of the primary and secondary objectives and lenses were selected that maximized the near-diffraction-limited field of view (*Figure 1—figure supplement 2*). Inspired by tilt-invariant imaging systems (*Kumar et al., 2018*; *Voleti et al., 2019*; *Yang et al., 2019*), the microscope was equipped with a high-speed galvanometer mirror conjugate to both the primary and secondary objective pupils, allowing for rapid light-sheet scanning in sample space, and rapid emission descanning prior to detection with the camera. Because the galvanometer mirror is the only moving part, sources of optical drift are minimized, and only a portion of the camera is necessary to detect the descanned fluorescence (~256×2048), which permits very high imaging rates (~800 planes per second). Furthermore, for illumination, we developed a versatile laser launch that can be reconfigured in a fully automatic fashion to illuminate the cells with either an oblique light-sheet that is equipped with resonant multi-directional shadow suppression (*Huisken and Stainier, 2007*), wide-field, or a laser-scanned and near-diffraction-limited beam for localized optogenetic stimulation, fluorescence recovery after photobleaching, or photoactivation (See Materials and methods, *Figure 1—figure supplements 3–7*). Owing to the large number of optics, absorption and spurious reflections resulted in a 59 and 47% decrease in fluorescence transmission for the laser-scanning and stage-scanning variants of the microscope, respectively, at 30-degrees. Transmission improved only slightly (3%) when the optical system was arranged at 0-degrees, indicating that the tertiary objective was indeed capable of capturing most of the transmitted light under a 30-degree tilt. The entire microscope was built in a standard inverted format with a motorized sample stage, objective and sample heating, and a temperature and $CO_2$-regulated environmental chamber.

### Instrument characterization

An obliquely oriented light-sheet is launched from a single objective into the specimen at an angle of 30 degrees, simultaneously illuminating a two-dimensional plane along the X and S axes. By scanning the laser in the Y-direction and collecting images at each intermediate XS plane, a three-dimensional volume is acquired (*Figure 2A,B and C*). Computationally shearing places these data into its proper Euclidian context and results in a parallelepiped-shaped image volume, which is readily visualized by imaging 100 nm green fluorescent beads embedded in a 1% agarose gel (*Figure 2D*). Here, at the highest illumination NA (0.34), a narrow strip of beads coincident with the illumination beam waist appear sharp. By fitting each bead to a three-dimensional Gaussian function, the raw (e.g. non-deconvolved) axial resolution for these data are 587 ± 18 nm (mean and standard deviation of the Full-Width Half Maximum, FWHM, *Figure 2E*). As the NA of the illumination decreases, the length and thickness of the light-sheet grows. While this improves the uniformity of the resolution and contrast throughout the field of view spanned by the XS plane, it ultimately reduces the raw axial resolution to 736 ± 10 nm and 918 ± 12 nm, for NA 0.16 and NA 0.06 illumination beams, respectively (*Figure 2D and E*). For most biological experiments reported here, we used an illumination NA of 0.16, yielding a Gaussian beam that has a thickness and propagation length of ~1.2 and~37 microns, respectively. Importantly, in cases where the illumination beam is thicker than the depth of focus of the detection objective, optical sectioning (e.g. the ability to reject out-of-focus fluorescence) will be slightly reduced. Nonetheless, as is evident from the PSF and optical transfer

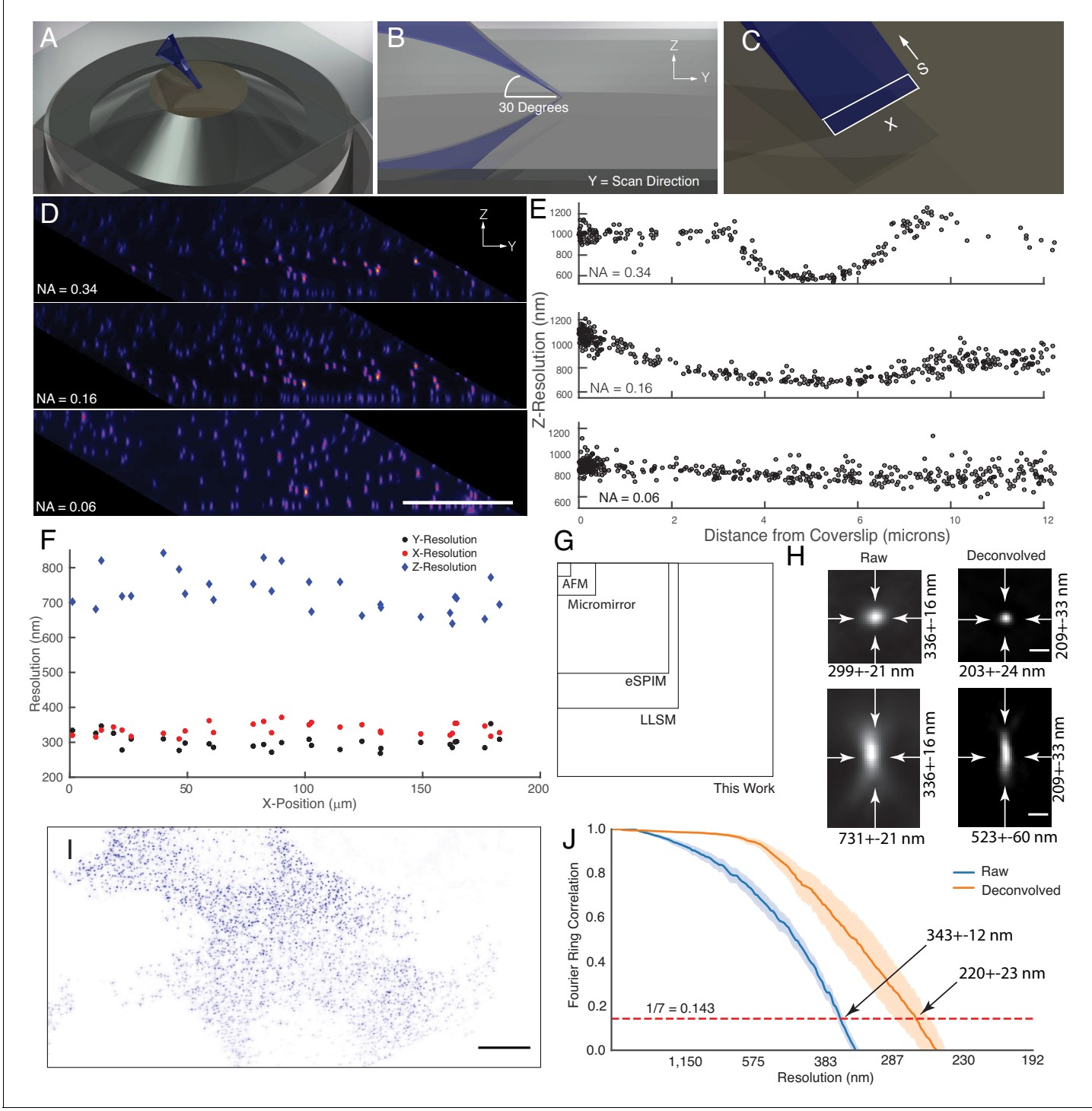

**Figure 2.** Microscope illumination geometry, light-sheet properties, resolution, and field of view. (**A**) The light-sheet is launched from a single primary objective at a (**B**) 30-degree oblique illumination angle, and rapidly scanned in the Y-direction to acquire a 3D volume. (**C**) At each intermediate position, a two-dimensional plane along the X and S axes is acquired with a scientific CMOS camera. (**D**) 100 nm fluorescent beads embedded in agarose. Sharp strip of beads reveals the position of the illumination light-sheet beam waist and scan trajectory along the Y-axis. Parallelepiped data geometry results from oblique illumination and computational shearing of the data. (**E**) FWHM of beads for different illumination light-sheet NAs, as a function of distance from the coverslip. Raw data, no deconvolution. The confocal parameter of the light-sheets projected onto the Z-axis is 3.0, 7.5, and >12 microns, for NAs of 0.34, 0.16, and 0.06, respectively. (**F**) Lateral and axial resolutions of surface immobilized fluorescent nanospheres and using an excitation NA of 0.16. Raw data, no deconvolution (**G**) Dimensions of the field of view for the OPM presented here when operated in a laser-scanning format compared to lattice light-sheet microscopy, eSPIM, a microfluidic-based micromirror, and an atomic force microscopy (AFM)-based

*Figure 2 continued on next page*

*Figure 2 continued*

cantilever. Lateral dimensions of the figure represent the X and Y axes. (**H**) Representative point-spread functions for surface immobilized fluorescent nanospheres, before and after deconvolution. Scale Bar: 500 nm. (**I**) Maximum intensity projection of a MV3 cell expressing genetically encoded multimeric nanoparticles. Deconvolved data is shown. Scale Bar: 10 microns. (**J**) Fourier Ring Correlation analysis of intracellular resolution. Solid lines show mean value, shaded area is the 95% confidence interval.

The online version of this article includes the following figure supplement(s) for figure 2:

**Figure supplement 1.** Rotationally averaged optical transfer functions for a widefield microscope and our OPM with increasing numerical aperture of the light-sheet illumination.

**Figure supplement 2.** Axial view of 100 nm fluorescent microspheres in agarose imaged with an NA of 0.06 for light-sheet excitation.

**Figure supplement 3.** Comparison of raw PSFs obtained with OPM, lattice light-sheet microscopy and spinning-disk microscopy.

**Figure supplement 4.** Lateral XY field of view of 100 nm fluorescent microspheres in agarose obtained with an optical scan along the Y-axis.

**Figure supplement 5.** Lateral resolution measured at different depths using 100 nm green fluorescent beads.

function, the OPM presented here delivers adequate optical sectioning over the full range of excitation NAs used in this manuscript (*Figure 2—figure supplement 1*). Theoretically, the maximum imaging depth of our remote focusing system is ~60 microns, beyond which tiling in the Z-dimension can be performed until one reaches the working distance of the primary objective (300 microns). Of note, the choice of illumination angle is accompanied by tradeoffs in light-sheet thickness, imaging depth, detection efficiency, and resolution (Appendix 2). Indeed, we observed a gradual loss in NA and thus resolution as our tertiary imaging system was adjusted from a 0 to a 30-degree tilt. Unlike single-objective light-sheet systems that use oil immersion objectives (*Fadero et al., 2018*), spherical aberrations are not immediately evident (*Figure 2D and E*, and *Figure 2—figure supplement 2*). For an oblique illumination angle of 30 degrees and an excitation NA of 0.16, the raw axial resolution of our system is similar to the raw 666 nm axial resolution reported for the most commonly used square illumination mode of lattice light-sheet microscopy (*Chang et al., 2020*; *Valm et al., 2017*) or a spinning-disk microscope using the same NA 1.35/100X objective used here (*Figure 2—figure supplement 3*).

In an effort to more-systematically evaluate microscope performance, we also measured the resolution for 100 nm coverslip-immobilized fluorescent beads. Here, we used an illumination light-sheet with an NA of 0.16. For raw data, we measured a resolution of $299 \pm 21$, $336 \pm 16$, and $731 \pm 21$ nm in X, Y, and Z, respectively, throughout a lateral field of view of ~180 x~180 microns (*Figure 2F*). In the central ~60 x~60 micron portion of the field of view, a slightly improved lateral resolution was observed ($284 \pm 12$ and $328 \pm 14$ for X and Y, respectively). Nonetheless, the resolution was relatively uniform throughout the full ~180 x~180 micron field of view (e.g. the footprint of the imaging volume in X and Y), which importantly is 2.6x, 3.8x, 37.9x, and 268x larger than reported for lattice light-sheet microscopy (*Chen et al., 2014*), eSPIM (*Yang et al., 2019*), micromirror (*Galland et al., 2015*; *Meddens et al., 2016*), and atomic force microscopy (AFM) cantilever-based methods (*Gebhardt et al., 2013*), respectively (*Figure 2G*). Of note, the field of view in lattice light-sheet microscopy, eSPIM and the OPM presented here is bounded in the X-dimension by the optics of the microscope and the finite size of the illumination beam and the camera chip size. In contrast, the Y-dimension is essentially unlimited in a sample scanning format. Nonetheless, through careful optical design, the lateral extent of our imaging field (e.g. the X- dimension) in a laser-scanning format is ~2.6 x larger than those reported for eSPIM (*Yang et al., 2019*). This is an important design parameter for live-cell microscopy, since laser-scanning based imaging is much more rapid than sample scanning and tiling approaches.

As an alternative resolution estimation metric, we also applied image decorrelation analysis, which evaluates the cross-correlation in frequency space between an image and its frequency-filtered equivalent (*Descloux et al., 2019*). Decorrelation analysis for the 180 × 180 micron field of view resulted in an aggregate raw lateral resolution (e.g. the average of both the X and Y dimensions) of $325 \pm 25$ nm (mean and standard deviation, *Figure 2—figure supplement 4*), which is in good agreement with our FWHM measurements. Deconvolution is commonly used in LSFM to improve image contrast and resolution. Here, 20 iterations of Richardson-Lucy deconvolution yielded resolutions of $203 \pm 24$, $209 \pm 33$, and $523 \pm 60$ nm in X, Y, and Z, respectively, and representative point spread functions for both raw and deconvolved data are shown in *Figure 2H*. Furthermore, microscope performance remained robust even when imaging beyond the nominal focal plane of the

primary objective (*Figure 2—figure supplement 5*). Importantly, a point-spread function obtained by imaging a sub-diffraction bead is sufficient to describe the resolution and optical sectioning capacity of a fluorescence microscope. Nonetheless, sub-diffraction beads do not necessarily capture the optical complexities of the intracellular environment. Thus, we also sought to estimate instrument performance using fixed cells expressing genetically encoded multimeric nanoparticles (GEMs), which are self-forming cytosolic 40 nm diameter icosahedral assemblies of fluorescent proteins (*Figure 2I*). Using Fourier Ring Correlation analysis, which evaluates the spatial frequency-dependent signal-to-noise for a pair of images, we measured an aggregate lateral resolution of $343 \pm 12$ and $220 \pm 23$ nm, for raw and deconvolved GEMs, respectively (*Figure 2J*). These resolution values were also in close agreement with those obtained with decorrelation analysis ($322 \pm 20$ and $251 \pm 3$ nm, raw and deconvolved mean and 95% confidence intervals, respectively). By comparison, the raw lateral resolution for lattice light-sheet microscopy for a GFP-like fluorophore is 312 nm (*Valm et al., 2017*). Thus, when compared to other single-objective LSFMs, OPMs, and even lattice light-sheet microscopy, the OPM presented here achieves a unique combination of resolution, field of view, and rapid imaging.

## Biological imaging of clathrin-mediated endocytosis, vimentin, and membrane dynamics

To evaluate microscope performance on biological specimens, we first imaged the endoplasmic reticulum in U2OS osteosarcoma cells (*Figure 3A*, and *Video 1*). Endoplasmic reticulum tubules were highly dynamic, and unlike methods that rely on imaging slightly out of a total internal reflection geometry (*Li et al., 2015*), could be imaged with high-resolution throughout the entire cell volume. For comparison, an image of the endoplasmic reticulum without deconvolution is provided in *Figure 3—figure supplement 1*. We also imaged vimentin in retinal pigment epithelial cells, which is an intermediate filament that is often associated with the epithelial to mesenchymal transition and hypothesized to reinforce polarity cues through crosstalk in the microtubule, actin, and integrin-mediated adhesion cellular systems. Here, vimentin appeared as moderately dynamic filamentous structures that extended from the perinuclear region to the cell periphery (*Figure 3B*, and *Video 2*). Vimentin filaments occupied both the apical and basal sides of the cell, as is visible in an axial cross-section through the cell (*Figure 3C*). To evaluate our ability to image more dynamic processes, we imaged clathrin -mediated endocytosis in retinal pigment epithelial cells that were labeled with the clathrin adapter protein AP2 fused to GFP (*Figure 3D and E*, *Videos 3* and *4*). Endocytosis could be observed through time, with individual endocytic pits appearing as point-like structures that initialized on the plasma membrane, locally diffused, and then disappeared upon scission and release into the cytosol. Lastly, we imaged MV3 melanoma cells tagged with a membrane marker. These cells displayed numerous dynamic cellular protrusions, including blebs and filopodia, which extended away from the coverslip and otherwise could not have been observed without high-spatiotemporal volumetric imaging (*Figure 3F–H*, *Video 5*). In particular, we could observe short lived filopodial buckling events (*Figure 3—figure supplement 2*).

## Natural killer cell mediated cytotoxicity

Natural Killer (NK) cells are one of the main components of the innate immune system and are able to directly recognize and destroy virally infected or oncogenically transformed cells via the formation of a multi-purpose interface called the immunological synapse with their target (*Mace et al., 2014*). An early landmark of this structure is the establishment of a complex actin scaffold (on the effector side) which formation (*Mace and Orange, 2014*) and dynamism (*Carisey et al., 2018*) are known to be critical to ensure the productive outcome of the cytotoxic process. By contrast, the contribution of the target cell in the establishment and maintenance of this unique structure, and the interplay between the plasma membranes of the two cells remains poorly understood, in part because most work has been performed on ligand-coated surfaces (*Brown et al., 2011*; *Carisey et al., 2018*). Much could be gained by imaging heterotypic cell-cell interactions directly, but this is particularly challenging as it requires a combination of resolution, speed, and field of view as these interactions are random and often short lived. Here, we imaged the formation of the cytolytic immunological synapse between a population of NK cells expressing a reporter for filamentous actin and myelogenous leukemia cancer cells expressing a fluorescent marker highlighting their plasma membrane

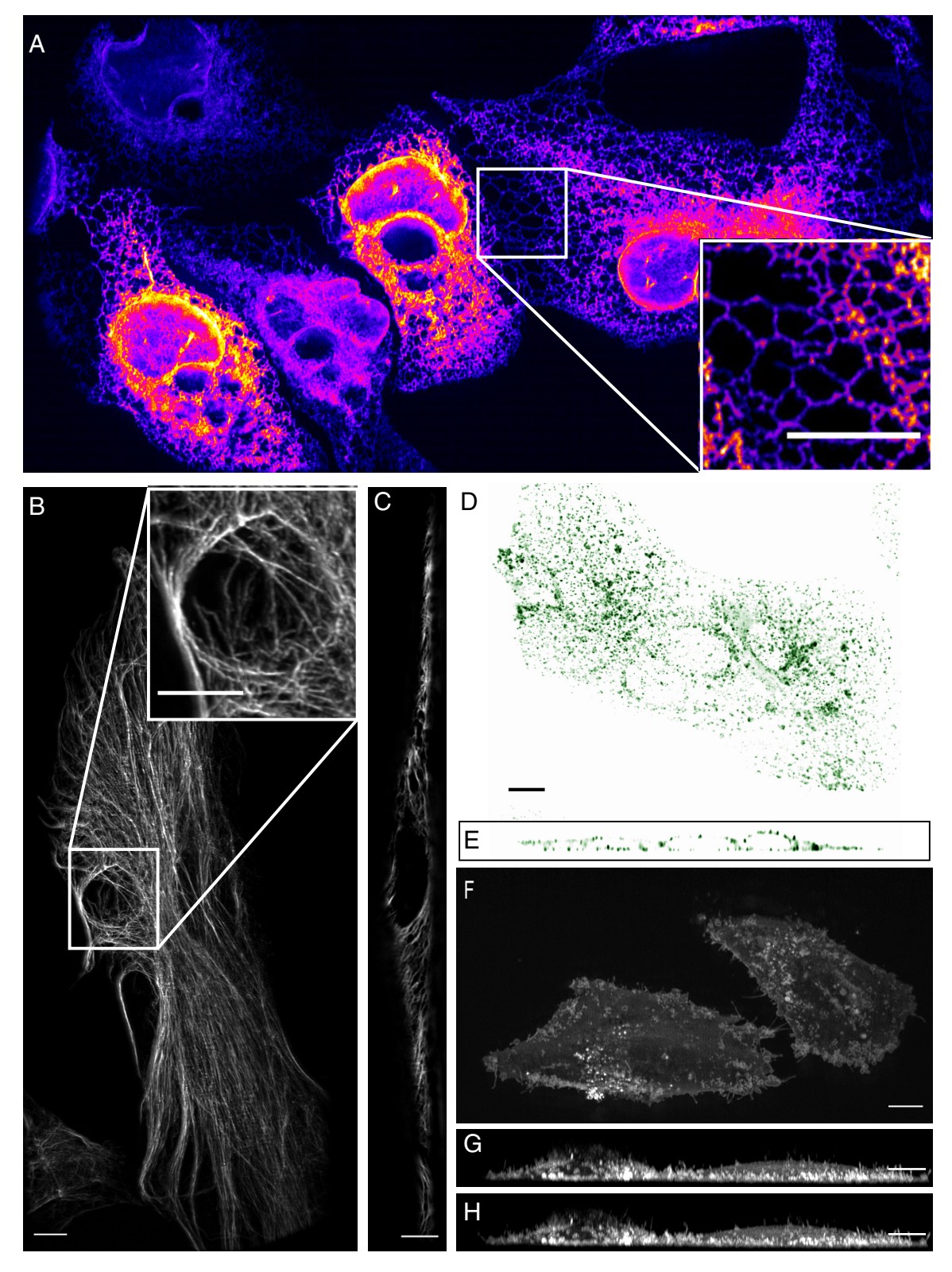

**Figure 3.** High-resolution biological imaging. (**A**) Endoplasmic reticulum in U2OS cells. Inset shows fine details in the dense, tubulated network. The lookup table was selected as it allows visualization of both bright and dim structures. (**B**) Vimentin in RPE hTERT cells. (**C**) Single slice through vimentin network. (**D**) Lateral and (**E**) axial view of clathrin-mediated endocytosis in ARPE cells. (**F**) Cortical blebs in MV3 melanoma cells. (**G**) Cross-section through MV3 cells at the 6$^{th}$ and (**H**) 12$^{th}$ time point. All data shown in this figure was deconvolved. All Scale Bars are 10 microns.

*Figure 3 continued on next page*

*Figure 3 continued*

The online version of this article includes the following figure supplement(s) for figure 3:

**Figure supplement 1.** Maximum intensity projection of endoplasmic reticulum in U2OS osteosarcoma cells, without deconvolution.
**Figure supplement 2.** Maximum intensity projection of filopodial buckling events.

(*Figure 4A*). Upon engaging with the cancer cell, the NK cell formed an immunological synapse with rapid actin accumulation in the plane of the immune synapse followed by the establishment of an actin retrograde flow along the cell axis in a similar fashion to what has been observed in T cells (*Chen et al., 2014*; *Ritter et al., 2015*). Interestingly, long tethers of membrane were pulled from the target cell synchronously with the actin retrograde flow (*Figure 4B,C and D*, *Video 6*). This could be a direct visualization of the first stage of trogocytosis, an important consequence of mechano-transduction at the immunological synapse that contributes to hypo-responsiveness in cytotoxic cells (*Miner et al., 2015*). Indeed, recent work performed on cytolytic T cells emphasized the crucial role of tension transmitted through the ligand-receptor axis on the organization of the underlying actin cytoskeleton (*Kumari et al., 2020*). Such observations highlight the importance of moving away from artificial ligand-coated surfaces, and to evaluate biological processes in more relevant contexts as enabled by high-resolution, high-speed, volumetric imaging.

## Imaging in biological microchannels – microtubules and nuclear shielding

Cells migrating through 3D microenvironments such as dense stromal tissues must navigate through tight pores between matrix fibers and are thus rate-limited by the cross-sectional diameter of their nucleus (*Wolf et al., 2013*). Understanding how cells adapt and effectively navigate these complex microenvironments is fundamental to multiple biological processes such as development, tissue homeostasis, wound healing and dysregulated in cancer cell invasion and metastasis. Our understanding of the mechanisms governing nuclear movement and protection in 3D environments is constantly emerging. For example, we now know that during 3D migration cells can tolerate and repair nuclear constriction events that cause nuclear herniation, rupture and DNA damage (*Denais et al., 2016*; *Raab et al., 2016*). Unfortunately, observing these events in vivo remains low throughput. In contrast, microfluidic devices provide an accurate and reproducible model of this phenomenon whereby cells can be subjected to precisely-defined mechanical constrictions with unique sizes and shapes that serve to recapitulate the biological microenvironment (*Garcia-Arcos et al., 2019*; *Raab et al., 2016*). Nonetheless, imaging this biological process three-dimensionally requires a large field of view, and both high spatial and temporal resolution. Unfortunately, confocal microscopes are accompanied by excessive phototoxicity to permit longitudinal observation of migration through confined microchannels. While light-sheet microscopy is an attractive alternative, the glass-polydime-thylsiloxane sandwich geometry of microfluidic devices is not compatible with traditional LSFMs that use water-dipping objectives and an orthogonal illumination and detection geometry, let alone can-tilever- or micromirror-based methods. Here, using our OPM, we were able to circumvent this and volumetrically image nuclear positioning and microtubule dynamics as cells navigated mechanical constrictions (*Figure 5A*, *Video 7*). Here, the microfluidic device consists of ~4 microns tall large and small circular posts with constriction junctions between large and small posts of 2.5 microns and two microns, respectively (*Figure 5B*). When cells were allowed to migrate within these microchannels, cells generated long, microtubule-rich protrusions, and migrated in a polarized manner. The nucleus was visibly compressed when viewed in the axial direction (*Figure 5C*) and appeared to be surrounded by microtubules on both the apical and

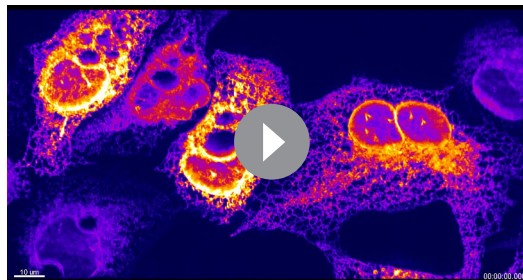

**Video 1.** Endoplasmic reticulum dynamics in osteosarcoma U2OS cells expressing Sec61-GFP. Time Interval: 0.84 s. Scale Bar: 10 microns.
https://elifesciences.org/articles/57681#video1

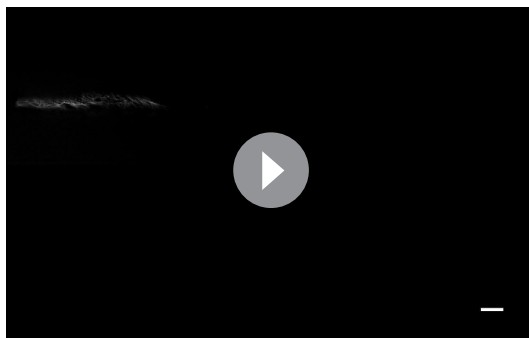

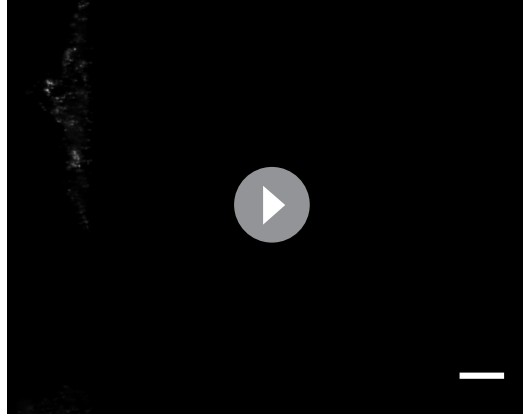

**Video 2.** 3D stack of RPE hTERT cells expressing GFP-vimentin. Data has been deconvolved and sheared into its proper Euclidian position. Scale Bar: 20 microns.
https://elifesciences.org/articles/57681#video2

**Video 3.** 3D stack of ARPE cells tagged with AP2-GFP, a marker for clathrin-mediated endocytosis. Data has been deconvolved and sheared into its proper Euclidian position. Scale Bar: 10 microns.
https://elifesciences.org/articles/57681#video3

basal surfaces of the nucleus (*Figure 5D and E*). Importantly, as these cells squeezed through the microfluidic device, they adopted particularly elongated morphologies (~80 microns) that otherwise would be challenging to observe if it were not for the large field of view of our OPM.

## High-speed imaging of calcium transduction and cytoplasmic flows

Three-dimensional imaging typically requires scanning heavy optical components (e.g. the sample or the objective), which limits the volumetric image acquisition rate. Nevertheless, because the OPM described here adopts a galvanometer-based scan-descan optical geometry, camera framerate-limited imaging is possible (*Kumar et al., 2018*; *Voleti et al., 2019*; *Yang et al., 2019*). Thus, we sought to image fast biological processes, including calcium wave propagation and the rapid diffusion of cytoplasmic tracers. For the former, we used the small-molecule calcium sensor Fluo-3, and imaged rat primary cardiomyocytes at a volumetric image acquisition rate of 10.4 Hz (*Figure 6A*, *Video 8*). Here, imaging was sufficiently fast to observe calcium translocation during spontaneous cardiomyocyte contraction. Such imaging can improve the understanding of single-cell calcium waves which is important for cardiac physiology and disease (*Gilbert et al., 2020*). Likewise, we also evaluated the rheological properties of the cytoplasm by imaging in live cells genetically encoded

multimeric nanoparticles. These nanoparticles appeared as near-diffraction-limited puncta that rapidly diffused and thus served as inert cytoplasmic tracers, and which we were able to image and track (*Jaqaman et al., 2008*) over 100 time points at a volumetric image acquisition rate of 13.7 Hz (*Figure 6B*, *Video 9*). Such tracking can help us understand how diffusion varies for different morphological domains, as well as how phenotypic changes alter cellular mechanics (*Delarue et al., 2018*; *Hannezo and Heisenberg, 2019*). Nonetheless, these questions can only be answered if equipped with the volumetric image acquisition speeds demonstrated here.

## Simultaneous volumetric imaging and optogenetic stimulation

In addition to its effects on proliferation and survival (*Mohan et al., 2019*), Rac1 also drives changes in cell shape and migration. Here,

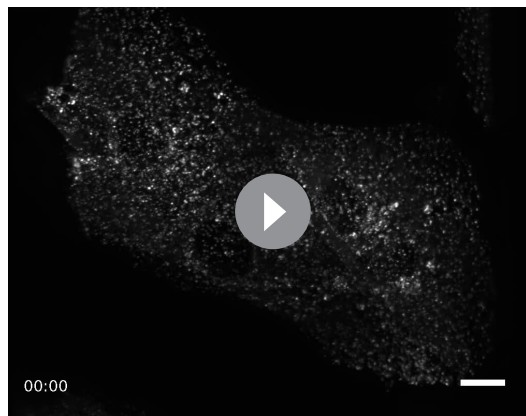

**Video 4.** Maximum intensity projection of ARPE cells tagged with AP2-GFP, a marker for clathrin-mediated endocytosis. Time Interval: 1.34 s. Scale Bar: 20 microns.
https://elifesciences.org/articles/57681#video4

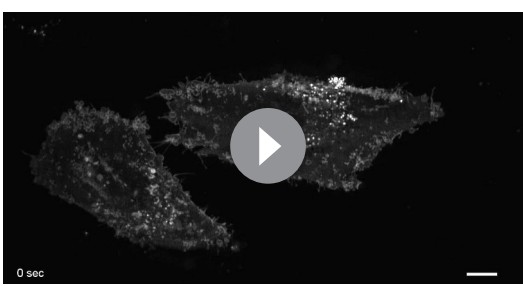

**Video 5.** MV3 cells expressing the biorthogonal membrane marker, CAAX-Halo-Tag, labeled with Oregon Green. Time Interval:1.09 seconcs, Scale Bar: 10 microns.

https://elifesciences.org/articles/57681#video5

leveraging our ability to perform simultaneous volumetric imaging and optical stimulation (*Figure 7A and B*, *Video 10*), we deployed a photoactivatable variant of Rac1 (PA-Rac1) in mouse embryonic fibroblasts (*Wu et al., 2009*). Subcellular optical stimulation of cells expressing PA-Rac1 resulted in large-scale dorsal ruffles that propagated from the cell edge near the activation region toward the cell nucleus (*Figure 7C and D*, *Videos 11* and *12*). Importantly, such dorsal ruffles would not be visible unless imaged volumetrically with high spatio-temproral resolution. By analyzing the protrusion dynamics in multiple cells (See Materials and methods), those expressing PA-Rac1 (N = 7) showed statistically significant increases in pro-trusion speed (p=0.04) and protrusion duration (p=0.02), but not the frequency of protrusion-retraction dynamics, upon photoactivation. Importantly, optically stimulated control cells (N = 6) did not show any significant change in protrusion speed, duration, or frequency in response to photoactivation (*Figure 7E*).

## Large field of view imaging of cortical neurons and ventral furrow formation

Neurons rapidly transduce action potentials across large spatial distances via their axonal or den-dritic arbors, respectively. However, many neuronal features, including synaptic boutons, are sub-micron in scale. Thus, imaging neuronal processes requires a combination of field of view, resolution, and speed (*Figure 8A*). Here, we imaged cortical neurons expressing GCaMP6f at a volumetric imaging speed of 7 Hz, and readily visualized both small scale morphological features as well as rapid action potentials (*Video 13*). Another field that benefits from fast imaging of large volumes is developmental biology, which aims to longitudinally track cell fate throughout each stage of embry-ological development. Indeed, many developmental programs, including ventral furrow ingression in *Drosophila*, are inherently three-dimensional as cells are rapidly internalized on the timescale of a few minutes along the anteroposterior axis of the embryo which spans ~230 microns. Here, we imaged ventral furrow ingression in a stage 6 Drosophila embryo (*Figure 8B*, *Figure 8—figure supplement 1*, *Video 14*). The formation of the ventral furrow, which is seen as a prominent groove run-ning along the anteroposterior axis of the embryo is clearly visible. As ventral furrow formation completes, germ band extension is initiated, and cells move toward the ventral midline and interca-late between one another (*Blankenship et al., 2006*). This global cell movement causes the tissue to elongate, which is immediately followed by rapid cellular rounding and mitotic events.

## Tissue-scale imaging

In addition to the rapid laser scan/descan illumination geometry, OPM is also compatible with a sam-ple scanning acquisition format that is essentially field of view unlimited. Indeed, by combining scan optimized equipment with fully automated fluidic handling, it is possible to image ~1 cm$^2$ of a thin tissue in less than 45 min per color and perform biochemistry, such as sequential multiplexed label-ing. To demonstrate this, we imaged an entire 30-micron thick slice of coronal mouse brain tissue (*Figure 9A*, *Video 15*) labeled with the nuclear marker DAPI. Within these data, even small features like nucleoli are clearly resolved from both lateral and axial viewing perspectives throughout the entire ~6×8 mm tissue slice (*Figure 9B and C*). Likewise, we also imaged a ~ 4×14 mm slice of 12-micron thick human lung tissue labeled for nuclei, angiotension-converting enzyme 2 (*ACE2*) mRNA, and surfactant protein C (*SFTPC*) protein (*Figure 9D*). Here, characteristic histological features, including bronchiole, alveoli and vasculature, are readily visible, albeit with molecular contrast and sub-cellular resolution (*Figure 9E,F and G*, and *Video 16*). Quantification of molecular expression within this tissue section provides spatial information on ~20,000 cells, and verifies our previous lim-ited quantification of *ACE2* expression in alveolar epithelial type II cells using confocal microscopy

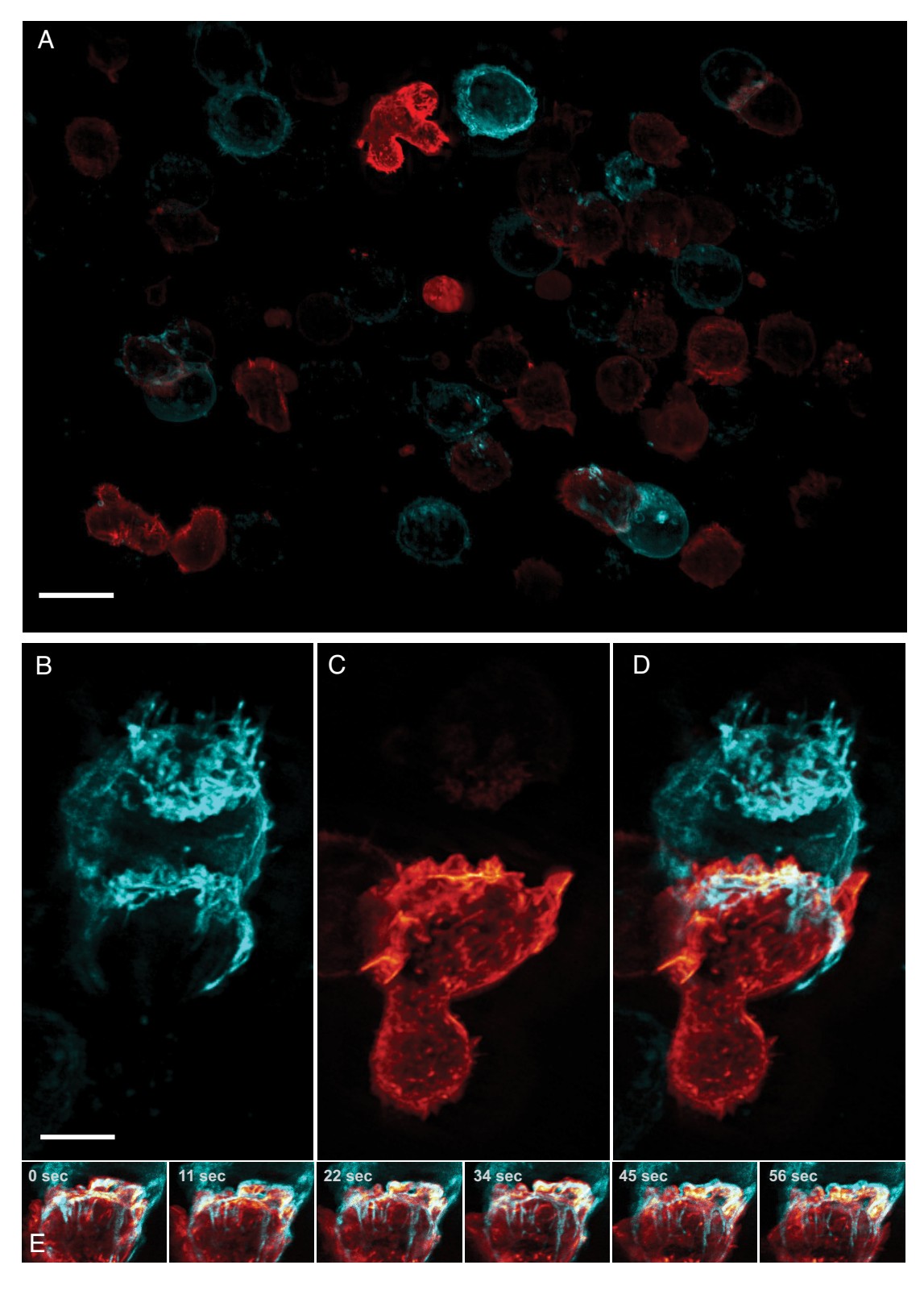

**Figure 4.** Formation of an immunological synapse between a natural killer cell and a target cell. (A). Subset of imaging field of view, showing a population of NK-92 natural killer cells expressing Life-Act-mScarlet and K562 leukemic cells expressing Lck-mVenus, which is myristolyated and localizes to the plasma membrane. Scale Bar: 20 microns. (B). K562 leukemic cell. (C) NK-92 natural killer cell. (D) Overlay of NK-92 and K562 cells during the formation and maturation stages of the immunological synapse. Scale Bar: 5 microns. (E) Upon formation of the synapse, centrifugal flows

*Figure 4 continued on next page*

driven by the NK-92 cell results in displacement of the K562 cellular membrane highlighted by the presence of membrane tethers, visible in the left panel and emphasized in the color merged frame sequence (bottom). Data is shown as maximum intensity projection and was deconvolved. Time Interval: 11.33 s.

(*Muus and Luecken, 2020*). Indeed, because we were not sterically restricted by the orthogonal illumination and detection geometry (*Figure 9—figure supplement 1*), the lateral dimensions of this human lung specimen were 8- and 1.5-fold larger than those of the biggest sample imaged with lattice light-sheet microscopy (*Gao et al., 2019*). However, in the third dimension, lattice light-sheet microscopy has in principle a 6.7x larger reach (2 mm working distance of the typically employed NA 1.1/25X detection objective compared to 300 microns working distance of our primary objective). In practice, optical aberrations limit high-resolution light-sheet microscopy to depths of a few hundreds of microns, even for highly transparent samples. Furthermore, our approach is fully compatible with automated fluid exchange, which is increasingly important for projects like the Human Cell Atlas that necessitate iterative imaging approaches for spatial -*omics* of RNAs and proteins at the single-cell level throughout entire tissues (*Chen et al., 2015*).

## Discussion

High-resolution light-sheet microscopy has yet to be widely adopted in biological laboratories and core facilities owing to routine problems with sample drift, contamination, and lack of user friendliness. Here, we show that an OPM with customized optics overcomes these challenges, and combines the ease of traditional sample mounting, environment maintenance, and multi-position stage control with the gentle, subcellular imaging afforded by selective plane illumination. For example, cells were easily identified in a traditional epi-fluorescence format prior to volumetric imaging in the light-sheet mode, the focus was maintained with readily available hardware solutions, and the environment remained sterile with $CO_2$, humidity, and temperature control. In addition to its ease of use, the OPM described here delivers spatial resolution that is on par with lattice light-sheet microscopy in its most commonly used square lattice illumination mode (*Chen et al., 2014*), albeit with a larger field of view and a volumetric imaging speed that is only limited by the maximum camera framerate and the emitted fluorescence photon flux. And unlike state-of-the-art multiview LSFM techniques that achieve a better axial resolution after image fusion and deconvolution, only a single imaging perspective is needed (*Guo et al., 2020*; *Wu et al., 2013*). Higher axial resolution can be achieved with Axially Swept Light-Sheet Microscopy (*Dean et al., 2015*) or the less commonly used hexagonal and structured illumination modes of lattice light-sheet microscopy. However, this requires a separate illumination objective, which introduces steric limitations, and is accompanied by additional drawbacks that include a shorter effective exposure time in Axially Swept Light-Sheet Microscopy, increased amounts of out-of-focus blur for hexagonal lattices, or the acquisition of five images per plane for structured illumination.

This is in stark contrast to previous generations of OPM (*Bouchard et al., 2015*; *Dunsby, 2008*; *Kumar et al., 2011*), which provided only moderate spatial resolution or a limited field of view. This revolution in OPM performance was triggered by the insight that one could combine a high-NA (~0.9) air secondary objective with a high-NA (~1.0) water-immersion tertiary objective (*Yang et al., 2019*). In such a system, the refractive index interface between the two objectives compresses and refracts the optical cone of light toward the

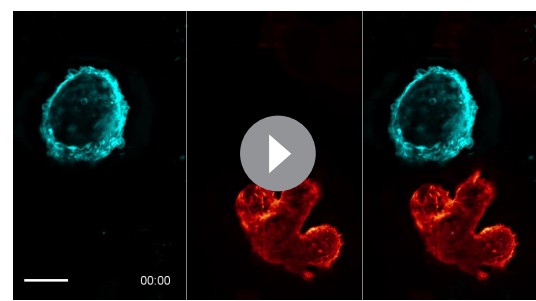

**Video 6.** An NK-92 natural killer cell forming an immunological synapse with a target cell. The NK-92 cell (Natural Killer cell line) was labeled with Life-Act-mScarlet, and is shown in orange. The target cell (K562 leukemia cell line) was labeled with Lck-mVenus and is shown in cyan. Time Interval:11.33 s. Scale Bar: 10 microns.

https://elifesciences.org/articles/57681#video6

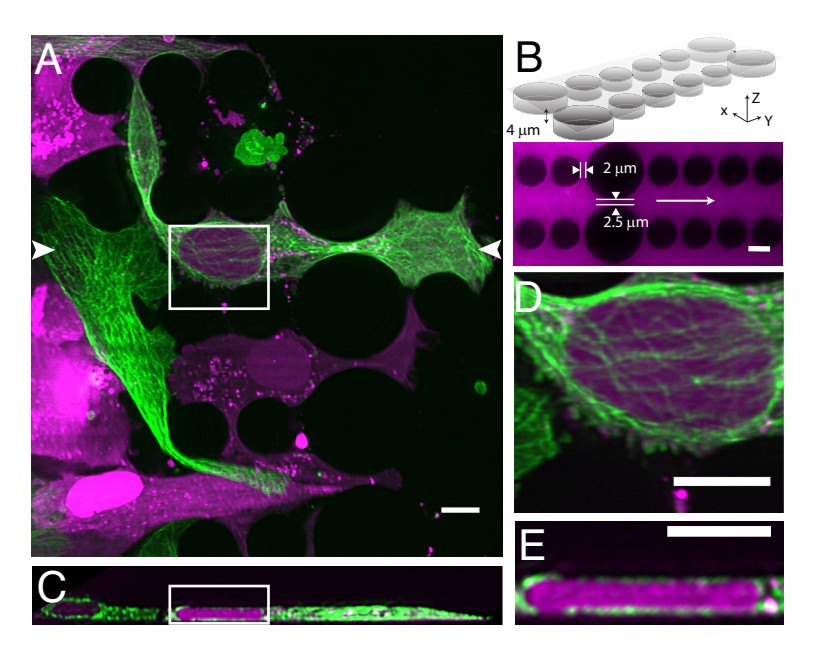

**Figure 5.** Cell migration through PDMS micro-confinement channels. (**A**) 1205Lu metastatic melanoma cells endogenously expressing eGFP-α-tubulin tagged microtubules with CRISPR (green) and a nuclear-localizing red fluorescent protein (3XNLS-mScarlet-I; magenta), migrating through a PDMS microchannel device. (**B**) Schematic drawing and fluorescence image of microfluidic device filled with TRITC-Dextran, where cells migrate in the horizontal dimension and squeeze between the pillars (large pillars are separated by 2.5 microns, and small pillars are separated by two microns). The white arrow in the fluorescence image marks the migration direction of the cells and also the scan direction (Y) of the light-sheet. (**C**) Axial cross-section of cells in a microchannel device shows top-down nuclear confinement as cells migrate through 4-micron tall channels. (**D**) Microtubule protofilaments wrap around both the basal and apical surfaces of the cell when migrating through confined spaces. (**E**) Zoom of the region shown in C. All data with the exception of (**B**) was deconvolved. All scale Bars: 10 microns.

tertiary objective and improves both sensitivity and resolution of the entire imaging system. In this work, we take this concept to its extreme with an optimized optical train, and we replace the tertiary objective (and its water chamber and coverslip) with a solid immersion objective that eases alignment and more efficiently compresses and refracts the optical cone of light (*Millett-Sikking and York, 2019*). We characterize the performance of this system and demonstrate that it has a lateral and axial resolving power that is similar to or better than many LSFM systems. We note that the theoretical NA of 1.28 would in principle allow even higher spatial resolution. Indeed, when using a zero-tilt angle for the tertiary objective, the system routinely delivered 270 nm scale raw lateral resolution across its field of view. This indicates that with reduced tilt angles, even higher resolution as demonstrated here should be possible. Why the resolution dropped notably in the tilt direction in this work is still under ongoing investigation.

As the volumetric image acquisition rate is not limited by piezoelectric scanning of either the sample or the objective, but rather a high-speed galvanometric mirror, very high temporal resolution is possible. Here, we demonstrated volumetric subcellular imaging at rates of 10 Hz, which permitted tracking of intracellular flows and calcium propagation. To demonstrate its utility, we performed a variety of imaging tasks that would be hard or impossible to perform on a traditional LSFM, including imaging in a microfluidic channel and the reproducible subcellular optogenetic activation of Rac1. Indeed, as shown by imaging cm-scale tissue slices, the stage-scanning variant of this OPM also permits applications that would otherwise not be feasible with comparable high-resolution light-sheet platforms.

As such, we believe that this OPM could displace laser-scanning and spinning-disk confocal microscopes as the workhorse of cell biology in both individual labs and user facilities. Importantly,

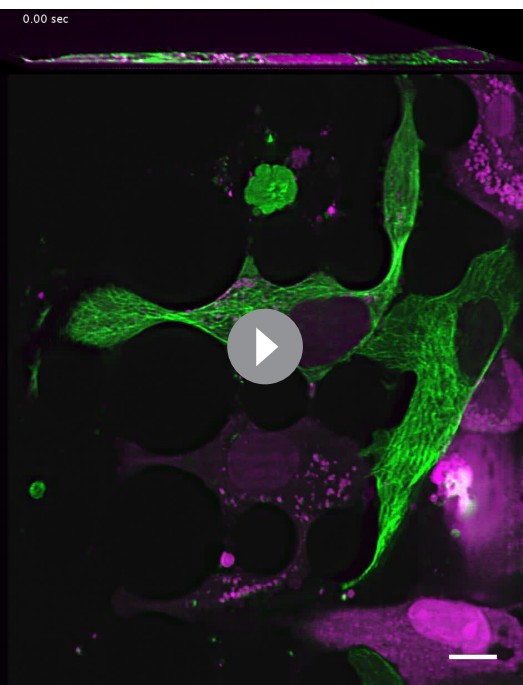

**Video 7.** 1250Lu metastatic melanoma cells expressing GFP-alpha-tubulin (green) and a 3XNLS-mScarlet-I nuclear marker (magenta). Nuclei often undergo compression and rupture as cell migrate and squeeze through pillars. Time Interval: 29.88 s. Scale Bar: 10 microns.

https://elifesciences.org/articles/57681#video7

its design allows integration into existing epi-fluorescence frameworks, which decreases the cost of building such an instrument. Furthermore, it combines spatial resolution of a spinning-disk microscope (*Figure 2—figure supplement 3*) with high volumetric acquisition speed and low phototoxicity afforded by LSFM. It provides comparable spatial resolution to leading light-sheet technologies, including the square illumination mode of lattice light-sheet microscopy, but with simpler sample handling, maintenance of a sterile environment, the ability to perform simultaneous multicolor imaging (*Chang et al., 2019*), and an essentially unlimited field of view in a sample scanning format. The principle drawback that is inherent to OPM systems is the reduced collection efficiency, which necessarily results from the large number of optics necessary to reorient the fluorescence emission (*Kim et al., 2019*). While these losses are non-negligible, they are in part offset by the higher overall NA of our OPM system relative to other light-sheet microscopes (Appendix 3), For example, in the absence of aberrations, and assuming a lower bound for the overall NA of 1.2 to collect at best ~1.35 and ~2.56 times more photons than a NA 1.1 or NA 0.8 objective, respectively. Consequently, for a given laser power, we achieve image contrast and rates of photobleaching comparable to lattice light-sheet microscopy (*Figure 9—figure supplement 2*). And lastly, we reported only on the highest resolution OPM variant that incorporates the bespoke glass-tipped tertiary objective. Indeed, many different variants of OPMs that operate across a range of NAs (1.0–1.35) and magnifications (20 - 100X) have already been designed and

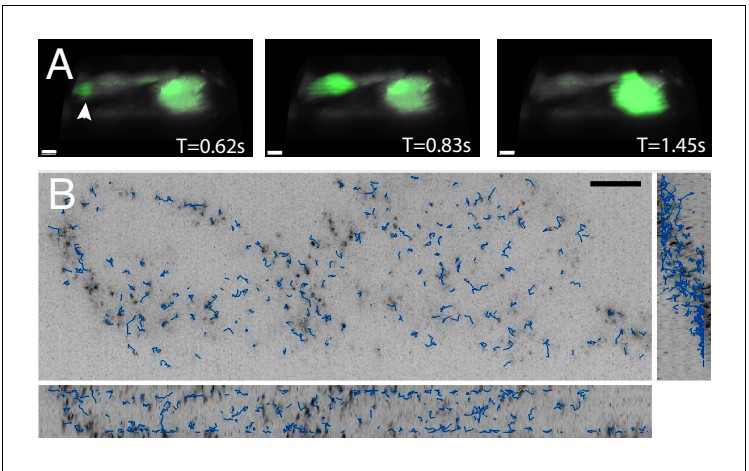

**Figure 6.** High-speed volumetric imaging of calcium waves and genetically encoded multimeric nanoparticles. (**A**) Primary rat cardiomyocytes were labeled with the small-molecule sensor, Fluo-3, and imaged at 10.4 Hz. Deconvolved data is shown. Scale Bar: 10 microns. (**B**) Imaging rheological tracers in the mammalian cytosol at 13.7 Hz. Deconvolved data is shown. Scale Bar: 10 microns.

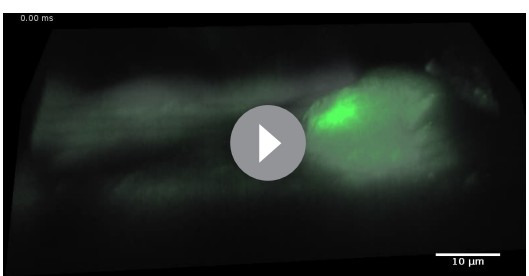

**Video 8.** 3D rendering of primary cardiomyocyte stained with Fluo-3, a small-molecule sensor for calcium (II). Green indicates fluctuations in intracellular calcium levels, and gray represents the cell boundary (calculated as an average of all imaging frames). Time Interval: 0.096 s. Scale Bar: 10 microns.
https://elifesciences.org/articles/57681#video8

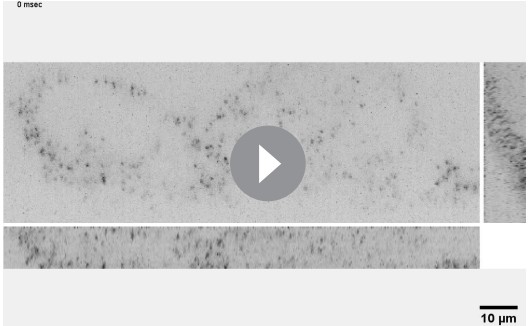

**Video 9.** Orthogonal maximum intensity projections of MV3 cells expressing cytosolic GEMs as rheological tracers. Particles were detected and tracked with the uTrack-3D software package. Time Interval: 0.073 s.
https://elifesciences.org/articles/57681#video9

numerically evaluated. Thus, we consider this the opening prelude to what may be the next generation of user-friendly, and broadly accessible LSFMs.

## Materials and methods

### Laser-scanning microscope setup

The entire microscope was built in a basic inverted geometry (RAMM-FULL, Applied Scientific Instrumentation) with a three-axis motorized stage controller. Two solid-state continuous wave lasers (Sapphire-488–200 and Sapphire-568–100, Coherent) and a continuous wave fiber laser (VFL-P-300–642-OEM1, MPB Communications) were independently attenuated with logarithmically spaced neutral density filter wheels, optically shuttered (VMM-D3, and LSS6T2, Uniblitz), and combined with dichroic mirrors (LM01-552-25 and LM01-613-25, Semrock). Much of this equipment was collected from a decommissioned OMX system, and future variants of the OPM microscope could include faster laser switching devices (e.g. solid-state lasers or acousto-optic devices). After the laser combining dichroic mirrors, the beams were focused through a 30-micron pinhole (P30D, ThorLabs) with a 50 mm achromatic doublet (AC254-050-A, ThorLabs) and recollimated thereafter with a 100 mm achromatic doublet (AC254-100-A, ThorLabs). Laser polarization was controlled with a half wave-plate (AHWP3, Bolder Vision Optik) that was secured in a rotation mount (RSP1 × 15, ThorLabs). The beam was then either reflected with a motorized flipper mirror (8892 K, Newport) toward the epi-illumination (for alignment) or laser spot (for optogenetics) path or transmitted toward the light-sheet illumination path.

For the light-sheet path (*Figure 1—figure supplement 4*), the light was first expanded in one dimension with a pair of cylindrical lenses (F = 25 mm, #68–160 Edmund Optics and ACY254-100-A, ThorLabs), and then focused into a 1D Gaussian profile using a cylindrical lens (ACY254-50-A, ThorLabs) onto a resonant galvo (CRS 12 kHz, Cambridge Technology) to reduce stripe artifacts by rapidly pivoting the light-sheet in sample space (*Huisken and Stainier, 2007*; *Figure 1—figure supplement 3*). Such a 1D light-sheet offers 100% spatial duty cycle, which reduces phototoxicity and photobleaching compared to light-sheets that are obtained by laterally scanning a 2D laser focus. An adjustable slit was placed at one focal distance in front of the cylindrical lens. This slit was used to adjust the effective NA of the light-sheet, which determines the light-sheet thickness and propagation length. For most experiments, the NA of the light-sheet was set to ~0.2, which creates a light-sheet with a Rayleigh length of about 21 microns. Opening the slit allows increasing the NA to 0.34, which is the practical limit for the chosen objective and inclination angle. The light was then relayed with a 100 mm achromatic doublet (AC254-100-A, ThorLabs) over a polarizing beam splitter (PBS251, ThorLabs), through a quarter wave-plate (AQWP3, Bolder Vision Optik) onto mirror galvanometer (6215H, Cambridge Technology), which backreflects the light through the same quarter wave-plate and polarizing beam splitter toward a multi-edge dichroic (Di03-R405/4888/561/635-t3−25 × 36, Semrock) that reflected the light toward the primary objective. The galvanometer mirror

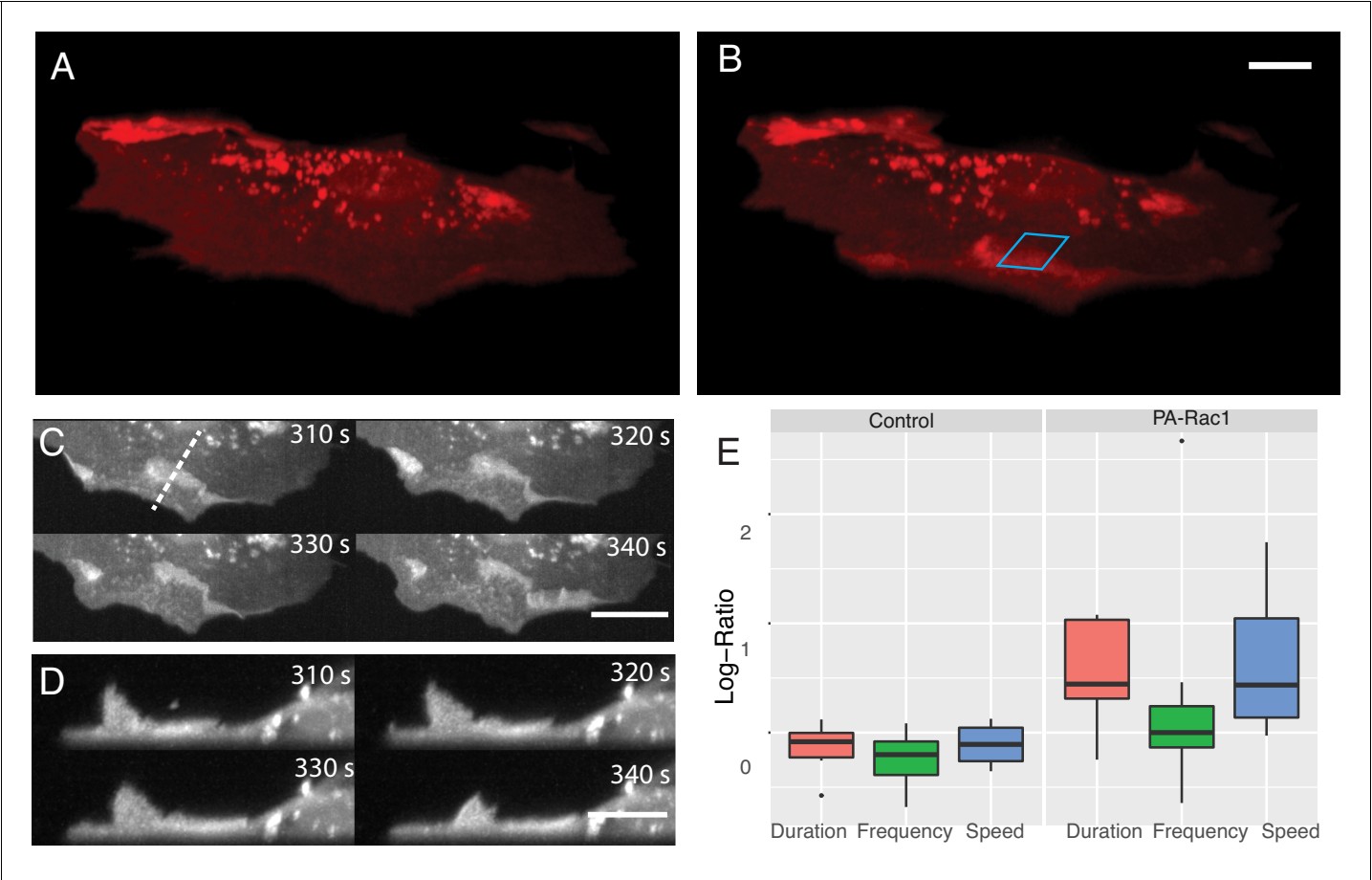

**Figure 7.** Simultaneous subcellular optogenetic stimulation of PA-Rac1 and volumetric imaging of morphodynamic changes in MEF cells. (**A**) Cell before optogenetic stimulation. (**B**) Localized optical stimulation of PA-Rac1 (within the blue box) was performed with a 488 nm laser operating in a laser-scanned illumination geometry synchronously with volumetric imaging using a 561 nm laser. Scale Bar: 10 microns. (**C**) Lateral maximum intensity projection of the cell during optical stimulation shows the dorsal ruffles moving from the cell periphery to the juxtanuclear cellular region. Scale Bar: 20 microns. (**D**) Orthogonal maximum intensity projection along the dotted line in (**C**) of dorsal ruffles. Scale Bar: 10 microns. (**E**) Hidden Markov model analysis gives the log-ratio difference between pre activation and activation response showing control cells (N = 6) with no difference in protrusion duration, speed, or frequency while cells expressing PA-Rac1 (N = 7) show statistically significant increases in protrusion speed (p=0.04) and duration (p=0.02) with no significant changes in frequency. All image data shown are raw.

allows control of the lateral positioning of the light-sheet. After the dichroic, the light was focused with a 200 mm tube lens (TTL200, ThorLabs), recollimated with a 39 mm scan lens (LSM03-VIS, Thor-Labs), reflected off of a 1D galvanometer mirror (6215H, Cambridge Technology), focused by a 70 mm scan lens (CSL-SL, ThorLabs), recollimated with a 200 mm tube lens (TTL200, ThorLabs), and imaged into the specimen with the primary objective (100X/1.35 MRD73950 Silicone Immersion Objective, Nikon Instruments). For detection (*Figure 1—figure supplement 7*), the fluorescence was descanned with the galvanometer mirror, transmitted through the multi-edge dichroic, and an image is formed by the secondary objective (CFI Plan Apo Lambda 40XC, Nikon Instruments), which is detected by the tertiary objective (AMS-AGY v1.0, Special Optics) and focused with a tube lens (ITL200, ThorLabs) onto a sCMOS camera (Flash 4.0 v3, Hamamatsu). Each imaging channel was collected sequentially (e.g. after a complete Z-stack), and the fluorescence was spectrally isolated with emission filters placed in a motorized filter wheel (FG-LB10-B and FG-LB10-NWE, Sutter Instruments). Detailed imaging parameters are listed in *Supplementary file 1* The scan galvanometer and resonant galvanometer were driven by a 28V (A28H1100M, Acopian) and a 12V power supply, respectively.

For widefield illumination (*Figure 1—figure supplement 5*), after being reflected by the flipper mirror, the light was focused onto a galvanometer mirror (6215H, Cambridge Technology) with an

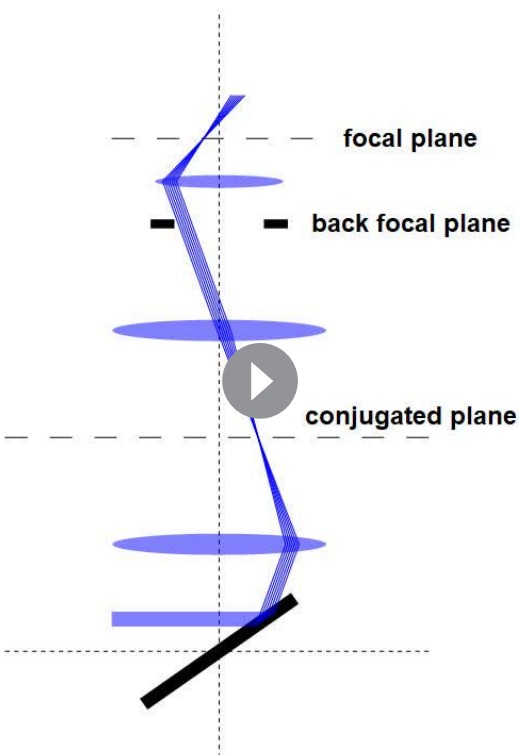

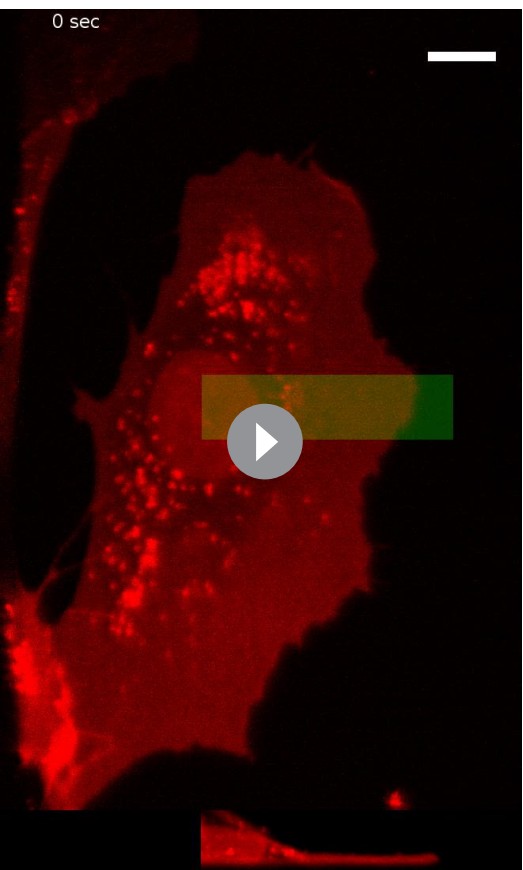

**Video 10.** Animation of light-sheet scanning and optogenetic activation. Blue rays represent the light-sheet illumination, and green rays indicate the near-diffraction-limited epi-illumination. Both beams are scanned with a mirror galvanometer (bottom) that is conjugate to the back focal plane of the primary objective (top).
https://elifesciences.org/articles/57681#video10

**Video 11.** Mouse embryonic fibroblasts expressing PA-Rac1 and mCherry. Movie shows 15 min in the absence of optical stimulation followed by 15 min of optical stimulation. The blue rectangle shows region undergoing optical stimulation with 488 nm light. An axial cross-section of the region in green is shown at the bottom. Scale Bar: 10 microns. Time Interval: 10 s.
https://elifesciences.org/articles/57681#video11

achromatic doublet (AC254-100-A, ThorLabs) that itself was mounted on a flipper mirror (8892 K, Newport). For laser spot illumination (*Figure 1—figure supplement 6*), the achromatic doublet was removed from the optical path, resulting in a collimated beam on the galvanometer mirror. By controlling this galvanometric mirror synchronously with the Z-galvanometer, the cell can be illuminated with arbitrary 2D patterns of light, which enables optogenetic stimulation, photoactivation, or fluorescence recovery after photobleaching. Thereafter, the epi-illumination and laser spot

illumination paths were reflected off the aforementioned polarizing beam splitter toward the multi-edge dichroic. The widefield illumination path proved useful for identifying interesting cells and for focusing the microscope, and the fluorescence was detected after the first tube lens with flipper-mounted dichroic mirror (Di02-R488-t3−25 × 36, Semrock) onto a sCMOS camera (Grasshopper 3, FLIR). The X, Y, and Z-resolution of the primary objective when imaged in a traditional widefield format was 243 ± 11 nm, 242 ± 15 nm, and 604 ± 31 nm, respectively.

**Video 12.** MEF PA-Rac1-mCherry cell movie same as *Video 11* show cell dynamics with Imaris 3D rendering. Time Interval: 10 s. Scale Bar: 10 microns.
https://elifesciences.org/articles/57681#video12

The data acquisition computer was a Colfax International ProEdge SXT9800 Workstation equipped with two Intel Xeon Silver 4112 processors operating at 2.6 GHz with 8 cores and 16 threads, 96 GB of 2.667 GHz DDR4 RAM, a Intel DC P3100 1024 GB M.2 NVMe drive, and a Micron 5200 ECO 7680 GB hard-drive for file storage. All software was developed using a 64-bit version of LabView 2016 equipped with the LabView Run-Time Engine, Vision Development Module, Vision Run-Time Module and all appropriate device drivers, including NI-RIO Drivers (National Instruments). Software communicated with the camera (Flash 4.0, Hamamatsu) via the DCAM-API for the Active Silicon Firebird frame-grabber and delivered a series of deterministic TTL triggers with a field programmable gate array (PCIe 7852R, National Instruments). These triggers included control of the optical shutters, galvanometer mirror scanning, camera fire and external trigger. The control software can be requested from the corresponding authors and will be distributed under an MTA with the University of Texas Southwestern Medical Center.

## Stage scanning microscope setup

The stage-scanning oblique plane microscope was built using an inverted geometry with a three-axis motorized stage with a constant scan speed optimized X stage (FTP-2000, Applied Scientific Instrumentation). Five solid-state continuous wave lasers (OBIS LX 405–100, OBIS LX 488–150, OBIS LS 561–150, OBIS LX 637–140, and OBIS LX 730–30, Coherent Inc) contained within a control box (Laser Box: OBIS, Coherent Inc) were combined with dichroic mirrors (zt405rdc-UF1, zt488rdc-UF1, zt561rdc-UF1, zt640rdc-UF1, Chroma Technology Corporation). After the laser combining dichroic mirrors, the beams were focused through a 30-micron pinhole (P30D, Thorlabs) with a 30 mm achromatic doublet (AC254-030-A, Thorlabs), recollimated with a 100 mm achromatic doublet (AC508-100-A, Thorlabs), and steered through an adjustable iris. The adjustable iris was used to control the diameter of the laser beam and the light-sheet NA at the sample. Light passed through an electro-tunable lens (EL10-30-C, Optotune) placed horizontally and relayed by two 100 mm achromatic doublets (AC254-100-A, Thorlabs) onto a 1-axis galvanometer mirror (GVS201, Thorlabs). The galvanometer mirror was placed in the back focal plane of a 300 mm achromatic doublet (AC508-300-A, Thorlabs). The line focus formed by pivoting the galvanometer mirror was formed on a 2-inch mirror that was used to control the light-sheet tilt at the sample plane. This mirror was placed in the back focal plane of a 180 mm achromatic doublet (AC508-180-A). This lens was placed such that the galvanometer mirror rotation was relayed to the back focal plane of the primary objective (100X/ 1.35 MRD73950 Silicone Immersion Objective, Nikon Instruments). Excitation light was reflected off a pentaband dichroic mirror (zt405/488/561/640/730rpc-uf3, Chroma Technology Corporation) and imaged into the specimen with the primary objective.

For detection, fluorescence was transmitted through the pentaband dichroic mirror, then transmitted through a 200 mm tube lens (MXA22018, Nikon Instruments), passed through an empty kinematic mirror cube (DFM1B, Thorlabs), a 357 mm tube lens assembly (AC508-500-A and AC508-750-A, Thorlabs), and an image is formed by the secondary objective (CFI Plan Apo Lambda 40XC, Nikon Instruments), which is detected by the tertiary objective (AMS-AGY v1.0, Special Optics) and focused with a tube lens (MXA22018, Nikon Instruments) onto a sCMOS camera (Prime BSI Express, Teledyne Photometrics). Each imaging channel was collected sequentially (e.g. after one complete strip scan of the stage), and laser light was blocked by two identical pentaband barrier filters (zet405/ 488/561/640/730 m, Chroma Technology Corporation), with one placed in infinity space before secondary objective and one in infinity space after the tertiary objective. The kinematic mirror cube after the primary tube lens was used to redirect light to either an inexpensive CMOS camera placed at the primary image plane (BFS-U3-200S6M-C, FLIR) or a wavefront sensor (HASO-VIS, Imagine Optic) to characterize the wavefront after the primary objective.

Acquisition was performed on a Windows 10 64-bit computer (Intel i7-9700K, 64 Gb memory, 12 TB SSD raid 0 array, Nvidia RTX 2060 GPU card) connected via 10 Gbps optical fiber to a network attached storage (DS3018XS, Synology), Data was acquired by scanning the scan optimized stage axis at a constant speed with the camera set to 'Trigger First' mode and triggered to start by the stage controller (Tiger, Applied Scientific Instrumentation) when the stage passed the user defined start point. The scan speed was adjusted so that the displacement between exposures was either 100 or 200 nm. This slow scan speed ensured that minimal motion artifacts occurred during stage scanning. The 'Exposure Out' trigger from the camera triggered one sweep of the galvanometer mirror across the FOV using a homebuilt data acquisition system (Teensy 3.5, PJRC and Power DAC

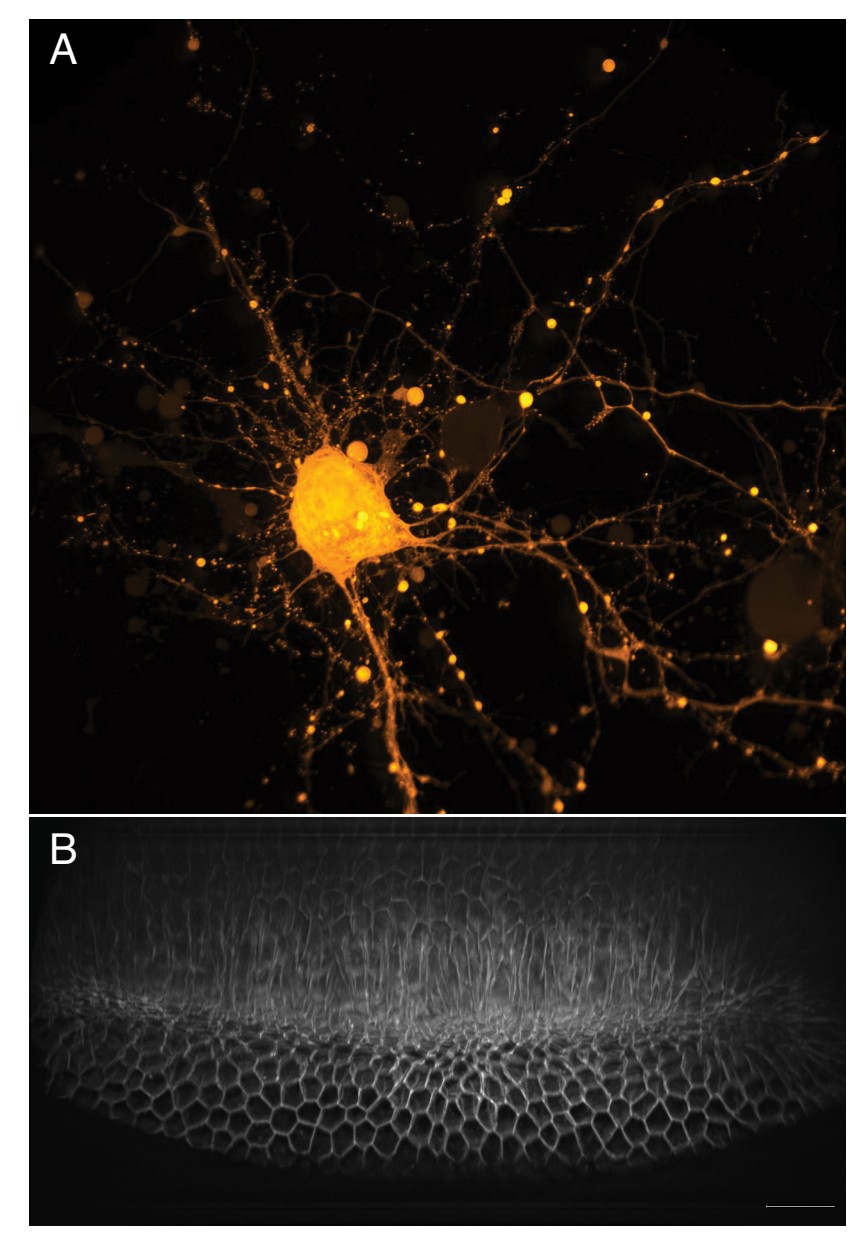

**Figure 8.** Large-scale imaging of cultured neurons and drosophila embryo gastrulation. (A) Cultured cortical neurons expressing the $Ca^{2+}$ biosensor GCaMP6f. Maximum intensity projection of deconvolved data. (B) Stage 6 *Drosophila* embryo expressing gap43-mCherry. Scale Bar: 20 microns. Maximum intensity projection of deconvolved data is shown in both (A) and (B).

The online version of this article includes the following figure supplement(s) for figure 8:

**Figure supplement 1.** Single XS camera view (single slice of a stack) acquired for *Drosophila* embryo before (A) and after (B) deconvolution.

module, Visgence, Inc). The camera chip was cropped to the area of interest containing the sample and data was saved as one TIFF file per image. Multiple laser lines are acquired sequentially by allowing the stage to reset to the original position and repeat the scan. A custom script in Micromanager 2.0 gamma sets the stage parameters and controls the scan. The stage-scanning post-processing control codes are available via the Quantitative Imaging and Inference Laboratory GitHub repository (http://www.github.com/QI2lab/OPM).

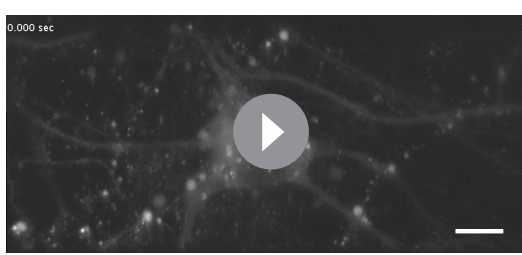

**Video 13.** Dissociated cortical neurons expressing the Ca²⁺ biosensor GCaMP6f, imaged volumetrically at 7 Hz. Scale Bar: 20 microns.
https://elifesciences.org/articles/57681#video13

## Transmission measurements

To evaluate the transmission efficiency, we measured the light throughput of the optical system with a 543 nm HeNe laser in transmission at both 0- and 30-degree tilts. Here, the diameter of the alignment laser was set to the size of the primary objective back pupil, with all of the optics and filters present in the optical path with the exception of the primary objective and the camera. When oriented at 30 degrees, the laser and stage-scanning variants had a 41% and 53% transmission, respectively, and at 0 degrees, we observed a 3% increase in transmission. 12% of the losses could be attributed to the scan lens, mirror galvanometer, tube lens combination, which can be eliminated with lens-free scanning (*Boden et al., 2020*). Nonetheless, the collection efficiency of both variants is greater than OPM designs that use beam splitters in their detection path (*Kim et al., 2019*).

## General alignment

In an effort to assist in the adoption of both the laser-scanning and stage-scanning OPM technologies described here, we provide a detailed discussion on how to align such systems in Appendix 4. Furthermore, a complete parts list for these microscopes, as well as other variants based on OPM, can be found in the work by Millett-Sikking and colleagues (*Millett-Sikking and York, 2019*).

## Data post-processing

For the laser-scanning system, analysis was performed on the local BioHPC high-performance computing cluster. Data was sheared with Python using a script originally developed by Dr. Bin Yang which applies the Fourier Shift Theorem. In instances where deconvolution was used, the raw data was deconvolved in a blind fashion where the experimentally measured PSF served as a prior. Of note, for the resolution measurements, we resampled the image data by a factor of 2 prior to deconvolution, reducing the lateral pixel size from 115 nm to 57.5 nm. As evidenced by the FRC measurements, this corresponds to zero-padding of the Fourier transform of the data, which allows the iterative deconvolution algorithm to reconstruct slightly out-of-band information (*Heintzmann, 2007*). For most of the biological data, this resampling was not performed, as the data size would have become limiting. Leaving the data sampled laterally at 115 nm pixel size results in a highest possible Nyquist limited resolution after deconvolution of 230 nm.

Both of these data post-processing functions are available via the AdvancedImagingUTSW GitHub repository (https://github.com/AdvancedImagingUTSW). As an example, shearing and deconvolution of a 2048 × 256×450 voxel image took ~5 and~120 s, respectively. Nonetheless, we believe that with GPU computing, real-time processing may be a possibility. For rotation into the traditional epi-fluorescence-like orientation, the freely available IMOD software package was used (*Kremer et al., 1996*). For the stage-scanning system, analysis was performed on an Ubuntu 20.04 LTS computer (2x Intel E5-2600, 128 Gb memory, 16 TB SSD RAID 0 array, Nvidia Titan RTX card) connected via 10 Gbps optical fiber to the same network attached storage as the acquisition computer. First, a retrospective flat-field was calculated per channel (*Peng et al., 2017*). Using Numpy and Numba libraries in

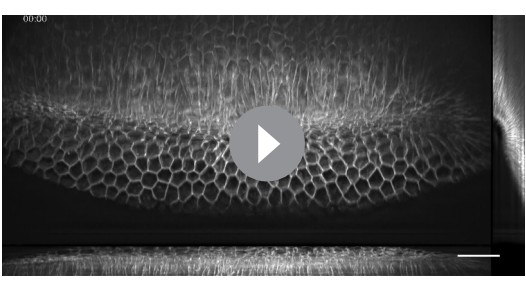

**Video 14.** Stage 6 *Drosophila* embryo undergoing gastrulation. Ventral furrow ingression occurs along the anteroposterior axis of the embryo and is immediately followed by rapid epithelial mitotic events. Time Interval: 23 s.
https://elifesciences.org/articles/57681#video14

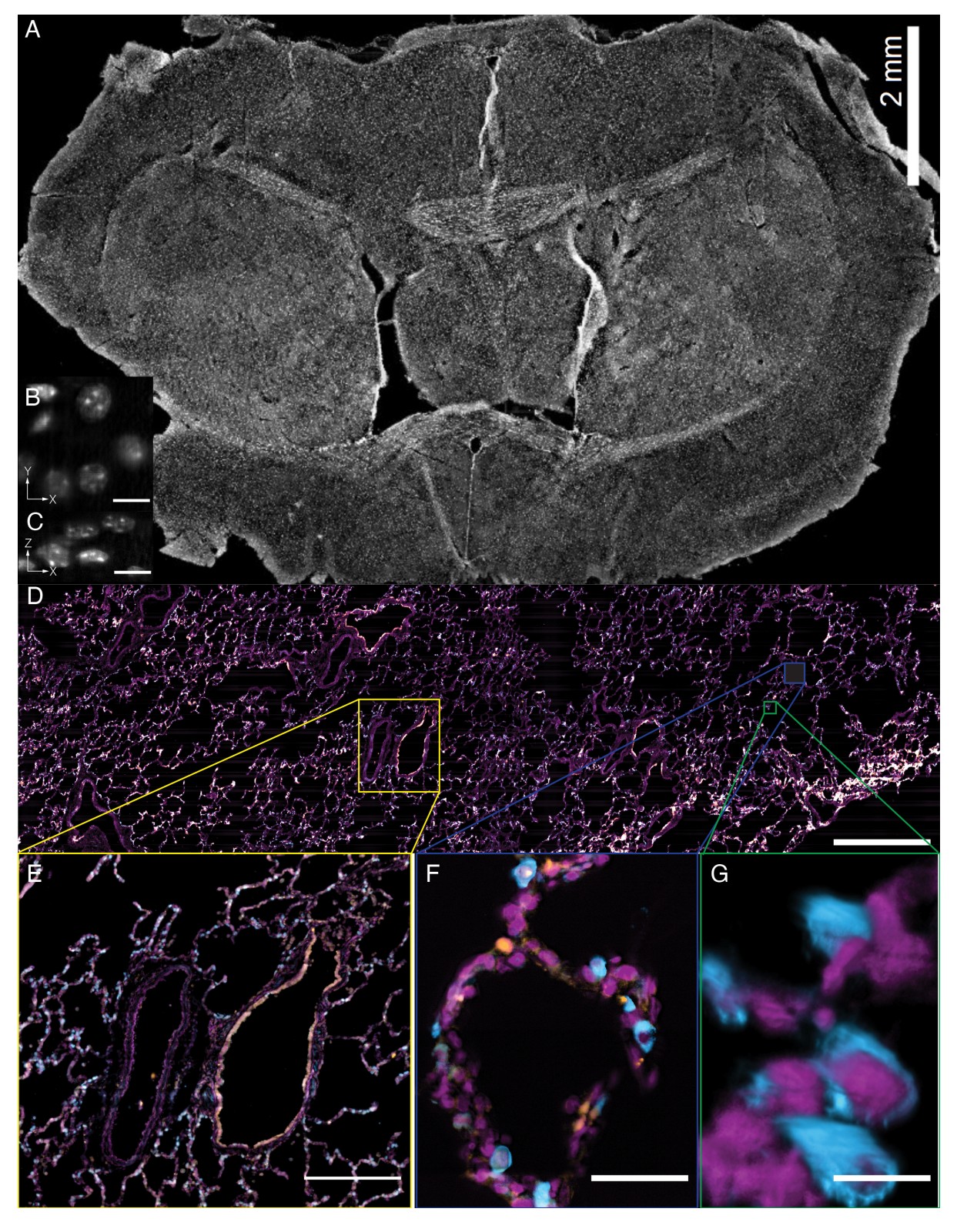

**Figure 9.** Tissue-scale imaging. (**A**) Maximum intensity projection of fused raw image for 30 um thick mouse brain tissue labeled for nuclei Scale Bar: 2 mm. (**B**) Individual raw XY slice of individual nuclei. Scale Bar: 250 microns. (**C**) Individual raw YZ slice of data in (**B**). Scale Bar: 250 microns. (**D**) Maximum projection of fused raw image for 15 um thick human lung tissue labeled for nuclei (magenta), SFTPC protein (cyan), *ACE2* mRNA (orange). Scale Bar: 2

*Figure 9 continued on next page*

*Figure 9 continued*

mm. (**E**) Maximum projection of raw data for yellow box in (**A**). Scale Bar: 0.5 mm. (**F**) Maximum projection of raw data for blue box in (**A**). Scale Bar: 75 microns. (**G**) 3D rendering of raw data for green box in (**A**). All data was deconvolved. Scale Bar: 25 microns.

The online version of this article includes the following figure supplement(s) for figure 9:

**Figure supplement 1.** Geometric considerations for the maximum sample size of lattice light-sheet microscopy.

**Figure supplement 2.** Photobleaching comparison for lattice light-sheet microscopy and OPM.

Python (*van der Walt et al., 2011*), data was split into tiles that fit in local memory, flat-fielded, orthogonally interpolated to deskew the stage scan (*Maioli, 2017*), saved as a BigDataViewer H5 file (*Pietzsch et al., 2015*), and stitched using default settings in BigStitcher (*Hörl et al., 2019*). The stage-scanning post-processing codes are available via the Quantitative Imaging and Inference Laboratory GitHub repository (http://www.github.com/QI2lab/OPM).

## Analysis of optogenetic responses

To evaluate the response of MEFs to blue light activation in PA-Rac1 and control cells we used an algorithm previously described to assess cell morphodynamics (*Welf et al., 2019*). Briefly, maximum intensity projection images of cells (mCherry) were pre-processed in Fiji via simple ratio bleach correction and a median filter with a 230 nm pixel size. The cell edge was then automatically detected, and small regions of interest (windows) of size 920 nm x 920 nm were placed around the periphery of the cell to assess cell edge velocity over time. The velocity profile was then analyzed with a Hidden Markov Model defining different protrusive states (*Welf et al., 2019*), which were then used to assess protrusion speed, frequency, and duration. The Hidden Markov Model analysis was done using R package 'depmixS4' (*Visser and Speekenbrink, 2010*). Here, we quantified the measures as a function of cell type (control vs. optogenetics cells). Log-ratio was used to quantify change via the difference in protrusion parameters before and during activation (e.g. $\log(\frac{Post-Activation}{Pre-Activation})$). The protrusion parameters were then assessed for difference between pre-activation and activation using a t-test for MEF control (N = 6) and PA-Rac1 (N = 7).

## Cell lines, Plasmids and Transfection

NK-92 cells were obtained from ATCC (CRL-2407) and maintained in alpha minimum modified Eagle medium, 0.2 mM myoinositol, 0.1 mM beta-mercaptoethanol, 0.02 mM folic acid, 12.5% heat-inactivated horse serum, 12.5% heat-inactivated FBS (Sigma–Aldrich), 2 mM L-glutamine and non-essential amino acids (ThermoFisher Scientific), supplemented with 100 U/mL Il-2 (Roche). K562 cells were obtained from ATCC (CCL-243) and cultivated in RPMI medium with high glucose, supplemented with 10% of heat-inactivated FBS (Sigma–Aldrich), 2 mM L-glutamine and non-essential amino acids (ThermoFisher Scientific). Both NK-92 and K562 cells were maintained in 37°C, 5% $CO_2$ tissue culture incubators and routinely confirmed to be mycoplasma negative using LookOut mycoplasma PCR detection kit (Sigma–Aldrich). pLifeAct-mScarlet-N1 and pLck-mVenus-C1 were gifts from Dorus Gadella (Addgene plasmids #85054 and #84337) and transfected by nucleofection using Amaxa Kit R per manufacturer's instructions (Lonza). Positive cells were amplified under antibiotic selection pressure and sorted for low or intermediate expression level of the fluorescently tagged protein on an Aria II Fluorescence Activated Cell

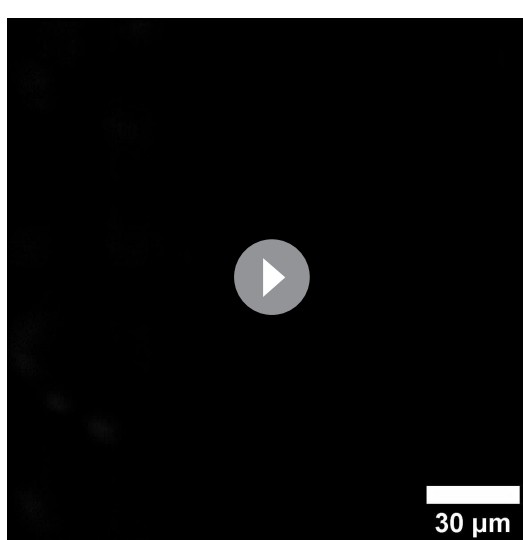

**Video 15.** Z-stack of raw data for 30-micron thick slice of coronal mouse brain tissue. Sub-nuclear features such as nucleoli, are readily evident owing to the high-resolution and optical sectioning of the OPM.

https://elifesciences.org/articles/57681#video15

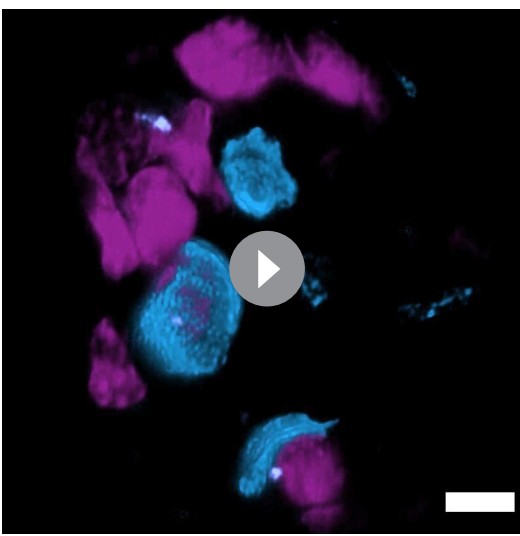

**Video 16.** Volumetric rendering of deconvolved data for human lung tissue as imaged with the stage-scanning variant of the OPM, which permits imaging of cm-scale objects. Nuclei (magenta) and SFTPC protein (cyan). SFTPC is a commonly used cytosolic marker of alveolar type 2 cells, which is resolved with subcellular resolution in 3D. Scale Bar: 10 microns.

https://elifesciences.org/articles/57681#video16

Sorter (BD). Each sorted population was then used for pilot experiments to determine the lowest possible expression level required for optimal imaging conditions. NK-92 cells expressing Life-Act-mScarlet and K562 cells expressing Lck-mVenus were mixed 1:1 and briefly spun down and resuspended in full prewarmed medium. The cell conjugates were then seeded in #1.5 uncoated glass bottom dishes and imaged as soon as possible for up to 45 min post mixing. All imaging was performed at 37°C using prewarmed media and a stage top insert enclosing the sample.

To stably express EGFP-Sec61b in U2OS cells, a TET-inducible EGFP-Sec61b fusion was knocked into the AAVS1 safe harbor locus (*Qian et al., 2014*). The homologous recombination donor (HRD) was generated by EGFP tag and Sec61b cDNA (a gift from Dr. Jennifer Lippincott-Schwartz, Addgene #90992) into AAVS1-TRE3G-EGFP (a gift from Su-Chun Zhang, Addgene #52343), which was linearized with MluI and SalI and used for Gibson assembly. U2OS cells were cultured in RPMI (Gibco, A4192301) media supplemented with 10% FBS and Pen/Strep at 37°C, 5% $CO_2$. For AAVS1 locus knock-in, 1 µg of HRD plasmid and 1 µg of AAVS1 T2 CRISPR plasmid (a gift from Masato Kanemaki, Addgene plasmid #72833) were transfected into U2OS cells using FuGENE HD (Promega) according to the manufacturer's instructions (*Natsume et al., 2016*). 48 hr after transfection selection was initiated with 1 µg/mL of puromycin. The FBS used was not tetracycline free, which resulted in sufficient EGFP-Sec61b expression without the addition of doxycycline.

The human melanoma cell line1205Lu (a gift from Dr. Meenhard Herlyn, Wistar Institute) was genotypically characterized as previously reported (*Smalley et al., 2007a*; *Smalley et al., 2007b*) and grown in high glucose DMEM (11965092, Gibco, Thermo Fisher Scientific) supplemented with 10% fetal bovine serum (10100147, Gibco, Thermo Fisher Scientific), 1x MEM Non-Essential Amino Acids (11140050, Gibco, Thermo Fisher Scientific) and 50 U/mL penicillin/50 µg/mL streptomycin (15070063, Gibco, ThermoFisher Scientific). pLenti CMV Hygro 3NLS-mScarlet-I was generated by Gateway Gene Cloning (Invitrogen, Thermo Fisher Scientific). First, 3XNLS-mScarlet-I (a gift from Dr. Dorus Gadella. Addgene #98816) was amplified by PCR (M0515, Q5 Hot start, New England Biolabs) to introduce 5' EcoRI and 3' XbaI restriction enzyme sites flanking either ends of 3XNLS mScarlet-I sequence (*Bindels et al., 2017*; *Chertkova et al., 2020*). The entry vector pENTR1A-GFP-N2 (FR1) (a gift from Dr. Eric Campeau and Dr. Paul Kaufman, Addgene #19364) along with the purified PCR fragment was digested with EcoRI and XbaI then ligated together with T4 DNA ligase (M0202, New England Biolabs). pENTR1a-3XNLS-mScarlet-I was then recombined using Gateway LR Clonase II (11791, Invitrogen, Thermo Fisher Scientific) as per the manufacturer's instructions into pLenti CMV Hygro DEST (W117-1, a gift from Dr. Eric Campeau and Dr. Paul Kaufman, Addgene #17454) to create the final vector which was sequence verified by Sanger sequencing (Australian Genome Research Facility). CRISPR-CAS9 Endogenous EGFP tagging of the TUB1AB genetic locus in 1205Lu cells was performed as previously described (*Khan et al., 2017*) with the following modifications. Cells were transfected with 3.5 µg of each CAS-9-guide and EGFP donor knock-in vector using Lipofectamine 2000 (11668019, Invitrogen, Thermo Fisher Scientific) for 6 hr before replacing media to allow cells to recover. To achieve a uniformly fluorescent population of cells, cells were sorted using a MoFlo Astrios EQ cell sorter (Beckman Coulter) to isolate cells with dual-positive expression profile of EGFP and mScarlet-I using a gating strategy isolating the top ~20% of GFP and an intermediate ~30–60% population of mScarlet-I. To label the nucleus, 1205Lu cells were lentivirally

transduced to stably overexpress 3XNLS-mScarlet-I as previously published (*Coleman et al., 2003*). After successful transduction, cells were grown in media containing 0.1 mg/mL Hygromycin (H3274, Roche).

MV3 cells were obtained from Peter Friedl (MD Anderson Cancer Center, Houston TX), and cultured in DMEM (Gibco) supplemented with 10% FBS (ThermoFisher) in 37°C and 5% $CO_2$. MV3 expressing genetically encoded multimeric nanoparticles (GEMs) were made by infection with lentiviral construct from Addgene (Plasmid #116934) (*Delarue et al., 2018*). Cells were FACS sorted to purify a population of cells expressing T-Sapphire GEMs. The membrane marker expressed in MV3 cells was created from the membrane targeting (i.e. CAAX) domain of KRas fused to the HALO tag and cloned into the pLVX vector. Cells were infected with virus containing this construct and selected for expression by G418 resistance.

Mouse embryonic fibroblast cells (MEFs) were obtained from ATCC (MEF CF-1, SCRC-1040) and cultured in DMEM (Gibco) supplemented with 10% fetal bovine serum (ThermoFisher) at 37°C and 5% $CO_2$. MEFs we infected with retrovirus encoding PA-Rac1 (pBabe-TetCMV-puro-mCherry-PA-Rac1, Addgene #22035) and selected for 1 week with 10 ng/mL of puromycin (10 ng/mL). Likewise, control cells were infected with lentivirus encoding cytosolic mCherry in a pLVX-Puro vector backbone (Clontech) (*Wu et al., 2009*). Imaging and photoactivation of MEFs was performed with cells plated on 35 mm dish with cover glass bottom (P35G-1.5–14 C, Mattek). For both control and PA-Rac1 cells imaging of mCherry with a 561 nm laser, and PA-Rac1 was stimulated with a 488 nm in a point scanning rectangular geometry and a laser power or 30 µW before the primary objective.

ARPE and RPE hTERT cells were generated and cultured as previously described (*Aguet et al., 2013*; *Gan et al., 2016*). The gap43-mCherry fly line used for imaging ventral furrow ingression is as previously described. (*Martin et al., 2010*). After dechornination and washing, the embryos were placed on the surface of a glass bottom Mattek dish and submerged in a droplet of water to prevent the embryos from drying out. Cardiomyocytes were isolated from the left ventricle of 1- to 2-day-old Sprague-Dawley rats. The isolation process and initial culture were described previously (*Morales et al., 2016*). Cells seeded on a 35 mm plate were stained with Fluo-3 AM calcium indicator (Thermofisher) at 1 µM for 20 min incubation after which seeded cells were image on the microscope using 488 nm excitation.

Rat primary neurons were obtained in accordance with protocols approved by the University of Texas Southwestern Medical Center Institutional Animal Care and Use Committee (IACUC). Cell Culture and Labeling Male and female embryonic day 18 (E18) primary cortical neurons were prepared from timed pregnant Sprague-Dawley rats (Charles River Laboratories, Wilmington, MA) as previously described (*Bock and Herz, 2003*). Embryonic cortices were harvested, neurons dissociated, and plated on 35 mm glass bottom culture dishes (MatTek P35G-1.5–10 C), coated with poly-D-lysine, at a density of 0.8 million neurons per dish. Neurons were cultured in completed Neurobasal medium (Gibco 21103049) supplemented with 2% B27 (Gibco 17504044), 1 mM glutamine (Gibco 25030081), and penicillin streptomycin (Gibco 15140148) at 37°C in a 5% CO2 environment. At day in vitro (DIV) two neurons were treated with AraC to prevent overgrowth of glia cells. Half of the culture media was renewed twice a week. On DIV5 neurons were infected with lentivirus encoding GCaMP6f. The lentivirus was generated by co-transfecting HEK 293 T cells with psPAX2, pMD2G (kindly provided by Didier Trono, Addgene numbers 12260, 12259), and pLV-GCaMP6f. Neurons were utilized for live-imaging at DIV12-16.

Mouse brain tissue was procured, cleared with ice-cold, nuclease-free 1x PBS solution and fixed with ice-cold 4%, nuclease-free PFA solution via transcardial perfusion. Following fixation, brain tissue was dissected and post-fixed for overnight at 4°C. Tissue was then cryo-preserved in 15% then 30% nuclease-free sucrose solution before freezing in OCT and cut into 40 micron coronal sections via vibratome. Sections were stored in nuclease-free, 1x PBS solution at 4°C prior to mounting and staining. To improve the optical clarity of the tissue samples, we removed light scattering proteins and lipids. To achieve this, the samples were first embedded to a polyacrylamide gel matrix and then enzymatically digested to remove proteins and chemically dissolved to remove lipids as previously described (*Moffitt et al., 2016*). In short, samples were washed for two minutes with degassed polyacrylamide (PA) solution consisting of 4% (vol/vol) 19:1 acrylamide/bis-acrylamide (161010144, BioRad), 60 mM Tris-HCl pH 8 (AM9856, ThermoFisher), and 0.3 M NaCl (AM9759, Thermofisher) and then washed for two minutes with PA gel solution which consists of PA solution with the addition of polymerizing agents TEMED (Sigma, T9281) and ammonium persulfate (A3678, Sigma). The

PA gel solution was then aspirated from the coverslip-mounted sample. To cast the gel film, 200 μL PA gel solution was applied to the surface of a Gel Slick (50640, Lonza) coated glass plate and the coverslip-mounted sample was inverted onto the PA Gel solution, creating a thin layer of gel solution between the two panes of glass. The PA gel was allowed to cast for 1.5 hr. Coverslip and affixed gel film were carefully removed from the glass plate. Following the gel embedding, samples were washed twice for 5 min with digestion buffer consisting of nuclease-free water with 0.8 M guanidine-HCl (G3272, Sigma), 50 mM Tris-HCl pH 8, 1 mM EDTA and 0.5% (vol/vol) Triton X-100 (T8787, Sigma). Once complete, samples were incubated in digestion enzyme buffer consisting of digestion buffer supplemented with (0.5%) proteinase K (P8107S, New England Biolabs) and 5% Pronase (11459643001, Sigma), at a concentration of 80 U/mL, at 37°C for 24 hr to clear the tissue. Once cleared, tissue was washed in 2x SSC buffer three times for 5 min. After clearing, tissue was stained with DAPI at a concentration of 50 μg/mL in 2x SSC buffer overnight at 37°C. Sample was then washed in 2x SSC buffer two times for 5 min. The sample was finally mounted in a flow chamber (no-heat FCS2, Bioptechs) and immersed in SlowFade Diamond Antifade Mountant (S36967, ThermoFisher).

Excised sub transplant-quality human lung tissues from donors without preexisting chronic lung diseases were obtained from the Marsico Lung Institute at the University of North Carolina at Chapel Hill under the University of North Carolina Biomedical Institutional Review Board-approved protocols (#03–1396). Human lungs were fixed with 4% PFA solution for overnight at 4°C. The fixed lungs were washed with nuclease-free 1x PBS, incubated with 30% Sucrose in PBS for overnight at 4°C. Sucrose was replaced by 1:1 solution of 30% Sucrose:OCT and tissue was incubated for 1 hr at 4°C before freezing in OCT and cut into 10 to 15 micron sections using cryostat. Proximity ligation in situ hybridization (PLISH) was performed as described previously (*Nagendran et al., 2018*). Briefly, fixed-frozen human lung sections were re-fixed with 4.0% formaldehyde for 20 min, treated with 20 μg/mL proteinase K (Thermo Scientific, EO0492) in 1x PBS for 9 mins at 37°C, and dehydrated with up-series of ethanol. The sections were incubated with gene-specific probes (*Supplementary file 2*) in hybridization buffer (1 M sodium trichloroacetate, 50 mM Tris (pH 7.4), 5 mM EDTA, 0.2 mg/mL heparin) for 2 hr at 37°C followed by 4 washes each 5 min with hybridization buffer. Common bridge and circle probes were added to the tissue section and incubated for 1 hr at 37°C followed by a wash with 1x PBS containing 0.05% Tween-20 (PBS-T) for 5 min. Sections were incubated with T4 DNA ligase (New England Biolab, M0202T) for 2 hr at 37°C followed by 2 washes each 5 min with hybridization buffer and a wash with 1X phi29 DNA polymerase buffer (Thermo Scientific, B62). Rolling circle amplification was performed by using phi29 polymerase (Lucigen, 30221) for 12 hr at 37°C followed by 2 washes each 5 min with label probe hybridization buffer (2x SSC, 20% Formamide). Tissues were incubated with fluorophore-conjugated detection probes in label probe hybridization buffer for 30 min at 37°C followed by three washes each 5 min with label probe hybridization buffer. Immunostaining was performed using standard protocols. Briefly, tissue sections were incubated with primary antibodies against pro-SFTPC (Millipore, ab3786, 1:200 dilution) in 1x PBS-T containing 1% BSA and 5 mM EDTA for 1 hr followed by three washes each 5 min with PBS-T. Sections were incubated with Donkey anti-rabbit IgG secondary antibody (Thermo Scientific, A31573, 1:400 dilution) for 45 min followed by three washes each 5 min with PBS-T. Sections were mounted in medium containing DAPI (Thermo Scientific, 00-4959-52).

The source, authentication method, and mycoplasma state for all cell lines are provided in *Supplementary file 3*.

## Microfluidic printing, fabrication, and casting

Masks were designed using Tanner L-edit IC Layout (Mentor, Siemens Business) and printed using a Heidelberg μPG 101 mask writer (Heidelberg instruments) to create a 5-inch square chrome photomask on a soda lime glass substrate. After photomask printing, photomasks were developed in AZ 726 MIF developer (MicroChemicals), etched in chromium etchant (651826, Sigma–Aldrich) and cleaned in an ultrasonic acetone bath (V800023, Sigma–Aldrich). To create master molds, 4-inch round silicon wafers were surface etched with hydrofluoric acid (339261, Sigma-Aldrich) to remove the surface silicone oxide layer in order to ensure photoresist adhesion to the underlying silicon wafer. Master mold of approximately 5-micron height were created using manufacturers recommended spin coating and post-spin coating processing guidelines for SU-8 2005 negative photoresist (MicoChemicals). SU-8 2005 spin coated silicon wafers were contact printed using the developed

photomasks on an EVG620 mask aligner (Ev Group) with 360 nm light exposure at a dose of 120 mJ/cm$^2$ and post-baked according to manufacturer recommended guidelines (MicroChemicals). Prior to the first casting, each master mold was vapor deposition silanized with Trichloro (1H, 1H, 2H, 2H-perfluorooctyl) silane (448931, Sigma-Aldrich) in a vacuum desiccator (Scienceware, Sigma-Aldrich) to create an anti-stiction layer for PDMS demolding as previously described (*Qin et al., 2010*). To create micro channels, Polydimethylsiloxane elastomer (PDMS) castings of the master mold were made using SYLGARD 184 silicone elastomer (Dow Corning) and prepared and cured according to manufacturer's recommended instructions. PDMS microchannels were cut to size using a razor blade before cell seeding ports were created using a 3.0 mm diameter biopsy punch (ProSci-Tech) at opposite ends of a 1500-micron length microchannel array. PDMS microchannel were bonded to glass bottom dishes (P35G-1.0–14 C, Mattek) using a corona tool (Electro-Technic Products Inc) as previously published (*Haubert et al., 2006*), sterilized with 100% ethanol before being incubated overnight with 20 µg/mL bovine telo-collagen I in PBS (5225, Advanced Biomatrix). On the day of assay, each dish was replaced with fully complemented media before seeding cells into each of the seeding ports.

## Silanization of coverslips for tissue slices

To promote covalent adhesion of tissue and polyacrylamide gel to glass coverslips, coverslips were silanized according to previously published methods (*Moffitt et al., 2016*). In short, 40 mm, #1.5 coverslips (0420-0323-2, Bioptechs) were washed at room temperature in solution consisting of 1:1 (vol/vol) 37% HCl and Methanol for 30 min, rinsed in DI water three times and then dried at 60–70° C. Following this, coverslips were washed in solution consisting of chloroform with 0.1% (vol/vol) triethylamine (TX1200, Millipore) and 0.2% (vol/vol) allyltrichlorosilane (107778, Sigma) for 30 min at room temperature. Finally, coverslips were washed once in chloroform and once in ethanol and then baked for 1 hr at 60–70°C. Coverslips were then stored in a vacuum desiccation chamber before use.

## Image based resolution metrics

Fourier ring correlation (FRC) was performed by first blind denoising individual raw images to remove correlated noise due to the sCMOS readout architecture (*Banterle et al., 2013*; *Broaddus et al., 2020*; *Hörl et al., 2019*; *Krull et al., 2019*; *Van den Eynde et al., 2019*; *van Heel and Schatz, 2005*). Raw images were then deconvolved, deskewed, and rotated into the coverslip reference frame as described above. The axial size of resulting raw and deconvolved imaging volumes was reduced to match the axial size of the cells within the region of interest. Because the transformed volumes were oversampled in the axial direction, even and odd images could be split into separate image stacks. FRC curves and resolution estimates were calculated using the 1/7 resolution criteria for all paired images. The mean and 95% confidence intervals were calculated across all FRC curves. Image decorrelation analysis was performed using provided FIJI plugin and default settings to the same images used for FRC (*Descloux et al., 2019*). The mean and 95% confidence intervals were calculated using all resolution estimates.

## Bleaching comparison between OPM and lattice light-sheet microscopymicroscopy

The Lattice Light-Sheet Microscope used for photobleaching experiments has been described in detail elsewhere (*Chang et al., 2019*). In an effort to facilitate comparison with the original Lattice manuscript (*Chen et al., 2014*), we modified our system such that it used the 28.6X NA 0.66 (54-10-7, Special Optics) and 25X NA 1.1 (CFI75 Apo 25XC W, Nikon Instruments) objectives for illumination and detection, respectively. A 500 mm tube lens (AC508-250-A, Thorlabs) is used to achieve a final magnification of 62.5X. We chose an annular mask with an outer and inner diameter of 3.76 and 2.98 mm, which corresponds to an outer and inner NA of 0.269 and 0.213, respectively. The selected annular mask and the corresponding pattern on the spatial light modulator results in a square lattice light-sheet with an effective NA of 0.16 (*Chang et al., 2020*), which matches the illumination NA of the OPM. The resulting lattice was rapidly dithered to create a time-averaged sheet of light during the image acquisition. Photobleaching experiments were performed on ARPE cells expressing EB3-mNeonGreen seeded on #1.5 polymer coverslip imaging dishes (µ-Dish 35 mm, ibidi) or 5 mm diameter coverslips for the OPM and Lattice Light-Sheet Microscopes, respectively. In both setups we

used similar imaging conditions that included an excitation wavelength of 488 nm, a laser power of 470 µW at the back pupil of the illumination objective, a 20 ms camera integration time, 201 image planes acquired with a 0.5 µm axial step size (before shearing), for 50 time points. EB3 comets were automatically detected using u-Track (*Jaqaman et al., 2008*), and their intensity plotted through time with MATLAB.

## Acknowledgements

The authors would like to thank Dr. Dana Reed for her generous support, as well as the lab of Professor Joseph Hill for providing primary cardiomyocytes for calcium imaging, and Dr. Rosa E Mino, Dr. Madhura Bhave, and Dr. Marcel Mettlen for providing the ARPE Cell line tagged with AP2-GFP. We thank Tamara Terrones for her technical assistance in preparing primary neurons. Microfluidics were prepared in part at the Queensland node of the Australian National Fabrication Facility, a company established under the National Collaborative Research Infrastructure Strategy to provide nano- and microfabrication facilities for Australia's researchers.

## Additional information

### Competing interests

Ariella B Hanker: Receives research grant support from Takeda. Carlos L Arteaga: Serves in an advisory role for Novartis, which has an investment interest in alpelisib; receives or has received research grants from Puma Biotechnology, Pfizer, Lilly, Bayer, Takeda, and Radius; holds stock options in Provista and Y-TRAP; serves or has served in an advisory role to Novartis, Immunomedics, Merck, Lilly, Symphogen, Daiichi Sankyo, Radius, Taiho Oncology, H3Biomedicine, OrigiMed, Puma Biotechnology, and Sanofi; and reports scientific advisory board renumeration from the Komen Foundation. Alfred Millett-Sikking, Andrew G York: Employee of Calico Life Sciences LLC. Kevin M Dean, Reto P Fiolka: Has an investment interest in Discovery Imaging Systems, LLC. The other authors declare that no competing interests exist.

### Funding

| Funder | Grant reference number | Author |
| --- | --- | --- |
| Cancer Prevention and Research Institute of Texas | RR160057 | Reto P Fiolka |
| National Institutes of Health | R00 GM120386 | Jens C Schmidt |
| National Institutes of Health | R01HL068702 | Douglas P Shepherd Rory Kruithoff |
| National Institutes of Health | R33CA235254 | Reto P Fiolka |
| National Institutes of Health | R35GM133522 | Reto P Fiolka |
| National Institutes of Health | K25 CA204526 | Erik S Welf |
| National Institutes of Health | P30 CA142543 | Carlos L Arteaga |
| National Institutes of Health | 1R01MH120131-01A1 | Kevin M Dean |
| National Institutes of Health | 1R34NS121873 | Kevin M Dean |
| National Institutes of Health | 5P30CA142543 | Kevin M Dean |
| Damon Runyon Cancer Research Foundation | DFS-24-17 | Jens C Schmidt |
| Chan Zuckerberg Initiative | HCA3-0000000196 | Yoshihiko Kobayashi Purushothama Rao Tata Douglas P Shepherd |
| Australian Research Council | FT190100516 | Samantha J Stehbens |
| Rebecca L. Cooper Medical Research Foundation | PG2018168 | Samantha J Stehbens |

| University of Queensland | RM2018002613 | Samantha J Stehbens |
| Company of Biologists | JCSTF1903138 | Robert J Ju |
| Robert A. Welch Foundation | I-1950-20180324 | Konstantin Dubrovinski |
| National Institutes of Health | R01GM110066 | Konstantin Dubrovinski |
| Human Frontier Science Program | LT000911/2018C | Jaewon Huh |

The funders had no role in study design, data collection and interpretation, or the decision to submit the work for publication.

## Author contributions

Etai Sapoznik, Resources, Validation, Investigation, Visualization, Writing - original draft; Bo-Jui Chang, Data curation, Formal analysis, Visualization; Jaewon Huh, Formal analysis; Robert J Ju, Resources, Formal analysis, Investigation; Evgenia V Azarova, Theresa Pohlkamp, David Broadbent, Yoshihiko Kobayashi, Purushothama Rao Tata, Konstantin Doubrovinski, Resources; Erik S Welf, Resources, Data curation, Validation, Visualization; Alexandre F Carisey, Resources, Validation, Visualization, Writing - original draft; Samantha J Stehbens, Resources, Validation, Methodology, Writing - original draft; Kyung-Min Lee, Resources, Validation; Arnaldo Marín, Ariella B Hanker, Jens C Schmidt, Resources, Writing - original draft; Carlos L Arteaga, Resources, Supervision, Writing - original draft, Writing - review and editing; Bin Yang, Andrew G York, Supervision, Writing - original draft; Rory Kruithoff, Resources, Visualization, Writing - original draft; Douglas P Shepherd, Resources, Data curation, Software, Supervision, Validation, Visualization, Funding acquisition, Writing - original draft; Alfred Millett-Sikking, Supervision, Validation, Writing - original draft; Kevin M Dean, Conceptualization, Resources, Data curation, Software, Supervision, Validation, Investigation, Writing - original draft; Reto P Fiolka, Conceptualization, Data curation, Software, Formal analysis, Supervision, Funding acquisition, Validation, Investigation, Visualization, Methodology, Writing - original draft, Project administration

## Author ORCIDs

Etai Sapoznik https://orcid.org/0000-0001-8472-0299
Bo-Jui Chang https://orcid.org/0000-0002-5513-7106
Jaewon Huh https://orcid.org/0000-0003-3954-8092
Robert J Ju https://orcid.org/0000-0002-9850-9803
Evgenia V Azarova https://orcid.org/0000-0002-3846-9176
Theresa Pohlkamp http://orcid.org/0000-0003-3923-1917
David Broadbent https://orcid.org/0000-0002-0940-1068
Alexandre F Carisey http://orcid.org/0000-0003-1326-2205
Samantha J Stehbens https://orcid.org/0000-0002-8145-2708
Ariella B Hanker https://orcid.org/0000-0002-8655-8341
Jens C Schmidt http://orcid.org/0000-0001-9061-7853
Yoshihiko Kobayashi http://orcid.org/0000-0001-7031-1478
Purushothama Rao Tata http://orcid.org/0000-0003-4837-0337
Douglas P Shepherd https://orcid.org/0000-0001-9087-0832
Kevin M Dean https://orcid.org/0000-0003-0839-2320
Reto P Fiolka https://orcid.org/0000-0002-4636-5000

## Decision letter and Author response

Decision letter https://doi.org/10.7554/eLife.57681.sa1
Author response https://doi.org/10.7554/eLife.57681.sa2

## Additional files

### Supplementary files
• Supplementary file 1. Description of sample type, fluorescent labels, imaging conditions, data post-processing, and rendering, for each figure.

• Supplementary file 2. Encoding probe sequences for proximity ligation RNA fluorescence in situ hybridization.

• Supplementary file 3. Source, authentication method, and routine testing performed on cell lines.

• Transparent reporting form

### Data availability
Manuscript data is available on Zenodo, under the https://doi.org/10.5281/zenodo.4266823.

The following dataset was generated:

| Author(s) | Year | Dataset title | Dataset URL | Database and Identifier |
|---|---|---|---|---|
| Sapoznik E, Chang BJ, Huh J, Ju RJ, Azarova EV, Pohlkamp T, Welf ES, Broadbent D, Carisey AF, Stehbens SJ, Lee KM, Marin A, Hanker AB, Schmidt JC, Arteaga CL, Yang B, Kobayashi Y, Tata PR, Kruithoff R, Dubrovinski K, Shepherd DP, York AG, Millet-Sikking A, Dean KM, Fiolka RP | 2020 | A Versatile Oblique Plane Microscope for Large-Scale and High-Resolution Imaging of Subcellular Dynamics - Public Data | https://doi.org/10.5281/zenodo.4266823 | Zenodo, 10.5281/zenodo.4266823 |

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

## Appendix 1

Theoretical differences between the eSPIM (*Yang et al., 2019*) and the OPM presented here. A complete discussion is reported elsewhere (*Millett-Sikking and York, 2019*).

- Our cumulative theoretical NA is >1.28, whereas eSPIM has an NA >1.06.
- Our FOV is theoretically diffraction limited throughout a 110 × 110 micron region and expected to achieve 80% of the NA throughout a field of view of 220 microns, whereas eSPIM achieves a field of view of ~70×70 microns.
- The remote refocus optics remain theoretically diffraction limited throughout a range of ± 30 microns, whereas eSPIM is ± 10 microns.
- Working distance of our primary objective is 300 microns, whereas eSPIM has a 170-micron long working distance.
- The glass-tipped tertiary objective can operate with tilt angles from 0 to 45 degrees, whereas eSPIM is limited by housing and water chamber to 0–30 degrees.
- The choice of a silicone immersion objective reduces depth-dependent spherical aberrations when imaging within biological samples.
- The tertiary objective is dedicated, plan, color corrected between 450–700 nm, and can extract images from any choice of secondary objective so long as they have ~100 microns of clearance. In contrast, eSPIM uses a water-dipping lens with a coverslip that is not plan or coverslip corrected and can only be paired with a limited selection of secondary objectives. Furthermore, the water chamber is quite complex.
- The tertiary objective can be paired with a NA 0.95 secondary objective, whereas the eSPIM design had a maximum NA of 0.9 for the secondary objective. This is in part possible by the increased clearance engineered into the glass frustum that is attached to the tertiary objective.
- The intermediate tube lenses and scan lenses deliver the full 200-micron diameter field of view from the primary objective, whereas eSPIM uses non-optimized achromatic doublets which limit the field of view to ~100 microns.
- The secondary objective used here is a widely available fluorescence objective, whereas the secondary objective used in eSPIM is an industrial objective that has been discontinued.

## Appendix 2

### Choice of illumination angle, and tradeoffs associated with the 0 to 45-degree tilt

The tertiary objective is capable of accepting tilt angles between 0 and 45 degrees; a choice that is accompanied by several tradeoffs. A detailed discussion, and derivation, is provided by Millett-Sikking and colleagues (*Millett-Sikking and York, 2019*), and is freely available online (see https://github.com/amsikking/SOLS_optimum_tilt/blob/master/SOLS_optimum_tilt_quadratic_estimate.ipynb). These can be summarized as follows:

- By definition, the excitation light-sheet and emission path must share the same primary objective NA.
- As the tilt angle increases, thinner light-sheets can be produced, as more NA is available to shape the illumination beam.
- As the tilt angle increases, the extent in the z-direction (normal to the coverslip) that can be covered with a given light-sheet confocal parameter increases (e.g. the useful length of the light-sheet). In contrast, a smaller tilt angle reduces the usable height of the imaging volume that can be covered with the light-sheet. Thus, from a light-sheet perspective, a large tilt angle is advantageous.
- For low tilt angles, the detection efficiency and resolution improve owing to decreased reflective losses at the air-glass interface of the tertiary objective. Furthermore, if the tilt angle is 18 degrees or less, no light-clipping occurs (*Figure 1—figure supplement 1B*).
- Pragmatically, one wants the light-sheet to be thin enough that its beam waist is comparable to the depth of focus of the detection objective. With the NA 1.35 silicone immersion objective used here, this results in a tilt angle that is close to 30 degrees.

## Appendix 3

### Estimation of theoretical NA and light-collection angle

To analyze the theoretical NA, it is helpful to first analyze the collection angle of the tertiary objective, which has a nominal NA of 1 for oil immersion (n = 1.51). The tertiary objective does not use oil, but has a glass frustum attached, which essentially mimics oil immersion and creates an optically flat refractive index interface at the objective's focal plane. The half opening angle of the tertiary objective (41.47 degrees) reaches exactly the critical angle for total internal reflection at the glass-air interface. This means that the objective theoretically can collect light over a half opening angle of 90 degrees, or two pi steradian solid angle. This becomes obvious by the formulae for the critical angle and the definition of the NA:

The critical angle for a glass-air interface is given as:$\sqrt{\alpha_{crit} = \sin^{-1}\frac{1}{1.51} = 41.47}$

The definition of NA, $\sqrt{NA = n\sin\alpha}$, can be rearranged to:$\sqrt{\alpha = \sin^{-1}\frac{NA}{n}}$. For NA = 1 and n = 1.51, a half opening angle $\alpha$ of 41.47 results, which is obviously the same as above for the critical angle.

Another way to look at our tertiary objective is to compare its function to an 'elusive' NA one air objective. A NA of 1 would mean a half opening angle of 90 degrees in air, which would be highly beneficial for a tertiary objective in any OPM system to capture the most light emanating from the secondary objective. However, an NA one objective is not practical, as it would either have an infinitely large pupil for a finite working distance, or a working distance of zero for a finite pupil. Therefore, so far air objectives have always had an NA smaller than one.

In a sense, the tertiary objective presented here is the best approximation to an NA one air objective. This thought experiment also makes clear why the tertiary objective presented here has zero working distance: indeed, its frustum exactly ends at the focal plane of the objective.

To understand what we can do with such a tertiary objective, we next analyze which angles emanating from the secondary air objective (NA of 0.95, 71.8 degrees half angle) can be collected, depending on the tilt angle.

Let us first consider a zero-tilt angle, that is the secondary and tertiary objective are in-line and share the same optical axis (*Figure 1—figure supplement 1A*). Two marginal rays (red and magenta) coming out of the secondary objective at the highest allowed angles (±71.8 degree) are getting refracted toward the normal of the air-glass interface. This notably reduces the half opening angle from 71.8 degrees to 26.2 degrees inside the glass medium.

Next we consider a tilt angle of 18.2 degrees. In *Figure 1—figure supplement 1B*, the same marginal rays (red and magenta) are still both coupled into the tertiary objective. But the magenta ray enters the tertiary objective at the critical angle. This is the limiting case for which still the full light-cone coming out of the secondary objective can be transmitted into the tertiary objective.

If the tertiary objective is tilted more than 18.2 degrees, in our case by 30 degrees, one can see that it cuts into the cone of light of the secondary objective (light blue, *Figure 1—figure supplement 1C*). Thus, the light that can be coupled in at the critical angle occurs now at an opening angle of 60 degree at the secondary objective, which reduces the useful angular aperture of the secondary objective by 11.8 degrees. The red marginal ray is still coupled in at the glass interface, thus the full half angle of 71.8 degrees can be used on this side of the pupil.

Thus, an angular range of 60+71.8 = 131.8 degrees can be transmitted for a tilt angle of 30 degree. This corresponds to a useful NA of 1.4*sin(131.8/2)=1.28 of the primary objectives' NA (nominally 1.35). Normal to the tilt direction, no clipping occurs by the tertiary objective. Thus in this direction, an theoretical NA of 1.4*sin(71.8)=1.32 can be used.

The exact solid angle used by the presented system cannot be derived from the simplified sketches shown here but must be computed analytically or numerically from the overlap of the Ewald's spheres of each objective. Nevertheless, the value of 1.28 can serve as lower bound of the theoretical NA of our system. Since the upper bound is not much higher (1.32), we have decided to use the lower bound as an approximation here.

## Appendix 4

### Microscope alignment

For alignment purposes, a collimated green laser beam is injected into objective one from above in the diascopic direction. To align optical elements downstream, objective one is removed, and the size of the alignment beam is adjusted with an iris such that it either has a similar size as the back pupil of the primary objective, or that it becomes a narrow pencil-like beam (e.g. the beam divergence introduced by a lens is smaller than the divergence of the beam itself and can thus travel straight through a lens without changing much in size). Using the small pencil beam for alignment, back reflections are checked for every lens introduced into the optical train. Using the larger collimated beam, the correct distance and alignment between each pair of lenses is confirmed with a shear plate interferometer (e.g. the beam remains collimated). The alignment of the secondary objective is particularly critical. We place an iris downstream of where objective two will eventually go, as well as a frosted glass window with an aperture just before it (DG10-1500-H1-MD, ThorLabs). The hole is adjusted carefully in its height and lateral position such that the beam downstream is not steered or clipped in any way (checked by iris downstream). Now Objective two is introduced. It is carefully translated and rotated, while checking the back reflections on the frosted glass disk. If the correct alignment is found, a series of concentric rings should appear on the frosted glass disk. Simply holding the frosted glass disk by hand into the setup is not good enough, as minute displacements of the glass disk change the apparent back reflections from the objective. The hole of the glass disk needs to be well centered on the alignment beam.

Another critical alignment step is that the pupils of objective 1 and objective two are conjugated to each other. This can be verified by the 'beam wobble method', which is introduced below: Send a small pencil alignment laser beam over the OPM galvo mirror and put a card after objective 2. The beam emerging from objective two should be slightly diverging. Adjust the iris to obtain the smallest spot on the card. Now put a sinusoidal drive signal on the galvo mirror. If the objective is in the correct position, the beam exiting the objective does not move. If it is axially at the wrong position, the beam will wobble back and forth. If this is the case, carefully translate the objective axially and find the position where the beam appears stationary on the card. This is the position where objective two is conjugate to the galvo mirror. For Objective 1, inject a laser beam in the opposite direction (it can be the optogenetics laser beam, or the light-sheet beam without the cylindrical lens in place), which will pass over the OPM galvo and exit Objective 1. Use again a card after objective one and limit the beam size of the laser such that the smallest spot appears on the card. Use the same wobble trick to find the position of objective one where the beam is stationary. Now both objective's pupil planes are conjugate to the galvo, and thus they are also conjugate to each other.

To align the tertiary imaging system, we recommend to first install it in straight transmission. If correctly aligned, the collimated green alignment laser (full beam width, objective one removed) has to appear on the center of the sCMOS camera and it has to form a sharp focus (FWHM slightly above on pixel). When imaging 100 nm fluorescent beads, the microscope system has to generate near-diffraction limited PSFs (~270 nm FWHM for green emission, no visible aberrations on the beads) over the full field of view (full chip of the sCMOS). It further has to deliver this performance over a ±10-micron focus range. To test this, translate beads on a coverslip up and down 10 microns, then refocus the tertiary objective to focus on them. No visible aberrations should appear at the axial positions outside the nominal focal plane. These measurements reveal that the aberration-free remote focusing works correctly. If aberrations occur outside of the nominal focal plane, check carefully if a slight lateral displacement of objective one relative to objective two can improve it. We found that such alignment errors show up strong in the non-nominal focal planes. Only if good 3D imaging performance is achieved in straight transmission we can proceed further. Now the tertiary imaging system can be tilted. Importantly, when it is correctly aligned, the alignment laser beam (objective one removed, full beam width) should appear centered and focused (the beam width should only marginally increase compared to the straight transmission case) on the sCMOS camera.

For routine alignment checks, objective one is removed every week, and the focusing of the alignment laser is verified on the sCMOS camera. Irises in the beam path are used to see if there is any obvious drift of any components. Typically, the stage that holds objective three drifts a bit with temperature, as well as the relative focal planes of objective 2 and 3. If all the rest of the beam path is

correct (i.e. beam travelling centered through the irises), then usually a slight translation of objective three is needed to bring the beam back to focus and the center of the camera. Temperature drift of the focal plane of the primary objective one is corrected separately by forming a widefield image after the first tube lens on an inexpensive CMOS camera. If this widefield image is in focus, is well centered and looks aberration-free, the image on the sCMOS camera after the tertiary imaging system should also be centered and aberration-free. This procedure allowed us to maintain near-diffraction limited performance over long time periods of repeated use.

