## [Decision Letter]

**Acceptance summary:**

In this manuscript, authors describe an Oblique Plane Microscopy method that uses a bespoke glass-tipped tertiary objective with higher numerical aperture to achieve a combination of higher resolution and larger field of view imaging of biological specimens mounted in a range of imaging chambers including microfluidic devices. This advancement provides a combination of improved resolution, imaging throughput and ease of use over existing methods.

**Decision letter after peer review:**

[Editors’ note: the authors submitted for reconsideration following the decision after peer review. What follows is the decision letter after the first round of review.]

Thank you for submitting your work entitled "A Single-Objective Light-Sheet Microscope with 200 nm-Scale Resolution" for consideration by *eLife*. Your article has been reviewed by three peer reviewers, and the evaluation has been overseen by a Reviewing Editor and a Senior Editor. The following individual involved in review of your submission has agreed to reveal their identity: Rory Power (Reviewer #3).

Our decision has been reached after consultation between the reviewers. There was a very detailed, in-depth discussion of the reviewer comments during the consultation session. The overall conclusion that the reviewers and the reviewing editor reached during the consultation session is what formed the basis for the final decision. Based on these discussions and the individual reviews below, we regret to inform you that your work will not be considered further for publication in *eLife*. The summary of the most pertinent points from the consultation was that the main claims of the manuscript, in particular the improved spatial resolution, are not substantiated and the data presented do not support the main conclusions of the manuscript. In addition, the conceptual innovation was deemed insufficient and technical improvements were deemed incremental. I hope that you will find the extensive comments of the reviewers useful for improving your manuscript.

Reviewer #1:

In this manuscript the authors describe an advancement of the single-objective, tilted-plane light-sheet concept. The general idea is to create a 2D light-sheet with as large of angle into the sample as the objective can support. The emission light field is relayed by the objective and other optics to a place in the emission pathway where the illumination/emission plane can be imaged by a high numerical aperture lens that is tilted to the optical axis. This allows the formation of an un-tilted plane on a detector. As is described in the Introduction of the paper, the work presented here is the evolution of this concept. Earlier designs used lower numerical aperture objectives to “un-tilt” the image and somewhat compromised the performance by not capturing all the light at the highest angles. In this design, no light is lost due to compromises in NA. This is accomplished by using a custom-designed objective that is essentially a glass immersion lens, improving the NA via higher index of refraction.

The ability to use standard cell preparations, coverslips and optics makes the tilted-plane concept extremely practical and powerful. The advancement reported here is incremental but takes the concept from some NA compromise to no NA compromise, so it is not clear why a titled-plane setup would not use this approach. I therefore think there is enough potential impact to publish in *eLife*.

Given that the impact of this work is largely of its ability for more researchers to build and engage with light-sheet instruments, the authors should provide more practical information on setup and alignment. For example, a picture in the supplementary information of an actual implementation, a basic alignment guide and supporting optical/mechanical parts. Or at minimum, explain how they will get this in the hands of users/builders.

Reviewer #2:

In this manuscript, Fiolka and co-workers attempt to improve the spatial resolution of state-of-the-art implementations of oblique plane illumination microscopy (OPM), and apply their technique to image subcellular dynamics in single cell samples. The title of the manuscript made me enthusiastic that the authors had achieved a technical breakthrough in light-sheet fluorescence microscopy (LSFM), but to my dismay my enthusiasm has been severely dampened upon reading their manuscript carefully. Not only have the authors failed to convince me that they have improved significantly upon state-of-the-art OPM, their data and analysis do not support their central claim of “200 nm scale” resolution. There is little here that advances light-sheet microscopy and the authors appear to have either wilfully or accidentally ignored a large body of literature demonstrating LSFM implementations with better collection efficiency and resolution than what they claim here. Biological insight is scant at best, with experiments presented as “one-offs” without any statistical rigor or attempt at reproducibility. *eLife* asks us to evaluate manuscripts that are of the “highest scientific standards and importance in all areas of the life and biomedical sciences”, and this paper does not meet this bar.

1) LSFM imaging with a single high NA objective is nothing new, and published work already demonstrates higher NA imaging with better collection efficiency and resolution than what is reported here. There are multiple papers that achieve LSFM with a high NA lens with NA > 1.1. The earliest I could find in my literature review is Gebhardt et al., published in 2013. Here a reflective surface in combination with 1.35 NA and 1.40 NA objectives was used to study single molecule transcription factor binding in live mammalian cells. This concept was refined and made easier to use in 2015 and 2016 by Galland et al. and Meddens et al., who used reflective microfluidic chips to perform single-molecule imaging of a variety of cell- and embryonic samples at high NA detection. Meddens used a 1.2 NA lens and Galland a 1.3 NA lens, and Galland specifically is notable since by slightly raising the sample, imaging of the entire cellular sample is possible without clipping (unlike in Gebhardt). While neither Meddens nor Galland used a 1.35 NA oil objective, this is presumably a trivial exchange on their system, and even with their reported objective lenses I suspect the lateral spatial resolution is improved over this work (more on that below). In reading Galland in particular, I was struck by the conceptual similarity and goals outlined in that manuscript and this one: “The soSPIM configuration enables broad modularity. Phase, differential interference contrast, wide-field, high-resolution and super-resolution imaging can be performed on the same microscope. Switching optical magnification does not involve any realignment. The requirement of perfect mechanical alignment of the two objectives used in traditional SPIM is obviated by the use of a single objective combined with software-controlled alignment of the light sheet along the mirror axis. Long-term stability is enhanced by the use of a single objective and the ability to implement the perfect-focus system built into inverted microscopes.”

Galland seems to achieve many of the stated goals of this technique.

And then there is “LITE” microscopy, published in JCB in 2018. Here an angled light-sheet (2.4 deg with respect to the horizontal) is used combination with high NA optics, up to an NA of 1.49. This latter NA enables higher resolution imaging than that shown here, as assayed by images of beads with FWHM < 250 nm, pre-deconvolution.

Like the OPM technique described in this manuscript, all of these techniques have their drawbacks – the first three require the addition of mirrored surfaces to bend the light-sheet, and LITE presumably suffers from out-of-focus illumination over the field of view and spherical aberration when using the 1.49 NA lens. A second drawback is that none of these techniques can easily achieve the speed of OPM-based methods.

Nevertheless, a significant advantage that all these “true” single-objective LSFMs have over the authors' work is access to the full NA of the detection lens, without the inherent losses that come from using 3 detection objectives (a 71% light loss is a hefty price to pay for a system that does not achieve the full NA of the primary lens). A biologist wishing to eke out the maximum signal and resolution from her dim samples (e.g. as single molecules or endogenously tagged CRISPR constructs often are) might well choose to capitalize on the advantages of these published techniques, despite their drawbacks. Thus, why is none of this prior work discussed in the manuscript? The most charitable explanation is that the authors of this work are unaware of it, but I am worried that in their effort to “sell” their current technique the authors have purposefully ignored this previous body of work. An intellectually honest comparison of their method in comparison to these other, higher NA methods would improve this work – in particular the claim that they achieve the “highest lateral resolution in light sheet microscopy” would seem to be strictly false based on these prior LSFMs operating at higher NA.

2) Resolution claims are not backed up by the data, and do not show a significant advance over previous work.

a) The authors claim a “200 nm scale lateral resolution”. Their sole evidence for this claim appears to be the images of 100 nm beads that they deconvolve with the Richardson-Lucy algorithm. Using RL on beads as an exclusive resolution metric is at best naïve, and at worst misleading and wrong. RL is known to turn beads into points, i.e. regardless of the physical limits of the optical system it is possible to claim an arbitrarily good resolution if one over-deconvolves. How were the number of iterations decided and why are the convergence curves not shown? The numbers the authors produce raise other red flags -

b) The authors deconvolve their beads until they reach a value of 189 ± 6 nm. The cutoff frequency for the lens they use, assuming 500 nm light, is λ/(2 * NA) = 500 nm / (2 * 1.35) = 185 nm. At face value, the deconvolved numbers would then suggest, in the best-case scenario, that the authors achieve an NA very close to 1.35. Indeed, the number they report makes me wonder if they simply deconvolved their data until they got close to this theoretical value. Unfortunately, their own raw measurements suggest otherwise, as for the same beads in widefield mode (sans OPM detection) they report ~240 nm FWHM. Assuming that there are not additional aberrations in this primary lens, and the ~240 nm corresponds to the full 1.35 NA, in OPM mode the authors are relatively far from achieving diffraction-limited performance – they report 284 nm x 328 nm lateral FWHMs in the OPM configuration. Given that resolution scales with NA, these numbers would suggest (at best) an NA of 1.35 (240/284) = 1.14 in the direction normal to the scan, and an NA of 1.35 (240/328) = 0.99 in the direction along the scan. The combination seems to be no (or marginally) better than what is achieved in lattice light-sheet microscopy with a 1.1 NA lens.

c) The authors argue that OPM occupies a unique niche among LSFMs, in its ability to easily integrate into existing optical microscopy workflows. I am sympathetic to this argument, but am not convinced that the technique described here is conceptually new or even advantageous compared to the previous state-of-the-art in this field: Yang et al.5. As the authors correctly point out, the conceptual advance realized by Yang is that sticking a piece of high refractive index material between secondary and tertiary lenses enables better performance than previous OPM. Using a bespoke objective instead of the awkward coverslip/water construction used in Yang is definitely a step in the right direction towards making the method more practical and commercially viable but is not a conceptual advance. On top of this, Yang reports pre-deconvolution bead-based FWHM values of 316±8nm and 339±18 nm laterally and 596 ± 32 nm axially. The lateral values are ~10% worse than those reported here, which is sufficiently close that I have a very difficult time believing that the authors' work represents a notable advance over the previous work in Yang et al. i.e. the minor improvement they report in pre-deconvolution FWHM values is unlikely to make any qualitative difference to the kind of data/insight obtained against this previously published paper, nor does it justify a quantitative jump from “300 nm scale” to “200 nm scale”. Moreover, the axial FWHM value reported in Yang is ~25% better than the 823± 31 nm number reported here. Taken together these comparisons would suggest that this work represents a rotation in parameter space rather than an advance.

d) Perhaps most remarkably, the authors make no attempt to verify “200 nm scale” in any of their images, e.g. by empirically investigating if they can separate biological features at this distance or by running Fourier-based methods that provide pixel-based resolution maps. Doing so would go a long way to clarifying if indeed they have achieved “200 nm scale” resolution vs. something more like “250 nm-scale” or even “300 nm scale” resolution in biological samples. Ideally these calculations would be performed with and without deconvolution, so that the effect of deconvolution is cleanly separated from the “raw” imaging performance. Speaking of raw performance, it is unclear to me which of the presented data is deconvolved vs. raw. Do the authors present any raw data anywhere in this paper? This point needs to be clarified.

e) What is the pixel size resulting from the imaging system, and the data presented here? It is concerning that all data, with the exception of the beads, seems to have been at least displayed in 115 nm x 115 nm x 100 nm voxels. Nyquist alone would dictate that “200 nm scale” lateral resolution is not possible.

3) The axial resolution values the authors report are significantly worse than state-of-the-art multiview LSFM techniques8-11, which can achieve up to a near twofold axial resolution enhancement compared to their (likely) over-optimistic deconvolved values of 570 nm. These references need to be cited as they set the bar as far as axial resolution, not the lattice light-sheet system. Speaking of axial resolution, why is the axial FWHM not reported in addition to the lateral FWHM measurements in Fiugre 2—figure supplement 5? Please provide all XYZ measurements before deconvolution, so the “raw” FWHM values are evident for the interested reader. This will help avoid confusion about likely over-aggressive deconvolution.

Reviewer #3:

The authors report a single objective light sheet (SOLS) microscope that provides unprecedented numerical aperture for light sheet imaging. The microscope utilizes a recently available solid-immersion objective lens to re-image an appropriately relayed tilted image plane onto a camera while encoding high NA information at smaller ray angles, thus allowing their capture without steric interference between the remote objective pair. This represents a natural progression from the eSPIM (Yang et al., 2019), which first illustrated the use of ray compression in SOLS. The application of this technology to several applications that would otherwise be extremely challenging in a light sheet/live-imaging context follow and illustrate that the microscope can image traditional sample preparations, in microfluidic chips and with high volumetric image rates. These applications provide an appropriate showcase for the technology and promising routes for further investigation without providing substantive biological conclusions. Given the technological focus of the article, I believe the article nonetheless warrants publication in *eLife* when the issues below are addressed. More generally, I believe this work and the recent developments on which it is built, will be of the utmost significance to the cell biology community in the coming years.

The following major points should be addressed before publication:

1) Having said that this is a technologically focused article, at least in terms of the innovation in the approach, I actually felt that the article was lacking in this regard. I am aware that much of the development around the solid-immersion objective has been reported via non peer-reviewed sources (GitHub etc.). However, as the first peer-reviewed study to utilize this technology, a more thorough description of the system would nonetheless seem warranted, particularly with regard to optical simulations, aberrations/useable viewing field (e.g. comparing simulations with observations) and a discussion of the various trade-offs etc. This may seem outside the scope of a typical *eLife* publication and the manuscript as prepared but without it, the greatest achievement of this work (the microscope development, not the biology in this case) is sidelined. The authors are uniquely placed to be able to provide this information.

2) The authors include several comparisons with lattice light sheet microscopy that could be supported better with corresponding data and expanded discussion. This is noted with regard to the comments below. Similarly, I believe that the authors are uniquely able to provide the additional information via simulations and data that likely already exists e.g. fluorescent bead PSF measurements (e.g. Chang et al., 2019.) The authors would also benefit from comparing against some other reports in high-resolution light sheet microscopy e.g. Bessel beam (one/two-photon excitation, axially-swept light sheet microscopy).

Introduction : The authors note that previous LSFM implementations require mounting either on a small cover glass (e.g. lattice light sheet) or in agarose tubes (more classical SPIM-type systems). However, the authors fail to note that in the latter case, this is actually a superior mounting method e.g. for developing embryos as the soft surrounding environment promotes normal development (e.g. Kaufmann et al., Development, 2012), whereas a hard glass interface does not. I appreciate that these applications are outside the scope of what is presented here but I still think further clarification is warranted and that the desire to use traditional preparations is rather an issue for cell cultures and high-throughput imaging. The authors should nonetheless note that coverslip mounting may produce biological artifacts in the context of their earlier studies using collagen gels for mounting with a long working distance SPIM approach (Welf et al., Dev. Cell, 2016).

Results: Please comment on the choice of angle and the trade-offs associated with the 0 – 45 degree tilt.

Subsection “Instrument Characterization”: Can you comment on how this compares to eSPIM (Yang et al., 2019). The lateral resolution appears to be slightly superior but the axial resolution seems to be lower (assuming a Gaussian beam for eSPIM). Is this a result of a longer thicker light sheet or a difference in how tilted the light sheet is with respect to the detection axis? This is surprising given the additional ca. 0.2 NA provided by the reported system.

and an 823 nm axial resolution?

Subsection “Instrument Characterization” final paragraph: Can you comment on the treatment here. Taking the efficiency simply as NA^2^ gives ca. 1.5, 1.9x respectively. This is without taking into the account the small NA loss relative to the 1.35 provided by the Si-immersion objective. Nevertheless, this analysis is welcome and generally underreported in single-objective light sheet microscopy.

Figure 2 a) Is there a reason to choose a different color lookup table for this image?

Discussion second paragraph: Repetitive with regard to the introduction.

“…it has the highest lateral resolving power (~200 nm) in light-sheet

microscopy. Indeed, owing to the narrow depth of focus provided by the optical design, the axial resolving power” This is the deconvolved resolution. It would be helpful to include PSFs from light sheet microscopes and report both the raw and deconvolved resolutions in each case. The authors previous efforts regarding field synthesis should yield the required data (Chang et al., 2019).

Regarding contrast and axial resolution, which are tied together, the lattice light sheet supposedly maintains a thin profile over a larger distance than a purely Gaussian light sheet. The authors should report on the useable field of view when noting comparisons of contrast and resolution (and whether this is dominated by the requirements of a thin light sheet or rather the detection optics/remote focus scheme). The Gaussian sheet should produce better contrast at the focus whereas the lattice trades some contrast for field of view. However, the high detection NA in the SOLS case does mean that the light sheet will be thicker at waist than the depth of field and so contrast is also lost here. Again, the authors are uniquely placed to make these comparisons in a more rigorous manner. At the minimum, it would be helpful to report the length and thickness of the light sheet produced at the 0.2 – 0.3 NA (reported in the Materials and methods), to provide some measure of the effective field of view and contrast for the lattice and SOLS cases.

With regard to laser power requirements and synchronous multicolor imaging, this is largely a result of how lattice light sheets have typically been generated. The authors own previous research regarding field synthesis (which should be cited) has reduced these issues for anyone who may be choosing whether to construct a SOLS or lattice light sheet system. The primary issue remaining is the lack of isolation between the objectives and media and the steric issues.

Subsection “Data Post-Processing”: Please comment on the time taken and computational resources required for shearing and deconvolving the datasets. Since the SOLS system appears a good fit for imaging facilities, the associated throughput rate is a consideration.

[Editors’ note: further revisions were suggested prior to acceptance, as described below.]

Thank you for submitting your article "A Versatile Oblique Plane Microscope for Large-Scale and High-Resolution Imaging of Subcellular Dynamics" for consideration by *eLife*. Your article has been reviewed by three peer reviewers, and the evaluation has been overseen by a Reviewing Editor and Anna Akhmanova as the Senior Editor. Two of the original reviewers were consulted as well as a new expert for third opinion. The following individual involved in review of your submission has agreed to reveal their identity: Rory Power (Reviewer #3).

The reviewers have discussed the reviews with one another and the Reviewing Editor has drafted this decision to help you prepare a revised submission.

As you will see from the comments, the reviewers agreed that the revisions improved the paper and that the presented technique is interesting and provides certain advantages over existing methods. As such, the reviewers found the paper appropriate for *eLife*. That being said, two of the reviewers had very strong remaining criticism on the way the claims are presented. They felt that the pros and cons of the new method were not fairly discussed and that the comparisons to other methods were not appropriately presented. In particular, the reviewers felt that the trade-off coming from the poor light efficiency of the system was not properly discussed, comparisons to existing methods tended to highlight the strengths of the current method while not properly discussing the weaknesses such as " listing one dimension of an acquired sample (the lung tissue explant here) that's larger (8X) than what was imaged in other publications and then claiming this instrument is an improvement while leaving out the other dimensions that are clearly worse (45X thinner). ". In the consultation session, there was also concern over how the field of view comparisons were made and whether a tiled acquisition was being compared to a non-tiled acquisition with other methods, which the reviewers did not find to be a fair comparison.

Overall, the reviewers and myself believe that the claims need to be tempered down and the presentation on pros/cons of the method must be improved. I append below detailed comments from the reviewers, which should help you make these improvements.

Reviewer #3:

The authors have made substantive changes to the manuscript following the first round of reviews. I remain unswayed in my opinion that this work is suitable for publication in *eLife* given required improvements. I believe that the changes made have improved the manuscript.

In particular, the assessment of the system PSF and resolving capability has been greatly improved by including the non-deconvolved PSFs, details of the deconvolution and Fourier ring correlation estimate of resolution. The discussion of true single objective light sheet systems and comparison with eSPIM places the reported technique in context and I believe helps to make its case far more than detracting from it. The appendices should go some way to assisting others in building such a system.

There are still some changes based on my suggestions, which under ideal circumstances would be adopted. However, the author's arguments for not having done so are well reasoned. My primary concern was the frequent comparison with lattice light sheet microscopy. Here I still find the manuscript somewhat lacking but do appreciate that the authors field synthesis lattice light sheet system does not use the typical configuration of objective lenses making comparison difficult. The benefits of the reported system run deeper than pure resolution claims in any case and given the author's recent publication (Cheng et al. 10.1364/OE400164) reporting on the Gaussian to lattice light sheet comparison the omission here is reasonably justified.

All things considered, I would recommend the article for publication.

Reviewer #4:

First, I wish to commend the authors for their thorough revision. They have addressed many of my concerns and I feel better about the overall manuscript. The emphasis on field of view over resolution, the Fourier-based methods of assessing resolution, the citing of previous work, the additional details put forth in the appendices, and more accurately referring to their method as an OPM have all significantly improved the work. Here are my remaining concerns/comments:

a) The authors now more fairly describe the advantages of their method over previous similar microscopes. They still do not clearly state what I see as weaknesses of their method, which seems important given the extensive comparisons to LLSM and other systems:

i) relatively low sensitivity. In their revised manuscript, the authors now seem to admit that the advantage in light efficiency offered by the 1.35 NA lens is less than optical losses through their emission path. In sum, these losses would seem to imply a strictly lower sensitivity than the more “conventional” LLSM with 1.1 NA detection. Please discuss this point clearly in the main text, rather than relegating it to Materials and methods.

ii) Low effective NA vs. theoretical NA. I appreciate the appendix describing the NA calculations, but I am still not convinced that the number presented is an “effective NA”, i.e the NA of an equivalent widefield with primary lens using this NA. In particular, I am still bothered by the apparent loss of resolution compared to their widefield performance, which I commented on previously and repeat here:

“Unfortunately, their own raw measurements suggest otherwise, as for the same beads in widefield mode (sans OPM detection) they report ~240 nm FWHM. Assuming that there are not additional aberrations in this primary lens, and the ~240 nm corresponds to the full 1.35 NA, in OPM mode the authors are relatively far from achieving diffraction-limited performance – they report 284 nm x 328 nm lateral FWHMs in the OPM configuration. Given that resolution scales with NA, these numbers would suggest (at best) an NA of 1.35 (240/284) = 1.14 in the direction normal to the scan, and an NA of 1.35 (240/328) = 0.99 in the direction along the scan. The combination seems to be no (or marginally) better than what is achieved in lattice light-sheet microscopy with a 1.1 NA lens.”

It is very reasonable to suppose that NA would scale with the apparent size of a subdiffractive object, regardless of suboptimal Strehl ratio, MTF, hardware etc. And if the NA of the primary lens is effectively less than 1.35, this would seem to further diminish the “effective” NA that is achieved in this manuscript. This point is important to address properly, because it has implications both for comparisons to other microscopes and it suggests that the deconvolved numbers the authors present are still overly optimistic (e.g. if the NA is not in fact 1.28, physically it does not seem possible to achieve a deconvolved resolution of ~203 nm as in Figure 1H). I would suggest replacing “effective NA” with “theoretical NA” everywhere the former is referred to, and being very clear in the manuscript that there is still residual resolution loss: i.e. the theoretical NA is not achieved at the tilt settings used in this paper. The authors could prove me wrong by providing a suitably compelling measurement of the “effective NA” rather than the theory – presenting a theoretical argument does nothing to convince me that in fact they achieve this value, which their own measurements seem to contradict.

b) Regarding deconvolution, I am still having difficulty understanding how ~203 nm resolution is possible with 115 nm voxels. Once more, Nyquist alone would seem to render this number incorrect. The authors' argument in using a factor 1.414 seems to boil down to: “other people have done it, so we should do it too”. This is not a good argument in my opinion, as it leads to a result that contradicts common sense. The authors digitally upsample the bead data before deconvolution, but I do not see how such digital upsampling could effectively beat Nyquist.

The authors in their rebuttal seem to suggest that presenting the raw values sans deconvolution is unusual in multiview LSFM (“I think this is by far more forthcoming than many other manuscripts, including Multiview LSFM techniques, which only report resolution after iterative deconvolution”). I am not sure what they are talking about: Wu 2013 (diSPIM), Keller 2015 (IsoView), classics in this area, state these values clearly in the main text.

c) The authors seem to acknowledge in their rebuttal that they do not achieve diffraction-limited performance (“we agree we do not achieve diffraction-limited performance”), but I still find multiple places in the manuscript where this is stated or implied. Please address this point.

d) Depth. What sets the effective depth limit of this technique, as presented here? 25-30 μm is relatively modest for a lens with a 300 μm working distance, so discussing theoretical and practical limits would be useful for a biologist who is searching for a practical solution for their sample.

e) The authors state numerous times that they achieve resolution on par with or better than LLSM. Please explicitly state the corresponding values reported in LLSM (or measure them with an LLSM with 1.1 NA detection), where relevant.

f) “And unlike state-of-the-art multiview LSFM techniques that achieve a slightly better axial resolution, only a single objective and imaging perspective is needed (Guo et al., 2020; Wu et al., 2013).” Wu 2013 achieves ~350 nm axial resolution, Wu 2017 achieves ~300 nm axial resolution, and Guo 2020 enhances the effective axial resolution of LLSM to ~380 nm resolution. These are not “slightly better”, they are significantly better (indeed, the degree of improvement is more convincing than what is shown here relative to LLSM). Also, as I think the authors acknowledge in their rebuttal, they do not use a single objective, rather they use 3. Please remove “slightly” for accuracy and/or rethink this sentence.

Reviewer #5:

In this manuscript, "A versatile oblique plane microscope for large-scale and high-resolution imaging of subcellular dynamics", Sapoznik et al. describe a variant of an oblique plane microscope (OPM) that makes use of a custom designed tertiary objective to capture a greater portion of the emitted rays through the oblique refocusing objectives. OPM combined with light sheet is attractive because it offers the potential for optically sectioned, low phototoxicity imaging using only a single primary imaging objective with 180 degree physical access. As the authors note, this concept itself is not new and many descriptions and implementations have been described before (originally with Dunsby in 2008 and more recently with Bouchard 2015 and Yang 2019). Thus, the primary technical innovation here is the implementation of the custom-designed solid immersion lens.

In general, I think this is a useful addition to the field and could be suitable for publication in *eLife*. An open top-light sheet microscope with increased sample accessibility is indeed useful. However, in its current form, the manuscript reads more like an aggressive sales pitch rather than a balanced discussion of the pros and cons of different microscope approaches. The presentation is over-dismissive of prior work, often mis-representing or cherry-picking specific comparisons to make this current instrument appear better, while neglecting to mention or discuss trade-offs wherein the instrument might perform worse. I noticed that the authors disclose financial relationships with companies that sell products that compete with commercial versions many of the instruments compared here. However, I feel that the readers of *eLife* would benefit if the authors focused on more thoroughly documenting their own scientific contribution, together with its trade-offs, rather than (often incompletely or inaccurately) characterizing prior work from others.

1) The resolution claims here are provided by FHWM measurements of isolated beads and of xy (lateral) measurements using correlation based approaches on the images. Importantly, optical sectioning is not discussed at all. In this instrument, the resolution is provided by the high NA primary objective, but due to the thick lightsheets (0.16 NA Gaussian beams) used in most of the measurements, the lightsheets are substantially thicker than the depth of field for the detection. This would be readily apparent as the "missing cone" in the optical transfer function and I suspect it would also affect correlation based axial resolution measurements if they were conducted on cells or embryos rather than isolated beads. In contrast, axial resolution in several other approaches (Gaussian, Bessel, Lattice etc) is obtained by using a lightsheet that is thin-compared to the detection depth of field. The authors' own prior work, Dean et al., 2015, have highlighted the advantages of this approach, but there is no mention of the relationship between optical sectioning and resolution here.

FWHM measurements in isolation are a limited picture of resolution. The authors should present optical transfer functions in the xy, xz, and yz planes to demonstrate true resolution and to what extent their system fills in the missing cone. They should also perform the raw image-based correlation measurements in the axial dimension (as they already do now for the lateral dimension) on a range of biological samples. This could be easily done with existing data.

2) The authors describe that, due to the optical path required for OPM, the instrument loses 71% of the photons from the sample before they hit the camera. I.E. it has a transmission efficiency of 29%. This is in stark contrast to the standard SPIM systems which due to the simple widefield detection optics should operate at very close to the ~80-90% transmission efficiency of commercial objectives. Further, it's not clear at what imaging angle this measurement was performed. This information is also important because the efficiency decreases further as the primary objective is utilized further away from its design standard angle of 0 degrees. In the appendix, the authors describe how the 1.2 NA of this system would somewhat compensate for this poor transmission over the 1.1NA lens used in other variants, but this analysis assumes that all rays have equal transmission efficiencies which generally isn't true. Thus even without any additional optics, an OPM system operating with an off-centered pupil will have a lower transmission efficiency than an objective with the same effective NA operating on axis.

Experimental characterization of how the transmission efficiency between a 1.2 NA objective operating with a centered pupil vs. a higher NA objective with an effective 1.2 NA due to pupil decentering with OPM and associated optics, and how this declines with the OPM angle would be extremely useful for the field. The authors are ideally poised to make these type of measurements.

Regardless, this is an important discussion, so I'm unclear why the measurements are not mentioned in the main text under "Instrument Characterization". They are instead provided in the fourth paragraph of Materials and methods section under "Laser Scanning Microscope Setup". Given that microscope end users are willing to pay thousands more for a back thinned camera with a 10% increase in quantum efficiency than a non-back thinned unit, the authors should mention that the OPM implementation here comes at the cost of reduced detection efficiency in the main text. The statement that the instrument is "sufficiently sensitive to detect single molecules (data not shown)" does not adequately address this concern. Especially when other approaches have clearly demonstrated their utility for live cell and super-resolution single molecule imaging with both dyes and fluorescent proteins alike. It's also not clear how the authors define "without obvious signs of phototoxicity". How was this measured and what is considered obvious?

3) I have concerns about how prior methods are discussed. For example, Lattice Lightsheet Microscopy is not a single instrument, but a general description of the use of excitation patterns based on optical lattices for microscopy. The specific choices for excitation objective and detection objective as well as the type of lattice light sheet and extent of optical confinement used will determine the resolution and optical sectioning and can be chosen/optimized for specific applications. Thus, statements like "Lattice Lightsheet microscopy has a resolution of xx" or "requires a 5mm coverslip" are no more accurate than saying "Gaussian beam microscopy has a resolution of xx and requires agarose sample embedding". It all depends on how one decides to configure an instrument and for what purpose they choose to balance the tradeoffs. It's fine to say that a specific publication reported certain values, but this should be accompanied by some context. In many cases, design considerations may have been chosen to make an instrument that is optimized for a different purpose or with additional features than the one presented here.

---

## [Author Response]

[Editors’ note: The authors appealed the original decision. What follows is the authors’ response to the first round of review.]

Reviewer #1:In this manuscript the authors describe an advancement of the single-objective, tilted-plane light-sheet concept. The general idea is to create a 2D light-sheet with as large of angle into the sample as the objective can support. The emission light field is relayed by the objective and other optics to a place in the emission pathway where the illumination/emission plane can be imaged by a high numerical aperture lens that is tilted to the optical axis. This allows the formation of an un-tilted plane on a detector. As is described in the Introduction of the paper, the work presented here is the evolution of this concept. Earlier designs used lower numerical aperture objectives to “un-tilt” the image and somewhat compromised the performance by not capturing all the light at the highest angles. In this design, no light is lost due to compromises in NA. This is accomplished by using a custom-designed objective that is essentially a glass immersion lens, improving the NA via higher index of refraction.The ability to use standard cell preparations, coverslips and optics makes the tilted-plane concept extremely practical and powerful. The advancement reported here is incremental but takes the concept from some NA compromise to no NA compromise, so it is not clear why a titled-plane setup would not use this approach. I therefore think there is enough potential impact to publish in eLife.Given that the impact of this work is largely of its ability for more researchers to build and engage with light-sheet instruments, the authors should provide more practical information on setup and alignment. For example, a picture in the supplementary information of an actual implementation, a basic alignment guide and supporting optical/mechanical parts. Or at minimum, explain how they will get this in the hands of users/builders.

We would like to thank the reviewer for their kind words, and we are happy that they recognize the power of the approach and believe that our method has sufficient impact for publication in *eLife*. We also agree that providing practical information on the setup and alignment of the microscope would be useful. As such, we now provide a basic alignment guide in Appendix 4. Furthermore, we have tried to make the Materials and methods section as exhaustive as possible, while pointing potentially interested readers to online sources of information that include complete parts lists.

Alignment:

We now provide in Appendix 4 a section with tips and tricks for aligning the microscope.

We are also working pro bono with Applied Scientific Instrumentation so that they may provide turnkey optical components for assembling the microscope.

In the Materials and methods, Laser Scanning Microscope Setup, second paragraph, last sentence, we now point individuals to the complete parts list provided by Millett-Sikking et al.

We are working to place a CAD rendering of the microscope online, but this is a work in progress. Nonetheless, if the reviewer sees this as critical, we assure you that we will have this completed prior to publication of the manuscript.

Reviewer #2:In this manuscript, Fiolka and co-workers attempt to improve the spatial resolution of state-of-the-art implementations of oblique plane illumination microscopy (OPM), and apply their technique to image subcellular dynamics in single cell samples. The title of the manuscript made me enthusiastic that the authors had achieved a technical breakthrough in light-sheet fluorescence microscopy (LSFM), but to my dismay my enthusiasm has been severely dampened upon reading their manuscript carefully. Not only have the authors failed to convince me that they have improved significantly upon state-of-the-art OPM, their data and analysis do not support their central claim of “200 nm scale” resolution. There is little here that advances light-sheet microscopy and the authors appear to have either wilfully or accidentally ignored a large body of literature demonstrating LSFM implementations with better collection efficiency and resolution than what they claim here. Biological insight is scant at best, with experiments presented as “one-offs” without any statistical rigor or attempt at reproducibility. eLife asks us to evaluate manuscripts that are of the “highest scientific standards and importance in all areas of the life and biomedical sciences”, and this paper does not meet this bar.

We appreciate the reviewer’s careful analysis of the manuscript, as well as their forthright criticism.

Resolution: We have substantially strengthened our resolution measurements and they now include Decorrelation Analysis and Fourier Ring Correlation Analysis on both beads and biological data, before and after deconvolution. We also present raw and non-deconvolved data for both beads and cells. These measurements again confirm that our microscope delivers resolution that is equal or better than Lattice Light-Sheet Microscopy for raw, and non-deconvolved data (see 10.1364/OE.400164, 10.1038/nature22369, 10.1364/BOE.11.000008). Further, we now more clearly show that this level of resolution can be maintained over a large field of view.

Literature: It was not our intention to “wilfully ignore” single objective light-sheet fluorescence microscopes based upon micromirrors, cantilevers, or “lateral interference tilted excitation”, and we have rectified this error by discussing these methods in the Introduction and comparing our performance to them throughout the manuscript. From our analysis, it is clear that many of these techniques cannot compete with the field of view, volumetric imaging capacity, resolution, imaging speed, and practical usability that we demonstrate with our method.

Biology: Admittingly, many of the biological experiments presented in our manuscript are indeed “one-offs”. However, we only ask that the reviewer holds us to the same standard as every other microscopy development paper, including those published in *eLife* (10.7554/*eLife*.14472, 10.7554/*eLife*.32671, 10.7554/*eLife*.40805, 10.7554/*eLife*.45919, 10.7554/*eLife*.46249). We consider it unfair to stipulate that we need to solve a biological conundrum while improving the field of view by a factor of 3.7 and matching the resolution of Lattice Light-Sheet Microscopy, a method which is onerously complex and incompatible with some of the samples imaged here. We have now more systematically analyzed PA-Rac1 stimulation and can show a significant increase in protrusion speed and duration compared to control cells.

Statistical rigor: This statement is demonstrably false, as we report quantitative metrics for the diffusion of cytosolic tracers, protrusion dynamics upon optical stimulation of Rac1, and evaluate the point-spread function of a large number of beads. Importantly, each measurement is accompanied by either standard deviations or 95% confidence intervals. With regard to the statistical robustness of the remaining biological observations, we are in agreement with Yu et al. (10.7554/*eLife*.46249) when they state that “the goal of the current paper was to demonstrate a technology, rather than test a hypothesis, [so] we did not pre-determine any sample sizes for this study.”

Scientific Standards: We have not only met the scientific standards commonly adopted in optical microscopy but exceeded them by meticulously providing details regarding the acquisition, processing, and analysis of our data, and by making publicly available all data and software. Historically, we have been penalized for not deconvolving our data, as well as for deconvolving our data. Unlike many other manuscripts, we clearly state the resolution of our raw data, and limit the number of iterations of Richardson-Lucy to avoid misleading results. Indeed, we limit the resolution enhancement to a factor of 2≈1.41, which is commonly accepted by many in the super-resolution fields as theoretically appropriate. In contrast, publications using lattice light-sheet microscopy regularly deconvolve their data until they achieve up to 1.9-fold resolution enhancement in the axial resolution and with one notable exception (10.1038/nature22369) fail to disclose raw resolution values (10.1364/OE.400164., 10.1364/BOE.11.000008).

Importance in all areas of the life and biomedical sciences: We have presented a truly versatile microscope that uniquely delivers resolution, field of view, speed, subcellular optogenetics, and the ability to image an incredibly diverse collection of specimens that range from single cells to entire tissues. We politely disagree with your diagnosis that it is not of importance in all areas of the life and biomedical sciences.

1) LSFM imaging with a single high NA objective is nothing new, and published work already demonstrates higher NA imaging with better collection efficiency and resolution than what is reported here. There are multiple papers that achieve LSFM with a high NA lens with NA > 1.1. The earliest I could find in my literature review is Gebhardt et al., published in 2013. Here a reflective surface in combination with 1.35 NA and 1.40 NA objectives was used to study single molecule transcription factor binding in live mammalian cells.

This is a classic paper from Sunnie Xie’s lab which introduces a light-sheet from the diascopic direction with a water immersion objective and a reflective cantilever. Nonetheless, by placing the specimen in contact with the water immersion objective and cantilever, it will no longer be sterile. Furthermore, the specimen must be in immediate proximity to the cantilever tip, which is technically challenging and limits the field of view to roughly 1/200^th^ of what we report for our OPM. Further, the light-sheet cannot reach the bottom of the cells, as it would get aberrated by the coverslip. Proper use of the AFM-based microscope also requires recalibration for every imaging error (see their methods). Perhaps most importantly, this method is incompatible with many of the specimens that we image with our OPM, including the microfluidic, tissue sections, neurons, etc. As such, while we agree with the author that it is a demonstration of a high NA single objective light-sheet, it lacks the general usability and performance (Field of view, volumetric imaging rate) that we demonstrate. But we include it now in our manuscript.

This concept was refined and made easier to use in 2015 and 2016 by Galland et al. and Meddens et al., who used reflective microfluidic chips to perform single-molecule imaging of a variety of cell- and embryonic samples at high NA detection. Meddens used a 1.2 NA lens and Galland a 1.3 NA lens, and Galland specifically is notable since by slightly raising the sample, imaging of the entire cellular sample is possible without clipping (unlike in Gebhardt). While neither Meddens nor Galland used a 1.35 NA oil objective, this is presumably a trivial exchange on their system, and even with their reported objective lenses I suspect the lateral spatial resolution is improved over this work (more on that below). In reading Galland in particular, I was struck by the conceptual similarity and goals outlined in that manuscript and this one: “The soSPIM configuration enables broad modularity. Phase, differential interference contrast, wide-field, high-resolution and super-resolution imaging can be performed on the same microscope. Switching optical magnification does not involve any realignment. The requirement of perfect mechanical alignment of the two objectives used in traditional SPIM is obviated by the use of a single objective combined with software-controlled alignment of the light sheet along the mirror axis. Long-term stability is enhanced by the use of a single objective and the ability to implement the perfect-focus system built into inverted microscopes.”Galland seems to achieve many of the stated goals of this technique.

Galland et al. use a sophisticated microfluidic method with embedded micromirrors to illuminate the specimen with a light-sheet. While we agree that changing the objective on this system is trivial, the photolithography, sequential anisotropic and dry silicon etching, and sputtering of gold surfaces to create a micromirror, is not. Furthermore, such a design requires that the specimen be placed into a microcavity, which again limits the types of specimens that can be imaged. Even for single cell aggregates, Galland et al. had to collect multiple images per image plane with the illumination beam defocused by 20 microns with an electrotunable lens, and then stitch the images together. Similarly, also the bottom of the cell cannot readily be imaged, as the light-sheet would be distorted. Very little information is provided regarding how they performed or analyzed their resolution, and a single value is provided (565 nm) for the nuclear lamina. While this may be similar to our axial resolution upon increasing the excitation NA, insufficient information is provided for proper evaluation as statistics are not provided. For samples larger than 28 microns, they have to substantially decrease the illumination of the NA. Thus, our OPM is much more user friendly and capable of imaging a much more diverse array of samples.

Meddens et al. also use a sophisticated microfluidic method with an embedded micromirror, but their illumination train does not have a tunable lens. Also, rather than having microcavities, Meddens et al. flow the cells into a microfluidic chip, which necessitates that the sample advantageously settles in a region adjacent to the mirror. Also, no information is provided regarding their resolution in the absence of STORM imaging. Thus, this method suffers from many of the same drawbacks as Galland et al. with regard to resolution, ability to image diverse specimens, field of view, speed, and more. Indeed, it is clear from their images that our OPM method delivers superior imaging results.

And then there is “LITE” microscopy, published in JCB in 2018. Here an angled light-sheet (2.4 deg with respect to the horizontal) is used combination with high NA optics, up to an NA of 1.49. This latter NA enables higher resolution imaging than that shown here, as assayed by images of beads with FWHM < 250 nm, pre-deconvolution.

As the reviewers has mentioned later, the use of a high NA oil immersion lens results in significant spherical aberrations, which necessarily leads to a depth dependent lateral and axial resolution as well as a diminished sensitivity (owing to a decreased Strehl ratio). Fortunately, while changing the objective in such a system is trivial, one must ask why they didn’t use a water or silicone immersion objective in the first place. Unfortunately this means that they are only competitive with our technology at the glass interface. Once they image a few microns into the specimen, the performance noticeably deteriorates. Also, in an effort to cover a field of view 150 microns, they use a light-sheet that is 4.3 microns thick. Thus, their optical sectioning will also be quite poor owing to a large amount of out-of-focus illumination and image blur. It is essentially “widefield” from the side. For higher NA illumination beams, the quadruple-slit photomask used results in periodic intensity modulation along its propagation axis of the illumination beam, rendering it non-quantitative. While quite convenient, LITE requires custom imaging chambers and is thus not compatible with the diverse array of specimens that we imaged (e.g., microfluidics, 96-well plates). Ultimately, LITE is in principle capable of imaging a similar sized field of view as us, but at a complete loss of optical sectioning capability. Further it lacks the imaging speed and versatility that makes our system powerful and is really only competitive in terms of resolution in the region adjacent to the glass coverslip.

Like the OPM technique described in this manuscript, all of these techniques have their drawbacks – the first three require the addition of mirrored surfaces to bend the light-sheet, and LITE presumably suffers from out-of-focus illumination over the field of view and spherical aberration when using the 1.49 NA lens. A second drawback is that none of these techniques can easily achieve the speed of OPM-based methods.

Thank you, we agree completely. We find that the addition of a mirrored surface near the specimen drastically limits the versatility of the microscope and makes routine manufacture of the mirrored surface or operation of the microscope quite technical. We agree that these techniques need to be discussed and their strength and weaknesses need to be better detailed.

Nevertheless, a significant advantage that all these “true” single-objective LSFMs have over the authors' work is access to the full NA of the detection lens, without the inherent losses that come from using 3 detection objectives (a 71% light loss is a hefty price to pay for a system that does not achieve the full NA of the primary lens). A biologist wishing to eke out the maximum signal and resolution from her dim samples (e.g. as single molecules or endogenously tagged CRISPR constructs often are) might well choose to capitalize on the advantages of these published techniques, despite their drawbacks.

The collection efficiency of microscopes varies widely and is rarely reported. For example, the collection efficiency of a single molecule confocal microscope can be as low as ~3% (see 10.1529/biophysj.108.134346). Nonetheless, we do agree that every microscope genre occupies its own niche. If a biologist needs every photon, we would recommend a TIRF (and potentially HiLO or grazing incidence modes) microscope because it combines high NA optics, 100% illumination duty cycles, optical sectioning, and aberration free detection. Unfortunately, TIRF and HiLO are not very useful for volumetric imaging. Nonetheless, we would not recommend that someone image with a severely aberrated PSF either (e.g., like LITE), as this will necessarily degrade their sensitivity as well.

Perhaps most importantly, we have clearly demonstrated that despite our losses, we are able to image a diverse range of specimens with negligible photobleaching or phototoxicity. Although we do not present the data, we have also performed live-cell single molecule imaging of genetically encoded fluorescent proteins with this microscope. This single molecule sensitivity is consistent with the eSPIM paper, as well as Xiang Zhang’s paper (which uses a polarizing beam splitter in the detection path that automatically throws away 50% of the detected light without accounting for other losses, 10.1038/s41592-019-0510-z). Thus, we consider the losses manageable, and we are also working on reducing them (e.g. reducing the number of relay lenses, e.g. via lensless optical scanning approaches).

Thus, why is none of this prior work discussed in the manuscript? The most charitable explanation is that the authors of this work are unaware of it, but I am worried that in their effort to “sell” their current technique the authors have purposefully ignored this previous body of work. An intellectually honest comparison of their method in comparison to these other, higher NA methods would improve this work – in particular the claim that they achieve the “highest lateral resolution in light sheet microscopy” would seem to be strictly false based on these prior LSFMs operating at higher NA.

As previously mentioned, we now discuss this literature in the manuscript. We did not purposefully ignore these papers in an effort to “sell” our manuscript. To suggest that we would ignore a subset of the literature in an effort to advance our own science at the cost of our colleagues is offensive. Instead, it was an honest error of judgment. When we wrote this manuscript, we were focused on lattice light-sheet microscopy, which is by many perceived as the dominant and ultimate light-sheet microscope for high-resolution imaging. While the reviewer is correct that single objective microscopes with high NA lenses have the potential to beat the resolving power of a lattice light-sheet instrument, lattice is much more versatile in the range of samples that are being imaged. As such, we were too narrowly focused on lattice light-sheet microscopy as the main competition.

Also, we made the semantic error of referring to our oblique plane microscope as a single objective light-sheet microscope. Regardless, we do not consider these techniques competitive in terms of instrument performance (again, diversity of specimens, speed, field of view, optogenetics…) when compared to our oblique plane microscope. We agree that the term highest resolution is as of now not warranted, and we have reduced the resolution claims accordingly.

2) Resolution claims are not backed up by the data, and do not show a significant advance over previous work.a) The authors claim a “200 nm scale lateral resolution”. Their sole evidence for this claim appears to be the images of 100 nm beads that they deconvolve with the Richardson-Lucy algorithm. Using RL on beads as an exclusive resolution metric is at best naïve, and at worst misleading and wrong. RL is known to turn beads into points, i.e. regardless of the physical limits of the optical system it is possible to claim an arbitrarily good resolution if one over-deconvolves. How were the number of iterations decided and why are the convergence curves not shown? The numbers the authors produce raise other red flags -

We now provide orthogonal measures of instrument resolution, including Decorrelation Analysis and Fourier Ring Correlation Analysis, and they are largely in agreement with the previous measurements. Importantly, we report these values for raw and deconvolved objects including beads and intracellular targets. Nonetheless, we would also like to state that a large number of microscopy papers report RL deconvolved resolution values (e.g., Lattice, iSIM, ISM, diSPIM, …).

RL: While we agree that RL deconvolution can result in unrealistic resolution values for point sources, we clearly state in the manuscript that we perform the minimum number of iterations necessary (always less than 20) to achieve a 2≈1.41 resolution enhancement, which is considered acceptable by in the super-resolution field as appropriate (i.e. iSIM, ISM, and other confocal photon-reassignment techniques claim two fold resolution improvement, where a factor 2 improvement comes physically from the small pinholes, and another 2 comes from iterative deconvolution. If their resolution gain by iterative deconvolution was illegitimate, it would essentially put them out of the super-resolution field). And the 1.41 gain is significantly less than others in the field, including the ~1.9 axial resolution enhancements in Lattice Light-Sheet Microscopy. Additionally, we did perform the convergence analysis, but did not present the data in this manuscript as it is not as clear cut as we hoped it would be (see Author response image 1, the axial resolution started to level off after 15 iterations, but the lateral resolution was still shrinking). Thus, we used the empirical square root of two criterion, which is not uncommon in the microscopy field. But importantly, we provide the raw data too, so the reader can clearly judge what comes from raw performance, and how much from deconvolution.

We are however sympathetic to the issues with deconvolution, and we have reduced our claims based on the deconvolved values, as estimated by FRC and image decorrelation analysis. We now emphasize the performance over the entire field of view, which we think is impressively uniform. This is a more valuable advantage of our microscope than resolution gains in the range of tens of nm.

b) The authors deconvolve their beads until they reach a value of 189 +/- 6 nm. The cutoff frequency for the lens they use, assuming 500 nm light, is λ/(2 * NA) = 500 nm / (2 * 1.35) = 185 nm. At face value, the deconvolved numbers would then suggest, in the best-case scenario, that the authors achieve an NA very close to 1.35. Indeed, the number they report makes me wonder if they simply deconvolved their data until they got close to this theoretical value. Unfortunately, their own raw measurements suggest otherwise, as for the same beads in widefield mode (sans OPM detection) they report ~240 nm FWHM. Assuming that there are not additional aberrations in this primary lens, and the ~240 nm corresponds to the full 1.35 NA, in OPM mode the authors are relatively far from achieving diffraction-limited performance – they report 284 nm x 328 nm lateral FWHMs in the OPM configuration. Given that resolution scales with NA, these numbers would suggest (at best) an NA of 1.35 (240/284) = 1.14 in the direction normal to the scan, and an NA of 1.35 (240/328) = 0.99 in the direction along the scan. The combination seems to be no (or marginally) better than what is achieved in lattice light-sheet microscopy with a 1.1 NA lens.

It is well known that the Abbe diffraction limit is an optimistic estimate for resolution and real-world high NA imaging systems achieve ~20% less resolution, due to many factors (Strehl ratio <1, attenuation of marginal rays, modulation transfer function, etc.). Thus, we do not agree with the estimates of the effective NA based solely on the achieved resolution. If we calculate analytically the opening half angle that we use, the effective NA for detection is 1.28 (Appendix 3). We agree that our measurements underperform in the oblique plane microscope format. We noticed that if we tilt the tertiary imaging path to zero degree, the lateral raw resolution gets uniformly ~270nm, which is more in line with the estimated NA. This is after going through a lot of lenses, which verifies that the basic optical configuration and alignment is sound. That said, it also shows that the tilting of the tertiary imaging system causes some residual resolution loss and stipulates that lower inclination angles could be more advantageous for achieving the highest resolution. We discuss these trade-offs now more clearly in Appendix 2.

However, we do agree with the statement that the practical resolution we achieve is not dramatic: Fourier based analyses put our lateral resolution at 220+-23nm (FRC) and 251+-3nm (Image Decorrelation) for deconvolved biological data, lattice light-sheet microscopy reports a lateral resolution value of 250nm. Nevertheless, the fact that one can get similar or even better performance with a single primary lens, enabling much more flexibility on sample mounting, is an important and remarkable finding that allows us to image a much more diverse set of specimens. Lastly, we notice that despite our efforts to unlock even higher resolution in OPM, it is equally advantageous to increase the field of view and the resolution uniformity, as this permits imaging throughput. We can cover 180x180 micron with quite uniform resolution, which is after reconsideration, more valuable than arguing about a +/- 10 nm resolution gain. As such, we de-emphasized some of our resolution claims and shifted some attention towards the field of view. Notably, lattice light-sheet microscopy uses a detection lens that is not corrected for the visible, and does not use a proper tube lens, which limits its useful field of view.

c) The authors argue that OPM occupies a unique niche among LSFMs, in its ability to easily integrate into existing optical microscopy workflows. I am sympathetic to this argument, but am not convinced that the technique described here is conceptually new or even advantageous compared to the previous state-of-the-art in this field: Yang et al.5. As the authors correctly point out, the conceptual advance realized by Yang is that sticking a piece of high refractive index material between secondary and tertiary lenses enables better performance than previous OPM. Using a bespoke objective instead of the awkward coverslip/water construction used in Yang is definitely a step in the right direction towards making the method more practical and commercially viable but is not a conceptual advance. On top of this, Yang reports pre-deconvolution bead-based FWHM values of 316± 8nm and 339± 18 nm laterally and 596 ± 32 nm axially. The lateral values are ~10% worse than those reported here, which is sufficiently close that I have a very difficult time believing that the authors' work represents a notable advance over the previous work in Yang et al. i.e. the minor improvement they report in pre-deconvolution FWHM values is unlikely to make any qualitative difference to the kind of data/insight obtained against this previously published paper, nor does it justify a quantitative jump from “300 nm scale” to “200 nm scale”. Moreover, the axial FWHM value reported in Yang is ~25% better than the 823± 31 nm number reported here. Taken together these comparisons would suggest that this work represents a rotation in parameter space rather than an advance.

We are intimately familiar with the eSPIM (Bin Yang is also a co-author on this manuscript), and we consider our manuscript to be a major advance in making the eSPIM design routinely useful for a much larger regimen of specimens. Indeed, our field of view is 3.7-fold larger than that of eSPIM, which takes the system from a dedicated single cell imaging system to one that can handle neurons, embryos, and even tissues. We also introduce the first optogenetic module that provides the level of control necessary to stimulate cells with arbitrary 2D patterns synchronously with volumetric imaging. While our resolution is similar, our detection path is much more apochromatic, and our alignment much more robust. We can also image deeper into a specimen owing to the larger working distance of the primary objective. We now detail all of the advantages between our system and eSPIM in Supporting Note 1.

We have not focused in the initial manuscript on axial resolution and used a rather low NA setting, which would fit a wide variety of samples. However, for the sake of exploring what axial resolutions are possible, we systematically varied the effective excitation NA. We can achieve the same axial raw resolution as in Bin Yang’s work using a Gaussian beam, albeit we note that this mode is best for shallow, adherent cells (applies both to eSPIM and our work).

d) Perhaps most remarkably, the authors make no attempt to verify “200 nm scale” in any of their images, e.g. by empirically investigating if they can separate biological features at this distance or by running Fourier-based methods that provide pixel-based resolution maps6,7. Doing so would go a long way to clarifying if indeed they have achieved “200 nm scale” resolution vs. something more like “250 nm-scale” or even “300 nm scale” resolution in biological samples. Ideally these calculations would be performed with and without deconvolution, so that the effect of deconvolution is cleanly separated from the “raw” imaging performance. Speaking of raw performance, it is unclear to me which of the presented data is deconvolved vs. raw. Do the authors present any raw data anywhere in this paper? This point needs to be clarified.

It is commonly accepted in the field that careful analysis of sub-diffraction beads is the best method to evaluate instrument performance. Evaluating biological targets for which no ground truth exists is potentially problematic. Nonetheless, we now provide Fourier-based analyses (FRC and image decorrelation analysis) of instrument resolution with and without deconvolution on a biological specimen. We also now provide images of beads before and after deconvolution (Figure 2), as well as a non-deconvolved image of the ER and a *Drosophila* Embryo (Figure 3—figure supplement 1 and Figure 8—figure supplement 1.). We also show the optogenetic data in its raw form, including top and cross-sectional views. We also clearly state which data is in raw and which in deconvolved form in a table in Supplementary file 1.

e) What is the pixel size resulting from the imaging system, and the data presented here? It is concerning that all data, with the exception of the beads, seems to have been at least displayed in 115 nm x 115 nm x 100 nm voxels. Nyquist alone would dictate that “200 nm scale” lateral resolution is not possible.

The first two dimensions in the voxel (115 nm) are physically set by the magnification of the optical train. The third dimension (100 nm) is set by the step size of the laser scan. In this manuscript, we Nyquist sampled according to our raw resolution. For deconvolution, the bead data, and biological data where resolution was quantified, was interpolated onto a finer grid prior to deconvolution. Without resampling, the maximum resolution (i.e. where critical Nyquist sampling is achieved) is 230 nm. We did not resample most of the biological data before deconvolution, because its massive size would become limiting. The term “200nm scale resolution” has been removed from the manuscript.

3) The axial resolution values the authors report are significantly worse than state-of-the-art multiview LSFM techniques8-11, which can achieve up to a near twofold axial resolution enhancement compared to their (likely) over-optimistic deconvolved values of 570 nm. These references need to be cited as they set the bar as far as axial resolution, not the lattice light-sheet system. Speaking of axial resolution, why is the axial FWHM not reported in addition to the lateral FWHM measurements in Figure 2—figure supplement 5? Please provide all XYZ measurements before deconvolution, so the “raw” FWHM values are evident for the interested reader. This will help avoid confusion about likely over-aggressive deconvolution.

We have not optimized axial resolution in the first version of the manuscript, but rather used a low NA light-sheet to cover most samples. If we increase the light-sheet NA, we achieve an axial raw resolution of 587± 18nm, which is in line with eSPIM and a better raw resolution value than ever reported for the widely used lattice light-sheet microscope (in the most commonly used square lattice mode). We now report the axial raw resolution for different settings of the light-sheet. We feel that these values are competitive with the main other technique used to image such samples as presented in the manuscript, the lattice light-sheet microscope.

We have provided in Figure 1 raw XYZ measurements on FWHM, as well as the deconvolved values. In the new Figure 2, we now also provide raw XYZ resolution values across the field of view, and supplement them with an Image decorrelation measurement. Further, we report FRC and image decorrelation analysis for the lateral resolution before and after deconvolution as well. I think this is by far more forthcoming than many other manuscripts, including Multiview LSFM techniques, which only report resolution *after* iterative deconvolution. We believe that the reader can get a clear picture how much is gained in raw resolution with our setup, and what is gained by deconvolution. We feel we have been forthcoming and transparent about our resolution values, much more so than any other publication in the light-sheet field that we are aware of.

While multiview techniques do achieve excellent axial resolution, at least two objectives have to interface with the sample. It is in this manuscript the explicit goal to have only one objective interfacing with the sample to enable broader applicability of various samples. You simply cannot fit a multi-well plate or other bulky sample holders in the mm-sized sample volume of a multiview microscope.

We now explicitly mention the multiview methods in the Discussion: “In addition to its ease of use, the OPM describe here delivers spatial resolution that is on par with Lattice Light-Sheet Microscopy, albeit with a larger field of view and a faster volumetric imaging capacity. And unlike state-of-the-art multiview LSFM techniques that achieve a slightly better axial resolution, only a single imaging perspective is needed.”

It is worth mentioning that these multiview LSFM techniques necessitate the same RL algorithm for image fusion that is the source of much consternation here. For example:

Wu et al. (10.1038/nbt.2713) – Converges after 30 iterations.

Guo et al. (10.1038/s41587-020-0560-x) – New back projector that converges in ~10 iterations.

Wu et al. (10.1038/s41467-017-01250-8) – Reflective imaging geometry, report between 20 and 100 iterations of RL in Supplementary file 3.

Reviewer #3:The authors report a single objective light sheet (SOLS) microscope that provides unprecedented numerical aperture for light sheet imaging. The microscope utilizes a recently available solid-immersion objective lens to re-image an appropriately relayed tilted image plane onto a camera while encoding high NA information at smaller ray angles, thus allowing their capture without steric interference between the remote objective pair. This represents a natural progression from the eSPIM (Yang et al., 2019), which first illustrated the use of ray compression in SOLS. The application of this technology to several applications that would otherwise be extremely challenging in a light sheet/live-imaging context follow and illustrate that the microscope can image traditional sample preparations, in microfluidic chips and with high volumetric image rates. These applications provide an appropriate showcase for the technology and promising routes for further investigation without providing substantive biological conclusions. Given the technological focus of the article, I believe the article nonetheless warrants publication in eLife when the issues below are addressed. More generally, I believe this work and the recent developments on which it is built, will be of the utmost significance to the cell biology community in the coming years.The following major points should be addressed before publication:1) Having said that this is a technologically focused article, at least in terms of the innovation in the approach, I actually felt that the article was lacking in this regard. I am aware that much of the development around the solid-immersion objective has been reported via non peer-reviewed sources (GitHub etc.). However, as the first peer-reviewed study to utilize this technology, a more thorough description of the system would nonetheless seem warranted, particularly with regard to optical simulations, aberrations/useable viewing field (e.g. comparing simulations with observations) and a discussion of the various tradeoffs etc. This may seem outside the scope of a typical eLife publication and the manuscript as prepared but without it, the greatest achievement of this work (the microscope development, not the biology in this case) is sidelined. The authors are uniquely placed to be able to provide this information.

We agree that a more thorough description of the solid-immersion objective would be advantageous. We have detailed some design choices and their consequences in Appendices 1-3. However, in the spirit of supporting the open scientific ideals of Andrew York and Alfred Millett-Sikking (which we believe are well-aligned with the mission of *eLife*), we have agreed to let them keep many of these details on their GitHub page and to cite it accordingly as a Zenodo submission. Additional complications arise from non-disclosed proprietary information (that even the corresponding authors are not privy too) regarding the tertiary objective per an agreement between Calico, ASI, and Special Optics. Nonetheless, we now provide a bullet list of these differences in Appendix 1.

2) The authors include several comparisons with lattice light sheet microscopy that could be supported better with corresponding data and expanded discussion. This is noted with regard to the comments below. Similarly, I believe that the authors are uniquely able to provide the additional information via simulations and data that likely already exists e.g. fluorescent bead PSF measurements (e.g. Chang et al., 2019.) The authors would also benefit from comparing against some other reports in high-resolution light sheet microscopy e.g. Bessel beam (one/two-photon excitation, axially-swept light sheet microscopy).

The reviewer is correct that we are uniquely suited to be able to provide this data, but wanted to note that our Field Synthesis work was performed with a pair of NA 0.8/40X objectives, and not the much more expensive objectives used in the original lattice manuscript. As such, a comparison between our NA 1.35 and a NA 0.8 objective would not be fair. Also performing such a careful analysis is no small undertaking, especially during a pandemic. Nevertheless, we feel strongly that by providing the raw and deconvolved data for each microscope technique we have published, we have left the readers a more transparent way to compare optical performance between the different microscopes. Raw resolution information has been completely absent in the seminal lattice light-sheet publications, and it is our hope that a recent publication from our lab (Chang et al., 10.1364/OE.400164) is sufficient to shed some light on this topic.

Introduction : The authors note that previous LSFM implementations require mounting either on a small cover glass (e.g. lattice light sheet) or in agarose tubes (more classical SPIM-type systems). However, the authors fail to note that in the latter case, this is actually a superior mounting method e.g. for developing embryos as the soft surrounding environment promotes normal development (e.g. Kaufmann et al., Development, 2012), whereas a hard glass interface does not. I appreciate that these applications are outside the scope of what is presented here but I still think further clarification is warranted and that the desire to use traditional preparations is rather an issue for cell cultures and high-throughput imaging. The authors should nonetheless note that coverslip mounting may produce biological artifacts in the context of their earlier studies using collagen gels for mounting with a long working distance SPIM approach (Welf et al., Dev. Cell, 2016).

We have adopted this recommendation.

Results: Please comment on the choice of angle and the trade-offs associated with the 0 – 45 degree tilt.

Appendix 2 now discusses this. In short, you want to tilt as little as needed for best resolution and light-collection. However, the shallower the light-sheet, the longer it needs to be and the thicker it gets. Further, there will be less numerical aperture for the light-sheet generation remaining. A reasonable criterion we have come up with is to require that you find a tilt angle that leaves enough numerical aperture for the light-sheet left such that one can create a beam waist that is as large as the axial width of the detection PSF. For our system, this tilt angle turns out to be pretty close to 30 deg.

Subsection “Instrument Characterization”: Can you comment on how this compares to eSPIM (Yang et al., 2019). The lateral resolution appears to be slightly superior but the axial resolution seems to be lower (assuming a Gaussian beam for eSPIM). Is this a result of a longer thicker light sheet or a difference in how tilted the light sheet is with respect to the detection axis? This is surprising given the additional ca. 0.2 NA provided by the reported system.

We have estimated in the original submission the numerical aperture of the light-sheet illumination. We have now added a camera conjugate to the pupil plane for better estimation of the numerical NA of the light-sheet. We have systematically varied the NA of the light-sheet and we can match the axial resolution reported by Bin Yang. However, this comes at the expense of volumetric coverage, i.e. a smaller range in the z-direction (normal to the coverslip) can be covered with thinner light-sheets. This is however not different to Yang’s work.

and an 823 nm axial resolution?

Thank you for pointing this out.

Subsection “Instrument Characterization” final paragraph: Can you comment on the treatment here. Taking the efficiency simply as NA^2^ gives ca. 1.5, 1.9x respectively. This is without taking into the account the small NA loss relative to the 1.35 provided by the Si-immersion objective. Nevertheless, this analysis is welcome and generally underreported in single-objective light sheet microscopy.

We have addressed this now more carefully in the Appendix 3. The effective NA is conservatively estimate as 1.28. As such, the efficiency is only 35% higher than the NA 1.1. lens. This is a lower bound that neglects the slightly higher NA that is available in the direction normal to the tilt.

Figure 2 a) Is there a reason to choose a different color lookup table for this image?

This lookup table was selected as it allows visualization of both bright and dim structures. We now clearly state this in the figure legend.

Discussion second paragraph: Repetitive with regard to the Introduction.

We agree, but Introductions and Discussions often have some redundancy.

“…it has the highest lateral resolving power (~200 nm) in light-sheetmicroscopy. Indeed, owing to the narrow depth of focus provided by the optical design, the axial resolving power” This is the deconvolved resolution. It would be helpful to include PSFs from light sheet microscopes and report both the raw and deconvolved resolutions in each case. The authors previous efforts regarding field synthesis should yield the required data (Chang et al., 2019).

We now provide PSFs in the raw and deconvolved state. While we agree that it would be helpful to include PSFs from other microscopes, as previously stated, that requires substantial work, for which we currently lack the time in the lab due to the COVID pandemic.

Regarding contrast and axial resolution, which are tied together, the lattice light sheet supposedly maintains a thin profile over a larger distance than a purely Gaussian light sheet. The authors should report on the useable field of view when noting comparisons of contrast and resolution (and whether this is dominated by the requirements of a thin light sheet or rather the detection optics/remote focus scheme). The Gaussian sheet should produce better contrast at the focus whereas the lattice trades some contrast for field of view. However, the high detection NA in the SOLS case does mean that the light sheet will be thicker at waist than the depth of field and so contrast is also lost here. Again, the authors are uniquely placed to make these comparisons in a more rigorous manner. At the minimum, it would be helpful to report the length and thickness of the light sheet produced at the 0.2 – 0.3 NA (reported in the Materials and methods), to provide some measure of the effective field of view and contrast for the lattice and SOLS cases.

At the moderate NAs used for illumination with our OPM, Gaussian beams and the square lattice from Lattice Light-Sheet Microscopy (the most commonly used variant) have indistinguishable propagation lengths (in terms of FWHM and confocal parameter) and beam waists (see Chang et al., 10.1364/OE.400164). Thus, we would expect that a Gaussian beam actually provides superior contrast for a given field of view since it does not have side lobes structures that result in out-of-focus illumination.

We now provide measurements for the propagation length and NA of the illumination beams used in Figure 2. As can be seen at the highest illumination NA (0.34), the beam waist is evident in the axial resolution for beads suspended in an agarose gel. Here, the axial resolution is dominated by the light-sheet thickness. As the illumination NA decreases and the beam waist grows larger than the depth of focus, the axial resolution eventually approaches ~800 nm.

With regard to laser power requirements and synchronous multicolor imaging, this is largely a result of how lattice light sheets have typically been generated. The authors own previous research regarding field synthesis (which should be cited) has reduced these issues for anyone who may be choosing whether to construct a SOLS or lattice light sheet system. The primary issue remaining is the lack of isolation between the objectives and media and the steric issues.

Owing to objections from reviewer #2, we have removed the comment regarding laser power. However, we have adopted your recommendation that Field Synthesis should be cited, as well as mentioning the sterics and isolation of the media.

Subsection “Data Post-Processing”: Please comment on the time taken and computational resources required for shearing and deconvolving the datasets. Since the SOLS system appears a good fit for imaging facilities, the associated throughput rate is a consideration.

The research code we are using can clearly be optimized, and we believe that it may be possible to shear and deconvolve the data in real-time using GPU. Nonetheless, we now provide an approximate time for shearing and deconvolution in the data post-processing section.

[Editors’ note: what follows is the authors’ response to the second round of review.]

Reviewer #3:The authors have made substantive changes to the manuscript following the first round of reviews. I remain unswayed in my opinion that this work is suitable for publication in eLife given required improvements. I believe that the changes made have improved the manuscript.In particular, the assessment of the system PSF and resolving capability has been greatly improved by including the non-deconvolved PSFs, details of the deconvolution and Fourier ring correlation estimate of resolution. The discussion of true single objective light sheet systems and comparison with eSPIM places the reported technique in context and I believe helps to make its case far more than detracting from it. The appendices should go some way to assisting others in building such a system.There are still some changes based on my suggestions, which under ideal circumstances would be adopted. However, the author's arguments for not having done so are well reasoned. My primary concern was the frequent comparison with lattice light sheet microscopy. Here I still find the manuscript somewhat lacking but do appreciate that the authors field synthesis lattice light sheet system does not use the typical configuration of objective lenses making comparison difficult. The benefits of the reported system run deeper than pure resolution claims in any case and given the author's recent publication (Cheng et al. 10.1364/OE400164) reporting on the Gaussian to lattice light sheet comparison the omission here is reasonably justified.All things considered, I would recommend the article for publication.

We would like to thank the reviewer for once again taking the time to carefully evaluate our manuscript. In an effort to accommodate your last request, we now provide PSFs obtained by Dr. Talley Lambert at Harvard Medical School on a Betzig-derived Lattice Light-Sheet Microscope. As can be seen in Figure 2—figure supplement 3, and the quantitative measurements in the corresponding figure caption, the two PSFs are comparable.

Reviewer #4:First, I wish to commend the authors for their thorough revision. They have addressed many of my concerns and I feel better about the overall manuscript. The emphasis on field of view over resolution, the Fourier-based methods of assessing resolution, the citing of previous work, the additional details put forth in the appendices, and more accurately referring to their method as an OPM have all significantly improved the work. Here are my remaining concerns/comments:a) The authors now more fairly describe the advantages of their method over previous similar microscopes. They still do not clearly state what I see as weaknesses of their method, which seems important given the extensive comparisons to LLSM and other systems:i) relatively low sensitivity. In their revised manuscript, the authors now seem to admit that the advantage in light efficiency offered by the 1.35 NA lens is less than optical losses through their emission path. In sum, these losses would seem to imply a strictly lower sensitivity than the more “conventional” LLSM with 1.1 NA detection. Please discuss this point clearly in the main text, rather than relegating it to Materials and methods.

We have moved this back into the Microscope Design portion of the main text. The text now reads “Owing to the large number of optics, spurious reflections resulted in a 59% and 44% decrease in fluorescence transmission for laser scanning and stage scanning variants of the microscope, respectively, at 30-degrees. Transmission improved slightly (3%) when the optical train was arranged at 0-degrees.”

ii) Low effective NA vs. theoretical NA. I appreciate the appendix describing the NA calculations, but I am still not convinced that the number presented is an “effective NA”, i.e the NA of an equivalent widefield with primary lens using this NA. In particular, I am still bothered by the apparent loss of resolution compared to their widefield performance, which I commented on previously and repeat here:“Unfortunately, their own raw measurements suggest otherwise, as for the same beads in widefield mode (sans OPM detection) they report ~240 nm FWHM. Assuming that there are not additional aberrations in this primary lens, and the ~240 nm corresponds to the full 1.35 NA, in OPM mode the authors are relatively far from achieving diffraction-limited performance – they report 284 nm x 328 nm lateral FWHMs in the OPM configuration. Given that resolution scales with NA, these numbers would suggest (at best) an NA of 1.35 (240/284) = 1.14 in the direction normal to the scan, and an NA of 1.35 (240/328) = 0.99 in the direction along the scan. The combination seems to be no (or marginally) better than what is achieved in lattice light-sheet microscopy with a 1.1 NA lens.”

Ideally, the resolution of our microscope would be identical to the resolution of the microscope objective when operating in a widefield imaging mode. However, this expectation seems unrealistic since we are using the objective under such non-ideal conditions. Indeed, we are imaging above and below the nominal focal plane of the primary objective and using physically imperfect optics to create a replica of this fluorescence above and below the nominal focal plane of secondary objective. As previously stated, real lenses rarely reach the full resolution predicted by ideal models. Thus, some degradation should be expected.

It is very reasonable to suppose that NA would scale with the apparent size of a subdiffractive object, regardless of suboptimal Strehl ratio, MTF, hardware etc. And if the NA of the primary lens is effectively less than 1.35, this would seem to further diminish the “effective” NA that is achieved in this manuscript. This point is important to address properly, because it has implications both for comparisons to other microscopes and it suggests that the deconvolved numbers the authors present are still overly optimistic (e.g. if the NA is not in fact 1.28, physically it does not seem possible to achieve a deconvolved resolution of ~203 nm as in Figure 1H). I would suggest replacing “effective NA” with “theoretical NA” everywhere the former is referred to, and being very clear in the manuscript that there is still residual resolution loss: i.e. the theoretical NA is not achieved at the tilt settings used in this paper. The authors could prove me wrong by providing a suitably compelling measurement of the “effective NA” rather than the theory – presenting a theoretical argument does nothing to convince me that in fact they achieve this value, which their own measurements seem to contradict.

We understand the reviewer’s concern and have adopted the term theoretical NA throughout the manuscript. We also state in the main text that the system performs closer to the expected NA when reducing the tilt angle of the tertiary imaging system, and that there is some residual resolution loss when significant tilt is introduced in the tertiary imaging system. In the instrument characterization section, the text now reads “Of note, the choice of illumination angle is accompanied by trade-offs in light-sheet thickness, imaging depth, detection efficiency, and resolution (Appendix 2). Indeed, we observed a gradual loss in NA and thus resolution as our tertiary imaging system was adjusted from a 0 to a 30-degree tilt.”

b) Regarding deconvolution, I am still having difficulty understanding how ~203 nm resolution is possible with 115 nm voxels. Once more, Nyquist alone would seem to render this number incorrect. The authors' argument in using a factor 1.414 seems to boil down to: “other people have done it, so we should do it too”. This is not a good argument in my opinion, as it leads to a result that contradicts common sense. The authors digitally upsample the bead data before deconvolution, but I do not see how such digital upsampling could effectively beat Nyquist.

One interpretation of iterative deconvolution is that it makes an image sharper by extrapolating low frequency information into a higher frequency space where previously there was little or no information. By zero-padding the data, we are essentially introducing empty frequency space that the iterative deconvolution routine can extrapolate into. Importantly, the FRC data supports this view. Here, the frequency support where meaningful information is contained cuts off sharply, and the iterative deconvolution extends this domain to higher frequency values. Thus, it does not matter if that space into which sample information is extrapolated is created by zero-padding, or by finer spatial sampling (which also creates void frequency space, albeit with some white noise). This is supported by work by Dr. Rainer Heintzmann which shows that one can get ~1.5x resolution “out of band” with Richardson Lucy owing to non-negativity assumptions and iterations in real space (DOI: 10.1016/j.micron_2006.07.009), depending on the sample and imaging conditions. Please also note that diSPIM by necessity makes use of the extrapolation capability of RL deconvolution. The physical coverage in reciprocal space from two views leaves two large missing cones in diagonal directions. Instead of filling these voids physically with information by acquiring views from other orientations, they are filled by RL deconvolution. However, while diSPIM never showed that this out-of-band information is valid, we show here in our work evidence using the Fourier ring correlation.

Overall, we think that this discussion about what RL deconvolution can and cannot do is reserved to specialized journals and publications such as the one by Rainer Heintzmann. We mention this now in the Materials and methods section.

The authors in their rebuttal seem to suggest that presenting the raw values sans deconvolution is unusual in multiview LSFM (“I think this is by far more forthcoming than many other manuscripts, including Multiview LSFM techniques, which only report resolution after iterative deconvolution”). I am not sure what they are talking about: Wu 2013 (diSPIM), Keller 2015 (IsoView), classics in this area, state these values clearly in the main text.

We apologize for the miscommunication. The raw resolution for each view is often reported in multiview LSFM manuscripts, but the fused resolution is often only reported after Richardson-Lucy deconvolution. This likely results from the fact that without deconvolution, 6-8 views are necessary to sufficiently sample Fourier space (see DOI: 10.1364/OE.15.008029). Nonetheless, these manuscripts are insightful:

Wu et al., 2013:

Axial resolution (Supplementary Table 1) – 1.47 microns raw, 0.8 microns deconvolved (a factor of 1.83).

Arithmetic fusion (Supplementary Table 1) – 0.98 microns raw, 0.62 microns deconvolved (a factor of 1.58).

Joint fusion and deconvolution (Supplementary Table 1) – 0.33 microns axial resolution.

Acquisition (see Online Methods) – “In each 3D stack, 50 or 100 xy planes separated by az step of 1 micorn of 0.5 microns were imaged”

Upsampling (see Online Methods) – “View_A_ is upsampled (i.e., coarsely sampled axial pixels were linearly interpolated to obtain an isotropic voxel size of 0.1625 x 0.1625 x 0.1625 microns^3^”.

Conclusion – Wu et al. upsample their data from 0.5 microns or 1 micron to 0.1625 microns and report a resolution of 0.33 microns. Their deconvolution is much more aggressive than ours.

Chhetri et al. (not Keller), 2015:

Axial resolution (Supplementary Figure 6) – 3.01, 2.95, 2.63, or 3.34 microns, depending upon the imaging axis.

Arithmetic fusion – Not provided.

Multiview Deconvolution (Supplementary Figure 6) – Report a resolution of 410, 420, and 450 nm.

Acquisition – Lateral voxel size of 0.4 microns reported. Axial dimension not reported.

Upsampling (Methods, IsoView multiview image registration) – “In the first step of our registration procedure, we perform a coarse image alignment. After cubic interpolation for generating voxels with isotropic size…”

Conclusion – Chhetri et al. upsample their data and report a resolution of ~400 nm. Also appears that their deconvolution is much more aggressive than ours.

As such, we believe that your concerns about extending beyond Nyquist sampling should also be applied for their deconvolved results. But as stated before, this is not contested by theoretical work on iterative deconvolution.

c) The authors seem to acknowledge in their rebuttal that they do not achieve diffraction-limited performance (“we agree we do not achieve diffraction-limited performance”), but I still find multiple places in the manuscript where this is stated or implied. Please address this point.

We no longer state or imply that we achieve diffraction-limited performance throughout the text with a few exceptions. In Appendix 1, bullet point 2, we clearly state that the FOV is theoretically diffraction limited (not experimentally diffraction limited), which is accurate. Same with bullet point 3. The remainder of the text now states “near-diffraction limited”, which is accurate, and backed up by our wavefront measurements (Strehl = 0.97 at 0-degrees, and 0.91 at 30-degrees, see Figure 1—figure supplement 2).

d) Depth. What sets the effective depth limit of this technique, as presented here? 25-30 μm is relatively modest for a lens with a 300 μm working distance, so discussing theoretical and practical limits would be useful for a biologist who is searching for a practical solution for their sample.

The effective imaging depth depends on two separate factors. The first and most intuitive factor is the working distance of the primary objective, which as you state, is 300 microns. The second factor is the distance over which the remote focusing system properly functions, which influences extent of the W dimension in Figure 1. In both the linear and tilted configurations, we achieved ~60-80 microns of high-quality remote focusing, and this allowed us to recently image through the entire tail of a zebrafish. Importantly, one can still tile the data in the Z direction within the working distance of the objective, provided the sample has not a significant refractive index mismatch. In an effort to communicate this topic, we now clearly state these practical limits in the main text of the manuscript. Specifically, it reads in the instrument characterization section that “Theoretically, the maximum imaging depth of our remote focusing system is 60 microns, beyond which tiling in the Z-dimension can be performed until one reaches the working distance of the primary objective (300 microns)”.

e) The authors state numerous times that they achieve resolution on par with or better than LLSM. Please explicitly state the corresponding values reported in LLSM (or measure them with an LLSM with 1.1 NA detection), where relevant.

The seminal lattice manuscript, as well as many that followed, provide insufficient evidence to back up their resolution claims. There is however one exception (DOI:10.1038/nature22369) that clearly states that their lateral and axial resolution varies between 294-370 nm and 649-947 nm, respectively, depending on the laser illumination wavelength. Likewise, we recently published a manuscript (DOI:10.1364/OE.400164) that exhaustively evaluated light-sheet properties (thickness, confocal parameter, resolution, OTF, etc.) for both Lattice and Gaussian-based light-sheets. Cumulatively, these two manuscripts clearly show that we achieve resolution that is on par or better than LLSM. Nonetheless, we now provide further evidence of this in the form of PSFs acquired on a LLSM by Talley Lambert at Harvard Medical School (a system built by Betzig’s group). Again, these data show that we are indeed on par, or better than, a LLSM.

The text now states: “And for an oblique illumination angle of 30 degrees, these raw axial resolutions are similar or better than the 666 nm axial resolution reported for the square illumination mode of Lattice Light-Sheet Microscopy”.

“By comparison, the raw lateral resolution for Lattice Light-Sheet Microcopy for a GFPlike fluorophore is 312 nm…”.

f) “And unlike state-of-the-art multiview LSFM techniques that achieve a slightly better axial resolution, only a single objective and imaging perspective is needed (Guo et al., 2020; Wu et al., 2013).” Wu 2013 achieves ~350 nm axial resolution, Wu 2017 achieves ~300 nm axial resolution, and Guo 2020 enhances the effective axial resolution of LLSM to ~380 nm resolution. These are not “slightly better”, they are significantly better (indeed, the degree of improvement is more convincing than what is shown here relative to LLSM). Also, as I think the authors acknowledge in their rebuttal, they do not use a single objective, rather they use 3. Please remove “slightly” for accuracy and/or rethink this sentence.

As stated previously, multiview light-sheet microscopes only achieve axial resolutions better than ours after image fusion and deconvolution. As we clearly show, our best raw axial resolution of 580 nm is significantly better than the 1.47micron raw axial resolution reported by Wu et al., 2013. And, these manuscripts are much more aggressive with their deconvolution than we are.

But in an effort to be constructive and not endlessly debate what resolution gains come from iterative deconvolution, maybe let us discuss what is physically possible before any deconvolution. The highest axial raw resolution for light-sheet before deconvolution is about 380nm (Dean et al., 2015), our best resolution is 200nm worse. Dean et al. used an NA of ~0.8 for light-sheet generation, ours is less than half of that. We can agree that when a single objective is used for *both* illumination and detection, you have less angular range available than in a dual or multi-objective geometry.

The reviewer concern that we use three objectives is again semantics in our opinion. In our setup, there is only one objective that interfaces with the sample. As mentioned before, it is clear that when one uses multiple objectives to illuminate and detect the sample, then one can get more access from a greater angular range, at the expense of a complicated sample interface.

In an effort to communicate this more effectively, we now state that “And unlike state-of-the-art multiview LSFM techniques that achieve a better axial resolution after image fusion and deconvolution, only a single imaging perspective is needed.”

Reviewer #5:In this manuscript, "A versatile oblique plane microscope for large-scale and high-resolution imaging of subcellular dynamics", Sapoznik et al. describe a variant of an oblique plane microscope (OPM) that makes use of a custom designed tertiary objective to capture a greater portion of the emitted rays through the oblique refocusing objectives. OPM combined with light sheet is attractive because it offers the potential for optically sectioned, low phototoxicity imaging using only a single primary imaging objective with 180 degree physical access. As the authors note, this concept itself is not new and many descriptions and implementations have been described before (originally with Dunsby in 2008 and more recently with Bouchard 2015 and Yang 2019). Thus, the primary technical innovation here is the implementation of the custom-designed solid immersion lens.In general, I think this is a useful addition to the field and could be suitable for publication in eLife. An open top-light sheet microscope with increased sample accessibility is indeed useful. However, in its current form, the manuscript reads more like an aggressive sales pitch rather than a balanced discussion of the pros and cons of different microscope approaches. The presentation is over-dismissive of prior work, often mis-representing or cherry-picking specific comparisons to make this current instrument appear better, while neglecting to mention or discuss trade-offs wherein the instrument might perform worse. I noticed that the authors disclose financial relationships with companies that sell products that compete with commercial versions many of the instruments compared here. However, I feel that the readers of eLife would benefit if the authors focused on more thoroughly documenting their own scientific contribution, together with its trade-offs, rather than (often incompletely or inaccurately) characterizing prior work from others.

We appreciate that you believe that this is a useful addition to the field and agree with your sentiment that open top light-sheet microscopes are indeed useful. It was not our intention to be over-dismissive of prior work, but rather to clearly delineate the differences between our technique and that of others, which was necessary to address many concerns raised by the reviewers in the first revision of this manuscript. Importantly, with regard to the OPM technology described here, we do not have any competing financial interests:

1) We have worked with Applied Scientific Instrumentation to make many of the optical element and made it easier for other labs to get started building their own OPM systems. Nonetheless, this arrangement has been purely pro bono. We do not hold patents on OPM or SCAPE technology.

2) Discovery Imaging Systems, LLC was formed in an effort to sell a cleared tissue light-sheet microscope based upon our Axially Swept Light-Sheet Microscopy (ASLM) technology. However, this venture is pending until the patent for ASLM is finalized.

3) The remaining of the competing financial interests are pharmaceutical in nature and can be wholly attributed to Drs. Carlos Arteaga and Ariella Hanker.

1) The resolution claims here are provided by FHWM measurements of isolated beads and of xy (lateral) measurements using correlation based approaches on the images. Importantly, optical sectioning is not discussed at all. In this instrument, the resolution is provided by the high NA primary objective, but due to the thick lightsheets (0.16 NA Gaussian beams) used in most of the measurements, the lightsheets are substantially thicker than the depth of field for the detection. This would be readily apparent as the "missing cone" in the optical transfer function and I suspect it would also affect correlation based axial resolution measurements if they were conducted on cells or embryos rather than isolated beads. In contrast, axial resolution in several other approaches (Gaussian, Bessel, Lattice etc) is obtained by using a lightsheet that is thin-compared to the detection depth of field. The authors' own prior work, Dean et al. 2015, have highlighted the advantages of this approach, but there is no mention of the relationship between optical sectioning and resolution here.

As this reviewer is aware, all light-sheet microscopes are accompanied by complex trade-offs that include illumination confinement, field of view, lateral resolution, axial resolution, sensitivity, speed, optical sectioning, etc… However, recent work has shown that the most widely used square lattices does not improve optical sectioning or axial resolution relative to Gaussian beams (DOI: 10.1364/OE.400164 and 10.1364/BOE.11.000008), which is why we chose to proceed with the latter.

In our view, the most promising methods to improve optical sectioning, axial resolution and field of view is either light-sheet generation with two photon Bessel beams, structured illumination, or using the ASLM principle, all of which come with intrinsic drawbacks. In principle, both methods are compatible with our microscope, but we have not realized these options in our microscope yet. Other light-sheet types (hexagonal lattice, 1PE Bessel, Airy beams) come with a significant loss of optical confinement.

For most biological imaging performed, we used a Gaussian beam with an effective NA of 0.16 and an anticipated thickness and confocal parameter of 1.2 and 36.9 microns, respectively. While this thickness is larger than the depth of focus, we would like to note that it is only ~1.33-fold larger. We now explicitly mention this in the instrument characterization as follows: “For most biological experiments reported here, we used an illumination NA of 0.16, yielding a Gaussian beam that has a thickness and propagation length of 1.2 and 37 microns, respectively. Importantly, because the illumination beam is thicker than the depth of focus of the detection objective, optical sectioning (e.g., the ability to reject out-of-focus fluorescence) is slightly reduced.”

FWHM measurements in isolation are a limited picture of resolution. The authors should present optical transfer functions in the xy, xz, and yz planes to demonstrate true resolution and to what extent their system fills in the missing cone. They should also perform the raw image-based correlation measurements in the axial dimension (as they already do now for the lateral dimension) on a range of biological samples. This could be easily done with existing data.

We now provide OTFs in Figure 2—figure supplement 1 and demonstrate that our system does indeed partially fill the missing cone. For comparison, we also provide a widefield OTF, which clearly shows the singularity at the origin and the missing cone of information. There is in our opinion a vast difference between a widefield OTF and any of the light-sheet OTFs that we show. The widefield system cannot reject any background, and hence has a singularity at its origin, which makes any deconvolution attempts difficult.

Owing to complexities that arose from the correlated noise distribution of modern CMOS cameras, computational shearing of the data and the parallelepiped shape of the imaging volume, Fourier Shell correlations turned out to be complex and problematically variable (we spent ~2 months working on this during the first revision cycle). Consequently, we provided Fourier Ring Correlation and Decorrelation Analyses of the lateral resolutions, and these results were largely in agreement with the FWHM and resolution metrics based in frequency space. Given this agreement between the to approaches of resolution measurements in the lateral dimensions, we do not see any reason why the axial resolution measured in real space should be inaccurate.

2) The authors describe that, due to the optical path required for OPM, the instrument loses 71% of the photons from the sample before they hit the camera. I.E. it has a transmission efficiency of 29%. This is in stark contrast to the standard SPIM systems which due to the simple widefield detection optics should operate at very close to the ~80-90% transmission efficiency of commercial objectives. Further, it's not clear at what imaging angle this measurement was performed. This information is also important because the efficiency decreases further as the primary objective is utilized further away from its design standard angle of 0 degrees. In the appendix, the authors describe how the 1.2 NA of this system would somewhat compensate for this poor transmission over the 1.1NA lens used in other variants, but this analysis assumes that all rays have equal transmission efficiencies which generally isn't true. Thus even without any additional optics, an OPM system operating with an off-centered pupil will have a lower transmission efficiency than an objective with the same effective NA operating on axis.

For the previous submission, we performed the measurement at a 30-degree tilt. We have more systematically repeated this measurement by carefully matching the diameter of the alignment laser to the size of the primary objective back pupil, with all of the optics and filters present in the optical path with the exception of the primary objective and the camera. The laser and stage scanning variants had a 41% and 53% transmission when oriented at 30 degrees. Accordingly, the scan lens, galvo, tube lens combo was responsible for the additional 12% decrease in transmission between the two systems. Placing the stage scanning variant at a 0-degree tilt only improved the transmission by 3% to 56%. This verifies the design predictions for our tertiary detection objective (i.e. capturing most of the light-cone emanating from the secondary objective under a moderate tilt angle). We now detail these measurements in the Materials and methods section.

Experimental characterization of how the transmission efficiency between a 1.2 NA objective operating with a centered pupil vs. a higher NA objective with an effective 1.2 NA due to pupil decentering with OPM and associated optics, and how this declines with the OPM angle would be extremely useful for the field. The authors are ideally poised to make these type of measurements.Regardless, this is an important discussion, so I'm unclear why the measurements are not mentioned in the main text under "Instrument Characterization". They are instead provided in the fourth paragraph of Materials and methods section under "Laser Scanning Microscope Setup". Given that microscope end users are willing to pay thousands more for a back thinned camera with a 10% increase in quantum efficiency than a non-back thinned unit, the authors should mention that the OPM implementation here comes at the cost of reduced detection efficiency in the main text. The statement that the instrument is "sufficiently sensitive to detect single molecules (data not shown)" does not adequately address this concern. Especially when other approaches have clearly demonstrated their utility for live cell and super-resolution single molecule imaging with both dyes and fluorescent proteins alike. It's also not clear how the authors define "without obvious signs of phototoxicity". How was this measured and what is considered obvious?

In the final paragraph of the Discussion, we now mention the reduced collection efficiency. Also, we have removed the statements regarding the detection of single molecules and phototoxicity.

3) I have concerns about how prior methods are discussed. For example, Lattice Lightsheet Microscopy is not a single instrument, but a general description of the use of excitation patterns based on optical lattices for microscopy. The specific choices for excitation objective and detection objective as well as the type of lattice light sheet and extent of optical confinement used will determine the resolution and optical sectioning and can be chosen/optimized for specific applications. Thus, statements like "Lattice Lightsheet microscopy has a resolution of xx" or "requires a 5mm coverslip" are no more accurate than saying "Gaussian beam microscopy has a resolution of xx and requires agarose sample embedding". It all depends on how one decides to configure an instrument and for what purpose they choose to balance the trade-offs. It's fine to say that a specific publication reported certain values, but this should be accompanied by some context. In many cases, design considerations may have been chosen to make an instrument that is optimized for a different purpose or with additional features than the one presented here.

In theory, lattice light-sheet is a generic imaging approach that is compatible with a range of objective and thus sample preparations. However, in practice, it is built and sold in a manner that does require 5 mm coverslips. This is how it was implemented by Betzig, 3i, and over 120 different labs. Also, the way that Lattice light-sheet is described in the literature makes comparisons challenging. It can be operated in a structured illumination, hexagonal, and square illumination modes, each with different inner and outer NAs, beam lengths, thicknesses, etc. For the comparisons here, we are referring the square illumination mode that accounts for >95% of the use cases (see Table S1, DOI: 10.1038/s41592-0190327-9), and which we have performed careful measurements (DOI: 10.1364/OE.400164). We now try to specify that we are referencing the axial resolution of the square illumination mode (the lateral resolution remains unaffected). Also, the text now reads “Sample preparation is an additional problem, as the orthogonal geometry of LSFM systems often sterically occludes standard imaging dishes such as multi-well plates.”